# Scaling Computer-Use Grounding via User Interface Decomposition and Synthesis

**Tianbao Xie** [*][h] **Jiaqi Deng** [*][h] **Xiaochuan Li** [*][h] **Junlin Yang** [*][h] **Haoyuan Wu** [h] **Jixuan Chen** [h]

**Wenjing Hu** [h] **Xinyuan Wang** [h] **Yuhui Xu** [s] **Zekun Wang** [h] **Yiheng Xu** [h] **Junli Wang** [h]

**Doyen Sahoo** [s] **Tao Yu** [†][h] **Caiming Xiong** [†][s]

[h] The University of Hong Kong    [s]Salesforce AI Research

## Abstract

Graphical user interface (GUI) grounding, the ability to map natural language instructions to specific actions on graphical user interfaces, remains a critical bottleneck in computer use agent development. Current benchmarks oversimplify grounding tasks as short referring expressions, failing to capture the complexity of real-world interactions that require software commonsense, layout understanding, and fine-grained manipulation capabilities. To address these limitations, we introduce OSWORLD-G, a comprehensive benchmark comprising 564 finely annotated samples across diverse task types including text matching, element recognition, layout understanding, and precise manipulation. Additionally, we synthesize and release the largest computer use grounding dataset JEDI, which contains 4 million examples through multi-perspective decoupling of tasks. Our multi-scale models trained on JEDI demonstrate its effectiveness by outperforming existing approaches on ScreenSpot-v2, ScreenSpot-Pro, and our OSWORLD-G. Furthermore, we demonstrate that improved grounding with JEDI directly enhances agentic capabilities of general foundation models on complex computer tasks with state-of-the-art performance, improving from 23% to 51% on OS-World. Through detailed ablation studies, we identify key factors contributing to grounding performance and verify that combining specialized data for different interface elements enables compositional generalization to novel interfaces. All benchmark, data, checkpoints, and code are open-sourced and available at `https://osworld-grounding.github.io`.

## 1 Introduction

Graphical user interface (GUI) grounding, the ability to accurately map natural language instructions to specific actions (including the positions of on-screen elements), is a cornerstone for computer use agents to effectively interact with GUIs on devices such as mobile phones and desktop computers. It plays a critical role, whether as an isolated component of human-machine interaction, a facilitator of multi-model collaboration agents, or a means to enhance end-to-end models.

Achieving practical GUI grounding requires software commonsense (e.g., understanding the meaning of icons, the functions of components, and specific software knowledge), layout understanding (e.g., interpreting a sidebar on one side or elements under a panel) and fine-grained component manipulation (e.g., adjusting a slider or selecting text on character level). Knowledge and grounding together enable comprehension and interaction. Additionally, rejecting infeasible instructions (e.g., mistaking Thunderbird for Firefox) is necessary to avoid unrecoverable states. Previous work around GUI grounding oversimplify these tasks as short referring expressions. Such descriptions are clear

---

* Equal contribution. †Corresponding authors. Work mainly done during TX's internship in Salesforce.

39th Conference on Neural Information Processing Systems (NeurIPS 2025) Track on Datasets and Benchmarks.

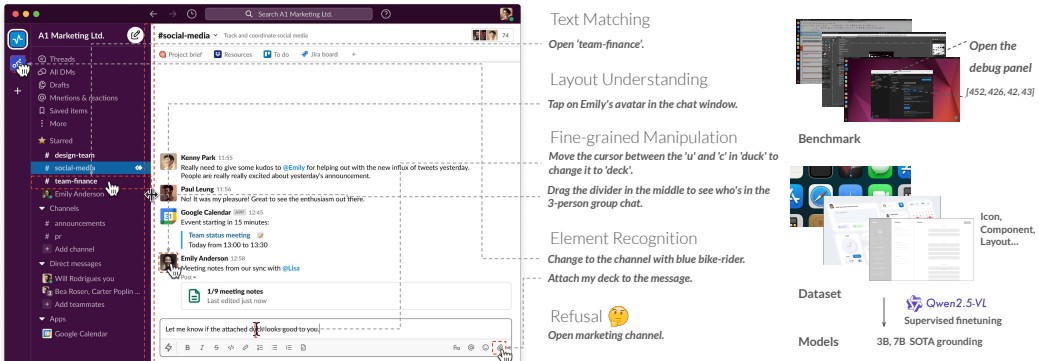

Figure 1: We have developed a comprehensive benchmark comprising 564 examples that cover the diverse task types that previous work has overlooked. Additionally, we synthesize and release the largest computer use grounding dataset containing 4 million examples, and train models that achieve state-of-the-art performance on this dataset.

but leave a gap with real-world requirements. As a result, existing benchmarks like ScreenSpot-v2 [10, 45] show saturation at early stages (~90%) accuracy by recent approaches [32] together with the progress of vision-language models (VLMs) [see 9, 42, 6, *i.a.*], primarily focusing on simple instructions to locate referenced elements in screenshots. Current evaluation approaches either lack nuance in their assessment criteria or artificially inflate difficulty through unnatural conditions, such as ScreenSpot-Pro's extreme resolutions that rarely occur in typical computing environments. Achieving practical grounding requires software context awareness and fine-grained manipulation capabilities for diverse GUI elements including dropdown menus, tabbed interfaces, scrollbars, and context-sensitive controls that have not been adequately measured or explored. On the data side, the primary capabilities of current grounding models arise from structured text and screenshot correspondences found on webpages(e.g., SeeClick [10], UGround [15], OmniParser [26], OS-Atlas [45], Aria-UI [49]). Alternatively, they rely on manually annotated data (e.g., Aguvis [48], UI-TARS [32]). The former can capture coarse-grained element understanding signals for webpage but lacks fine-grained operational capabilities for UI elements. The latter, due to high manual annotation costs, struggles to scale effectively.

To better assist the community in addressing GUI grounding challenges, we start with benchmarks and data as shown in Figure 1. We develop the OSWORLD-G, comprising 564 finely annotated samples that systematically cover text matching, element recognition, layout understanding, fine-grained manipulation and infeasibility, with annotations for the element types required to solve each task. On the data side, we collect and synthesize the largest-scale open grounding dataset JEDI in the web and desktop domain through multi-perspective decoupling of tasks. Additionally, we train multi-scale models on this dataset to validate its effectiveness.

Our evaluation on ScreenSpot-v2, ScreenSpot-Pro and OSWORLD-G demonstrates that our approach significantly outperforms existing models in aspect of grounding ability. Beyond standalone grounding performance, we show that improved grounding directly translates to enhanced agentic capabilities on complex tasks in OSWorld [46] and WindowsAgentArena [7] benchmarks. Through detailed ablation studies, we identify key factors that most significantly contribute to grounding performance, providing insights for future data collection and training efforts to enhance such abilities. Our case studies verify the effectiveness of our decomposition hypothesis, demonstrating that combining specialized data for different interface elements enables compositional generalization to novel interfaces.

## 2 Approach

**Task Definition** A **Multimodal Agent** is an AI system that visually perceives the GUI from the environment. At each step $t$, it receives a visual observation $O_t$ (e.g., pixel data $\in \mathbb{R}^{H \times W \times C}$) and executes an action $a_t$ based on a natural language instruction $I$ and its current observation (and potentially history). The agent learns a policy $\pi : (O_t, I, \text{state}_t) \rightarrow a_t$ to generate the sequence of actions $A = \{a_1, \ldots, a_n\}$, purely from visual perception without access to the GUI's underlying code

or APIs. An action $a_t$ consists of an action type (e.g., `click`, `move_to`, `type`) and action parameters that typically involve coordinates, represented as either a point $(x, y)$ or a bounding box $(x, y, w, h)$ to specify the target GUI element. **GUI Grounding** represents the core capability enabling the policy $\pi$ to function effectively at each step $t$. Given a potentially step-specific interpretation or sub-instruction $I_t$ (derived explicitly or implicitly from $I$) and the current observation $O_t$, grounding is the process of mapping these inputs to the specific, executable action $a_t$. Achieving accurate grounding for each $(I_t, O_t)$ pair is a fundamental objective in training the agent and a key determinant of the policy's success on the overall task.

## 2.1 OSWORLD-G

### 2.1.1 Benchmark Construction

We sample screenshots from the rollout of previous models on OSWorld [46], as this is currently one of the most widely adopted benchmark environments for evaluating computer use agents, covering diverse elements, fine-grained components, and rich layouts. The screen size is set to 720p and 1080p. Following ScreenSpot and ScreenSpot-Pro, we annotate these screenshots with instructions and corresponding bounding boxes. Even for fine-grained manipulation tasks such as text editing, we can identify specific pixel regions that are sufficient for creating appropriate bounding boxes. For evaluation, we determine whether the coordinates in the agent's predicted actions fall within the annotated bounding boxes, and calculate accuracy based on this spatial containment criterion. We utilize the CVAT [2] platform to collect annotations of objects corresponding to instructions. Each annotation is performed by individuals highly familiar with the software details and is verified through actual testing in the real software, particularly for edge cases. Following the initial annotations, we conduct multiple verification rounds based on feedback from preliminary experiments. For each example in OSWORLD-G, we assign a fine-grained tag that identifies the element types required to solve the example. Additionally, we provide a refined annotation for each example that rephrases the original instructions to decompose the necessary GUI knowledge needed to complete the task. In total, we collect 564 samples, annotated with 32 different UI-types, each with a paraphrased instruction that requires no software knowledge to execute. The average annotation time per sample is approximately 0.5 human-hours. We provide the annotation workflow in the Appendix A.1.4.

### 2.1.2 Data Types

Leveraging the fine-grained element type tags, we categorize tasks into capability dimensions that directly reflect core model competencies: *text matching*, *element recognition*, *layout understanding*, *fine-grained manipulation*, and *refusal handling*, as presented in Table 1.

**Text Matching & Element Recognition** Most cases in GUI grounding require simply text matching and element recognition as two fundamental capabilities. Text matching involves grounding actions according to explicit textual information provided in instructions (e.g., "Select 'As Attachment'"). This requires matching the specified text to locate the appropriate screen region. Element recognition encompasses multiple aspects of visual understanding: identifying visual patterns such as icons or images (e.g., "Click on Ellipse icon"), and importantly, recognizing elements based on their implied functionality rather than explicit labels. For example, recognizing a "Save" button by its floppy disk icon, a "Settings" option by its gear icon, or a "Search" function by its magnifying glass symbol—all cases where the agent must associate visual elements with their functional purpose, even when no explicit text label is present.

Table 1: Distribution of examples in the OSWORLD-G benchmark categorized by GUI grounding capabilities and their corresponding interface element types. Full table can be refer to Appendix A.1.1

| Capabilities | Element Types | # of Examples |
|---|---|---|
| Text Matching | Label | 268 |
| Element Recognition | Icon, Image, Button | 337 |
| Layout Understanding | Tab, Menu Bar, Dropdown Menu, Panel/Container, . . . | 252 |
| Fine-grained Manipulation | Slider, Stepper, Text Field, Input Box, Divider, Table, . . . | 154 |
| Refusal | – | 54 |

---

[2] https://app.cvat.ai/

**Layout Understanding** GUIs are typically designed with modular structures. Knowledge of layout hierarchy is critical to locate elements precisely, as visually similar elements may exist across different modules, and describing elements often requires referencing their position within the layout. For instance, instructions like "Close the top notification bar" require correct identification of the notification bar area, as multiple similar close buttons may appear throughout the interface. Other cases require identification of toolbars, panels, pop-up windows, and other common GUI modules.

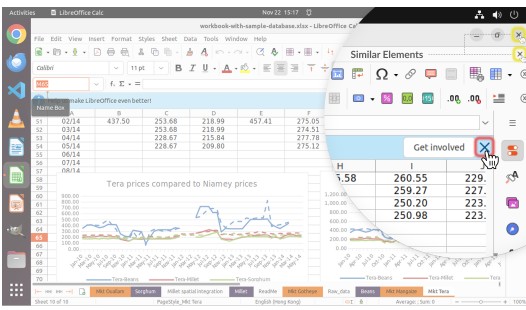

Close the top notification bar

Figure 2: Example of layout understanding case in OSWORLD-G.

**Fine-grained Manipulation** Computer use agent tasks frequently involve text editing operations. Instructions such as "Select the place between the world 'person' and the number '1'" require precise cursor placement between specific letters, which may occupy only a small portion of the screen. Such actions demand the ability to perform operations with high precision within relatively small screen regions. Beyond text, this capability extends to interaction with compact components like sliders, steppers, table cell and other small elements.

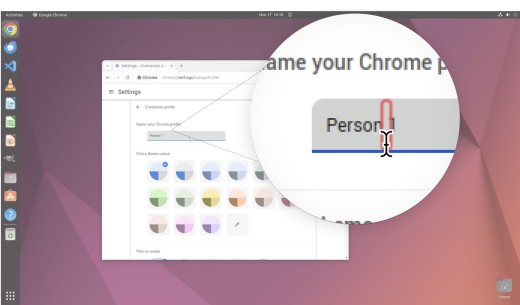

Select the place between the word "person" and the number "1"

Figure 3: Example of fine-grained manipulation case in OSWORLD-G.

**Infeasible** Certain tasks may arise from hallucinated or incorrect low-level user instructions or automated planning suggestions. An example could be an instruction like, "Click to open the Firefox browser," when the shown screenshot does not contain a Firefox icon or any visible reference to it. A distinct subset of OSWORLD-G tasks with 54 examples explicitly highlights these infeasible scenarios. These tasks are valuable for evaluating a system's ability to reject impossible instructions gracefully, preventing errors and ensuring safer, more robust interactions.

## 2.2 JEDI Data Construction

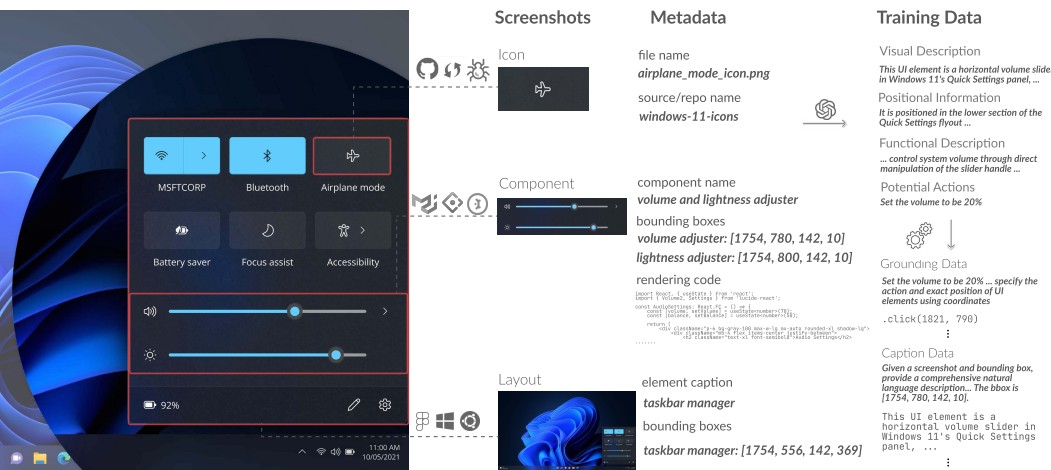

Figure 4: An overview of the synthetic data generation pipeline, demonstrating how screenshots and metadata are collected and synthesized, and subsequently converted into training data.

To enable robust GUI grounding, we construct the world's largest multimodal dataset tailored for computer-use grounding scenarios, containing 4 million newly synthesized examples. Our grounding data collection process centers on gathering pairs of *screenshots* and *metadata* (information such as filename, rendering code, element bounding box, *etc*.), which are then further transformed into *training data* which contains queries and corresponding answers for VLMs to learn from it. Previous methods in Figure 4 provides an overview of this pipeline.

### 2.2.1 Icon

*Icons* are essential visual elements in graphical user interfaces that convey functionality through compact, recognizable imagery. To create a comprehensive collection of icons and corresponding metadata for grounding, we employ three complementary data collection strategies.

**GitHub Repositories and Specialized Icon Websites** Many open-source software projects archive their design icons within GitHub repositories. To acquire a varied collection, we systematically mine repositories containing the key term such as "icon" applying filtering criteria including star count, quantity of icon images, and temporal relevance. This yield icons representing various design paradigms such as flat design, fluent design, and skeuomorphism. To supplement our collection with production website icons, we implement a targeted web crawling pipeline that identifies and extracts icon elements from popular websites across various categories, capturing both visual assets and associated metadata including class names, aria labels, and contextual information. We leverage these icons by generating detailed descriptions through LLMs and creating training scenarios where models identify target icons based on textual descriptions. This comprehensive approach provides access to contemporary icons in their natural context, allowing us to capture emerging design patterns not yet available in open-source repositories.

**Reverse Engineering Software** To address the gap in desktop software icons, we employ reverse engineering techniques using specialized tools like IconsExtract to extract icons directly from executable files, DLLs, and system libraries across Windows, macOS and Ubuntu. We target a diverse range of software including Windows system applications and commonly used desktop applications. This method provide access to thousands of production-quality icons representing real-world software functionality.

### 2.2.2 Component

A *component* refers to a functional unit composed of icons, UI elements, and text, collectively enabling specific modes of computer-based interaction. Components serve as fundamental interaction units essential for user engagement in digital environments.

**Synthesis Process** We collect screenshots and associated metadata primarily through a code-and-rendering pipeline. By leveraging mainstream production-level UI component libraries commonly used in front-end development (e.g., Material UI), we select components and use their example code as the base code. We then employ LLMs to synthesize functional cases for specific tasks (such as a slider for air conditioning control) using the base code as context. We render these within a React application to obtain visual screenshots and extract corresponding metadata, such as the element position tree, built-in component names, and coordinates. This approach allows us to generate diverse component examples with precise ground truth source code.

**Real-world Augmentation** We observe that common interactive behaviors such as scrolling a webpage, clicking a cell in spreadsheet or resizing a text box in slides are underrepresented in code-based libraries. To address this gap, we further source real-world screenshots from existing websites and applications. For these, we utilize HTML parsing and application-specific tools (e.g., `python-pptx`) to extract structured metadata. All the implementation details can be found in the Appendix A.2.3.

### 2.2.3 Layout

A *layout* refers to the spatial arrangement of UI elements and components within an application or across the entire operating system. Layout understanding is crucial for tasks that require reasoning about the overall structure of the screen, enabling agents to interpret and interact with complex, multi-element interfaces at the application or system level.

**Prototype Designs** UI prototype platforms such as Figma [3] provide numerous website and application design templates, including authentic specifications for production applications like VSCode, Zoom, and Microsoft 365. These designs offer valuable ground truth information, as each element includes designer-specified bounding boxes, component types, and functional descriptions. By using the official APIs of these platforms, we exported the designs as high-quality images while preserving their structured metadata, including hierarchical relationships between elements and positional data.

**Real-World Application Screenshots** To further improve scalability and diversity, we supplement our dataset with raw screenshots captured from real-world applications running on operating systems. We collect these screenshots by leveraging agent rollout data from OSWorld and WindowsAgentArena. Subsequently, we utilize the object detection model from OmniParser-v2 to generate bounding boxes for interface elements, thereby obtaining the necessary metadata.

### 2.2.4 Data Processing

After obtaining screenshots (icons, components, layouts) and metadata (filenames, paths, rendered source code, UI designer annotations, *etc.*), we convert them into an image-text-to-text multimodal question-answering format, creating richer and more natural language-oriented data suitable for VLM training. Our processing approach remain consistent across the different data types. We employ a VisualSketchpad [17]-like prompting methodology with models such as GPT-4o and Claude to generate enriched annotations based on the original screenshots and metadata, describing both appearance and functionality. We construct two complementary training formats: (1) *grounding format*, where the model receives a screenshot with instructions and must predict actions or relevant bounding boxes; and (2) *description format*, where the model receives a screenshot with bounding boxes and must provide descriptive information. For screenshots with multiple potential query-answer pairs, we compress them into single conversation to improve training efficiency.

### 2.2.5 Supplementary Training Data

To enhance the model's ability to identify and reject infeasible actions, we construct a refusal part in out dataset by mismatching existing instructions with unrelated screenshots, yield over 2.6 million examples. We further sample and manually inspect a subset of these examples to verify that the vast majority indeed reflects truly infeasible actions. In addition, we integrate and unify new datasets from previous work (human-labeled or synthesized) such as SeeClick, OS-Atlas, follow the practice from Aguvis [48]. We observe that synthetic data obtained directly from the Internet such as SeeClick, OS-Atlas contain noisy examples, we use UI-TARS-72B to filter them and keep the labeled and predicted matching part of the data. Full data statistics in Table 9.

## 3 Experiments

We first adapt previous benchmarks for testing our data effectiveness. We adapt different sizes of the latest Qwen2.5-VL [6] as our backbone model, set the maximum pixel limit to approximately 1080p. Model finetuning takes approximately 20 hours for the 3B model, and 30 hours for the 7B model, conducted using cluster of 128 CPU cores, 512GB memory, and 64 NVIDIA H100 GPUs.

### 3.1 Grounding Ability

We select several benchmarks for GUI grounding. The most commonly used benchmarks in the past include ScreenSpot-v2 (Table 2), ScreenSpot-Pro (Table 3), which focuses on high-resolution and professional software charts, UI-Vision [29] (Table 4), which focuses on fine-grained evaluation of computer use agents in real-world desktop environments, and OSWORLD-G (Table 5), which we use to evaluate model performance on fine-grained and functional components.

The results show that fine-tuning existing open-source models on our data achieves state-of-the-art performance, surpassing other dedicated computer use model such as Operator (unpublished data and model) and UI-TARS (unpublished data) with a small model size. On OSWORLD-G, we observe that models generally achieve the highest accuracy on examples involving text matching, outperforming their abilities in element recognition and layout understanding, with the lowest performance observed in fine-grained manipulation tasks. Notably, although we included refusal data during training to encourage the model to reject instructions referring to elements not present on the screen, the model rarely produces refusal responses. Similarly, in all models except Gemini-2.5-Pro, especially those specifically trained for computer-use tasks, refusal predictions are consistently absent.

---

[3] https://www.figma.com/

Table 2: Comparison of various planners and grounding methods on ScreenSpot-v2. The highlighted column presents the overall average performance across all categories

| Planner | Grounder | Mobile | | Desktop | | Web | | Avg |
|---|---|---|---|---|---|---|---|---|
| | | Text | Icon/Widget | Text | Icon/Widget | Text | Icon/Widget | |
| - | SeeClick | 78.4 | 50.7 | 70.1 | 29.3 | 55.2 | 32.5 | 55.1 |
| | OS-Atlas-Base-7B | 95.2 | 75.8 | 90.7 | 63.6 | 90.6 | 77.3 | 85.1 |
| | UI-TARS-7B | 96.9 | 89.1 | 95.4 | 85.0 | 93.6 | 85.2 | 91.6 |
| | UI-TARS-72B | 94.8 | 86.3 | 91.2 | 87.9 | 91.5 | 87.7 | 90.3 |
| | Operator | 47.3 | 41.5 | 90.2 | 80.3 | 92.8 | 84.3 | 70.5 |
| | Qwen2.5-VL-3B | 93.4 | 73.5 | 88.1 | 58.6 | 88.0 | 71.4 | 80.9 |
| | Qwen2.5-VL-7B | 97.6 | 87.2 | 90.2 | 74.2 | 93.2 | 81.3 | 88.8 |
| | Qwen2.5-VL-32B | 97.9 | 88.2 | 98.5 | 79.3 | 91.2 | 86.2 | 91.3 |
| GPT-4o | OS-Atlas-Base-7B | 96.2 | 83.4 | 89.7 | 69.3 | 94.0 | 79.8 | 87.1 |
| | OmniParser-v2 | 95.5 | 74.6 | 92.3 | 60.9 | 88.0 | 59.6 | 80.7 |
| | JEDI-3B | 96.6 | 81.5 | 96.9 | 78.6 | 88.5 | 83.7 | 88.6 |
| | JEDI-7B | 96.9 | 87.2 | 95.9 | 87.9 | 94.4 | 84.2 | 91.7 |

Table 3: Comparison of models on ScreenSpot-Pro. The highlighted column presents the overall average performance across all categories.

| Agent Model | Development | | | Creative | | | CAD | | | Scientific | | | Office | | | OS | | | Avg | | |
|---|---|---|---|---|---|---|---|---|---|---|---|---|---|---|---|---|---|---|---|---|---|
| | Text | Icon | Avg | Text | Icon | Avg | Text | Icon | Avg | Text | Icon | Avg | Text | Icon | Avg | Text | Icon | Avg | Text | Icon | Avg |
| SeeClick [10] | 0.6 | 0.0 | 0.3 | 1.0 | 0.0 | 0.6 | 2.5 | 0.0 | 1.9 | 3.5 | 0.0 | 2.0 | 1.1 | 0.0 | 0.9 | 2.8 | 0.0 | 1.5 | 1.8 | 0.0 | 1.1 |
| Qwen2-VL-7B [42] | 2.6 | 0.0 | 1.3 | 1.5 | 0.0 | 0.9 | 0.5 | 0.0 | 0.4 | 6.3 | 0.0 | 3.5 | 3.4 | 1.9 | 3.0 | 0.9 | 0.0 | 0.5 | 2.5 | 0.2 | 1.6 |
| ShowUI-2B [23] | 16.9 | 1.4 | 9.4 | 9.1 | 0.0 | 5.3 | 2.5 | 0.0 | 1.9 | 13.2 | 7.3 | 10.6 | 15.3 | 7.5 | 13.5 | 10.3 | 2.2 | 6.6 | 10.8 | 2.6 | 7.7 |
| CogAgent-18B [16] | 14.9 | 0.7 | 8.0 | 9.6 | 0.0 | 5.6 | 7.1 | 3.1 | 6.1 | 22.2 | 1.8 | 13.4 | 13.0 | 0.0 | 10.0 | 5.6 | 0.0 | 3.1 | 12.0 | 0.8 | 7.7 |
| Aria-UI [49] | 16.2 | 0.0 | 8.4 | 23.7 | 2.1 | 14.7 | 7.6 | 1.6 | 6.1 | 27.1 | 6.4 | 18.1 | 20.3 | 1.9 | 16.1 | 4.7 | 0.0 | 2.6 | 17.1 | 2.0 | 11.3 |
| Claude [3] | 22.0 | 3.9 | 12.6 | 25.9 | 3.4 | 16.8 | 14.5 | 3.7 | 11.9 | 33.9 | 15.8 | 25.8 | 30.1 | 16.3 | 26.9 | 11.0 | 4.5 | 8.1 | 23.4 | 7.1 | 17.1 |
| Operator [30] | 50.0 | 19.3 | 35.1 | 51.5 | 23.1 | 39.6 | 16.8 | 14.1 | 16.1 | 58.3 | 24.5 | 43.7 | 60.5 | 28.3 | 53.0 | 34.6 | 30.3 | 32.7 | 45.0 | 23.0 | 36.6 |
| OS-Atlas-7B [45] | 33.1 | 1.4 | 17.7 | 28.8 | 2.8 | 17.9 | 12.2 | 4.7 | 10.3 | 37.5 | 7.3 | 24.4 | 33.9 | 5.7 | 27.4 | 27.1 | 4.5 | 16.8 | 28.1 | 4.0 | 18.9 |
| UGround-V1-7B [15] | - | - | 35.5 | - | - | 27.8 | - | - | 13.5 | - | - | 38.8 | - | - | 48.8 | - | - | 26.1 | - | - | 31.1 |
| UI-TARS-2B [32] | 47.4 | 4.1 | 26.4 | 42.9 | 6.3 | 27.6 | 17.8 | 4.7 | 14.6 | 56.9 | 17.3 | 39.8 | 50.3 | 17.0 | 42.6 | 21.5 | 5.6 | 14.3 | 39.6 | 8.4 | 27.7 |
| UI-TARS-7B [32] | 58.4 | 12.4 | 36.1 | 50.0 | 9.1 | 32.8 | 20.8 | 9.4 | 18.0 | 63.9 | 31.8 | 50.0 | 63.3 | 20.8 | 53.5 | 30.8 | 16.9 | 24.5 | 47.8 | 16.2 | 35.7 |
| UI-TARS-72B [32] | 63.0 | 17.3 | 40.8 | 57.1 | 15.4 | 39.6 | 18.8 | 12.5 | 17.2 | 64.6 | 20.9 | 45.7 | 63.3 | 26.4 | 54.8 | 42.1 | 15.7 | 30.1 | 50.9 | 17.5 | 38.1 |
| Qwen2.5-VL-3B | 38.3 | 3.4 | 21.4 | 40.9 | 4.9 | 25.8 | 22.3 | 6.3 | 18.4 | 44.4 | 10.0 | 29.5 | 48.0 | 17.0 | 40.9 | 33.6 | 4.5 | 20.4 | 37.8 | 6.6 | 25.9 |
| Qwen2.5-VL-7B | 51.9 | 4.8 | 29.1 | 36.9 | 8.4 | 24.9 | 17.8 | 1.6 | 13.8 | 48.6 | 8.2 | 31.1 | 53.7 | 18.9 | 45.7 | 34.6 | 7.9 | 22.4 | 39.9 | 7.6 | 27.6 |
| Qwen2.5-VL-32B | 74.0 | 21.4 | 48.5 | 61.1 | 13.3 | 41.1 | 38.1 | 15.6 | 32.6 | 78.5 | 29.1 | 57.1 | 76.3 | 37.7 | 67.4 | 55.1 | 27.0 | 42.3 | 63.2 | 22.5 | 47.6 |
| JEDI-3B | 61.0 | 13.8 | 38.1 | 53.5 | 8.4 | 34.6 | 27.4 | 9.4 | 23.0 | 54.2 | 18.2 | 38.6 | 64.4 | 32.1 | 57.0 | 38.3 | 9.0 | 25.0 | 49.8 | 13.7 | 36.1 |
| JEDI-7B | 42.9 | 11.0 | 27.4 | 50.0 | 11.9 | 34.0 | 38.0 | 14.1 | 32.2 | 72.9 | 25.5 | 52.4 | 75.1 | 47.2 | 68.7 | 33.6 | 16.9 | 26.0 | 52.6 | 18.2 | 39.5 |

Table 4: Comparison of models on element grounding tasks in UI-Vision. The highlighted column presents the overall average performance across all categories.

| Model | Basic Overall | Functional Overall | Spatial Overall | Final Avg |
|---|---|---|---|---|
| Claude-3.7-Sonnet [1] | 9.48 | 7.73 | 7.60 | 8.27 |
| Qwen-2.5VL-7B | 1.24 | 0.79 | 0.51 | 0.85 |
| MiniCPM-V-8B [51] | 7.11 | 5.30 | 1.45 | 4.34 |
| ShowUI-2B | 8.07 | 7.67 | 2.07 | 5.94 |
| Aria-UI | 12.2 | 14.0 | 3.98 | 10.1 |
| UGround-v1-7B | 15.4 | 17.1 | 6.25 | 12.9 |
| OSAtlas-7B | 12.2 | 11.2 | 3.67 | 9.02 |
| Aguvis-7B | 17.8 | 18.3 | 5.06 | 13.7 |
| UI-TARS-7B | 20.1 | 24.3 | 8.37 | 17.6 |
| SeeClick | 9.42 | 4.68 | 2.07 | 5.39 |
| UI-TARS-72B | 31.4 | 30.5 | 14.7 | 25.5 |
| JEDI-3B | 22.3 | 25.2 | 9.35 | 18.7 |
| JEDI-7B | 32.3 | 30.5 | 12.8 | 24.8 |

Table 5: Performance comparison of models on OSWORLD-G across multiple capability dimensions. The highlighted column presents the overall average performance across all categories.

| Agent Model | Text Matching | Element Recognition | Layout Understanding | Fine-grained Manipulation | Refusal | Overall |
|---|---|---|---|---|---|---|
| OS-Atlas-7B | 44.1 | 29.4 | 35.2 | 16.8 | 7.4 | 27.7 |
| UGround-V1-7B | 51.3 | 40.3 | 43.5 | 24.8 | 0.0 | 36.4 |
| Aguvis-7B | 55.9 | 41.2 | 43.9 | 28.2 | 0.0 | 38.7 |
| UI-TARS-7B | 60.2 | 51.8 | 54.9 | 35.6 | 0.0 | 47.5 |
| Seed1.5-VL [38] | 73.9 | 66.7 | 69.6 | 47.0 | 18.5 | 62.9 |
| UI-TARS-72B | 69.4 | 60.6 | 62.9 | 45.6 | 0.0 | 57.1 |
| Gemini-2.5-Pro | 59.8 | 45.5 | 49.0 | 33.6 | 38.9 | 45.2 |
| Operator | 51.3 | 42.4 | 46.6 | 31.5 | 0.0 | 40.6 |
| Qwen2.5-VL-3B | 41.4 | 28.8 | 34.8 | 13.4 | 0.0 | 27.3 |
| Qwen2.5-VL-7B | 45.6 | 32.7 | 41.9 | 18.1 | 0.0 | 31.4 |
| Qwen2.5-VL-32B | 63.2 | 47.3 | 49.0 | 36.9 | 0.0 | 46.5 |
| JEDI-3B | 67.4 | 53.0 | 53.8 | 44.3 | 7.4 | 50.9 |
| JEDI-7B | 65.9 | 55.5 | 57.7 | 46.9 | 7.4 | 54.1 |

## 3.2 Agentic Ability

We hope that the data and benchmark we provide will ultimately serve as a critical signal in fostering the agentic capabilities required, rather than merely enhancing specific grounding abilities. We evaluate our approach on the computer use benchmarks in online environments, namely OSWorld [46, 47] and WindowsAgentArena [7]. We employ foundation models like GPT-4o or o3 as the planner model, which receives high-level instructions and, at each step, predicts the next low-level natural language instruction based on the current observation and action history. Our JEDI model then takes these low-level instructions and predicts the concrete actions to execute. To control for confounding variables, we do not introduce any specialized agent architecture or model scheduling [2].

The results demonstrate that, when using our model as the grounding component, a simple agent with foundation models that are not specialized in computer use tasks can achieve state-of-the-art performance, surpassing previous approaches that used 72B-scale models for grounding, and matching the performance of specialized models. Additionally, our agent system exhibits a similar trend to Operator, with performance improving as deployment scale increases. These findings suggest that, given the current reasoning capabilities of large language models, supplementing them with enhanced grounding ability—either through additional data or external systems—can be a starting point to build highly effective agentic systems.

Table 6: Success rate on the OSWorld and WindowsAgentArena benchmarks. JEDI with GPT-4o results are the average success rate of 4 runs with standard deviation. More detailed performance see A.6.

| Planner | Grounding | OS SR | WAA SR |
|---|---|---|---|
| | GPT-4o | 5.0 | 9.4 |
| | Kimi-VL [39] | 8.2 | 10.4 |
| | UI-TARS-72B | 22.7 | - |
| | o3 | 23.0 | - |
| | Operator [30] | 32.6 | - |
| | OpenCUA-32B [43] | 34.8 | - |
| | Claude 4 Sonnet | 43.9 | - |
| GPT-4o | Aguvis-72B | 17.0 | - |
| GPT-4o | JEDI-3B | 24.0 $_{\pm 1.05}$ | 33.03 $_{\pm 1.64}$ |
| GPT-4o | JEDI-7B | **27.0** $_{\pm 1.81}$ | **33.7** $_{\pm 0.82}$ |
| o3 | JEDI-7B | **51.0** | - |

## 4 Analysis

### 4.1 Effectiveness of Knowledge

GUI grounding also requires knowledge and even reasoning. We aim to investigate the performance of pure grounding when almost no additional knowledge is required. To this end, we assume that the instruction recipient possesses minimal prior experience with GUI interactions, and we re-annotate the entire benchmark to minimize the background knowledge needed to understand each instruction. This is achieved by relying on easily identifiable universal features such as color and shape. For example, the instruction "Open the filter function for search settings." is refined, based on the screenshot, to "Click the button that includes an icon of a funnel on the right of the 'search settings' bar." We conduct experiments on several models and present the performance comparison before and after instruction refinement in Figure 5. First, we observe that model performance generally improves after instruction refinement.

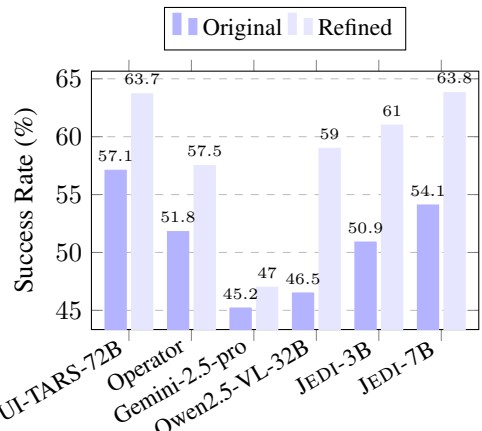

Figure 5: Success rates of various models on the OSWORLD-G benchmark with original and refined instructions.

This suggests that if we can supplement models with relevant interaction experience or provide more precise expressions—either manually or via upstream models—grounding performance can be enhanced. Second, after instruction refinement, our model achieves performance comparable to the largest state-of-the-art model, UI-TARS-72B. This indicates that, with appropriate data such as our JEDI dataset, smaller models are already sufficient in terms of pure grounding ability, and further advantages may lie in the supplementation of background knowledge.

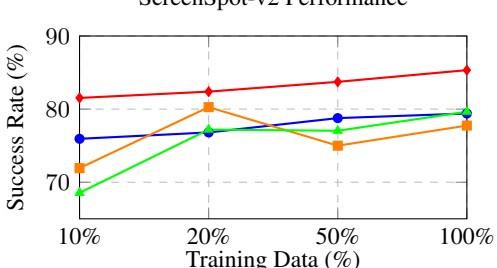
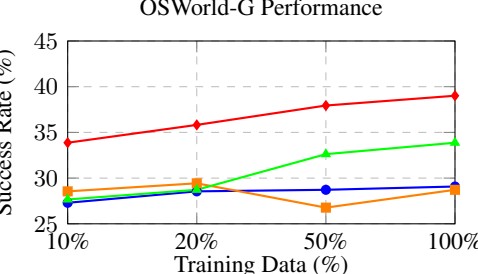

Figure 6: The effect of training data percentage on Qwen2.5-VL-3B model performance across different UI elements. Blue line: Icon; Orange line: Component; Green line: Layout; Red line: All. Left: ScreenSpot-v2 benchmark; Right: OSWORLD-G benchmark.

## 4.2 Performance as Data Scaling

We aim to investigate whether collecting data through our pipeline enables further performance improvements as the data scale increases. We sample data of icon, component, and layout at proportions of 10%, 20%, 50%, and 100%. For each data proportion, we train the models for the same number of steps, ensuring that all models are sufficiently trained to allow a fair comparison of final performance under equal computational resources. The results are shown in Figure 6. First, we observe that as the data scale increases, model performance continues to improve, with no sign of saturation. This suggests that further scaling up the data using our proposed approach can yield additional gains. Second, we note that scaling up a single data type (e.g., component) can lead to performance fluctuations. In contrast, scaling up mixed data types results in more stable improvements, indicating that combining data from multiple sources is beneficial.

## 4.3 Case Study

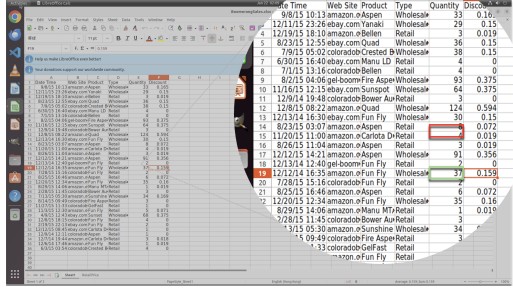

(a) Instruction: Click on the quantity of product in 12/12/14 16:35.

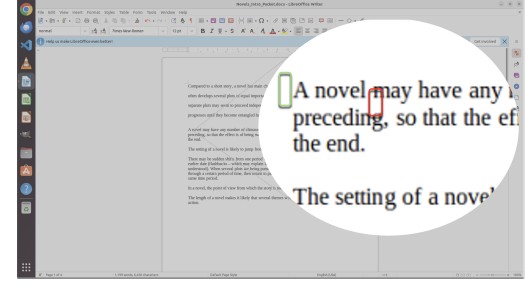

(b) Instruction: Place the cursor before the capital 'A' in the paragraph about novel climaxes.

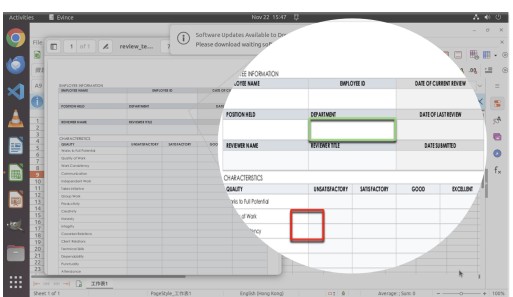

(c) Instruction: Fill up the middle space of the second blank line in the visualized information form.

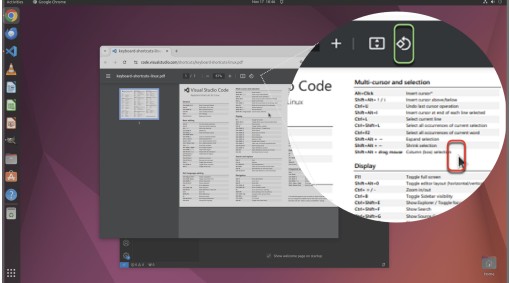

(d) Instruction: Button to rotate the pdf.

Figure 7: Qualitative comparison showing JEDI's enhanced fine-grained operation and GUI understanding compared to Qwen2.5-VL-7B-Instruct across four cases. The green square represents the click position of JEDI, while the red square indicates the click position of Qwen.

We conducted a detailed comparison of JEDI-7B and Qwen2.5-VL-7B-Instruct using OSWORLD-G. To illustrate the improvements of JEDI, we selected representative cases where their results differed,

as shown in Figure 7. In each subfigure, the green square represents the click position of JEDI, which is the correct grounding action, while the red square indicates the erroneous click position of Qwen. In these examples, JEDI showcases exceptional fine-grained operational capabilities and comprehension skills in locating and matching information. As illustrated in the subfigure 7a, JEDI successfully identifies the target cell without an explicit location (like "E19") by using information from both the timestamp and the table header. Similarly, by understanding the paragraph text and accurately identifying relative positions, JEDI effectively addresses the case presented in the 7b. Furthermore, as illustrated in the 7c, by learning from web page layouts, JEDI exhibits generalization to desktop environments, accurately locating the specified blank cell based on the positional description. Additionally, benefiting from training on extensive icon data, JEDI successfully associated the icon (a counter-clockwise arrow) with its corresponding function ("rotate"), as depicted in the 7d. Further analysis of additional examples can be found in Appendix A.5.1 and A.5.2.

## 5 Related Work

**Digital Agents** Multimodal agents can be broadly categorized into digital and physical agents [34, 35, 14]. Existing digital agent research focuses on establishing environments for mobile and web interaction [34, 24, 28, 40, 50, 56, 33, 53, 19, 13, 41], with subsequent works extending to real-world computer interaction scenarios [46, 7]. Recent advances include enhanced GUI understanding through visual encoding architectures [16, 6], reinforcement learning frameworks introduced to web/mobile operations [31, 5], agentic-frameworks [55, 15, 2, 49, 25] and joint visual-language modeling [45, 48, 32]. However, current methods face precision limitations on grounding due to homogeneous synthetic training data [10, 15, 49, 36], which overlook the systematic support for fine-grained component operations (e.g., slider adjustments, nested menu selections), finally limits the upper policy execution as well as further learning. Furthermore, the sources of data which could be beneficial for enhancing GUI interaction abilities are underexplored.

**GUI Grounding** GUI grounding remains a core challenge for digital agents executing actions in real world environment. Recent approaches have shifted from relying on textual information such as HTML/accessibility information to pure visual solutions [10, 55, 15, 26, 23, 52]. However, both existing training data and evaluation paradigms suffer from oversimplification—whether through screenshot-text pairings or manual annotations—failing to capture the complexity of natural language instructions and action execution, particularly in tasks requiring understanding of expressed intent rather than simple referencing, screen-level comprehension (such as identifying active windows), and fine-grained operations (like sliders and drag-and-drop), thus hindering meaningful assessment and advancement in these critical areas. We point out the problems by proposed benchmark and bridging these gaps through multiple aspects of synthetic data. The comparison with previous work is shown in Table 8.

## 6 Conclusion

We highlight overlooked GUI grounding challenges such as fine-grained manipulation and layout understanding, introducing OSWORLD-G with 564 annotated samples for evaluation. We set up multiple pipelines to construct a dataset containing 4 million examples to address these challenges. Our models trained on this dataset achieve competitive results on ScreenSpot-v2, ScreenSpot-Pro, and OSWORLD-G, while also boosting agent performance in OSWorld and WindowsAgentArena. These results demonstrate the effectiveness of addressing previously identified gaps in GUI grounding research.

## Acknowledgements

We thank Binyuan Hui, Weilu Xu, Dunjie Lu, Zhiyong Wu, Weiyun Wang, Hao Hu, Bowen Wang, Eric Xin Wang, Yuhao Yang, Junlei Zhang, Victor Zhong, Yujia Qin for their helpful feedback on discussion around this work.

# 7 Limitations

In this work, we mainly discuss the data synthesis methods while figuring out the essential factors. Screen capture data can be extracted from internet images and videos by neural networks, which can further expand the dataset. This approach can significantly expand the screenshot metadata, thus enlarging the grounding data. Due to resource restrictions, we leave this for further scaling through industrial efforts. Rejecting infeasible actions is crucial, as it helps prevent errors and mitigates the risks associated with incorrect instructions. Refusal modeling in GUI grounding remains a significant challenge, as models show limited improvement due to the inherent limitations in pretraining and the hallucination phenomenon in VLMs. While we find this problem has inherent complexity and challenges, this provides direction for future in-depth research and optimization. On the other hand, based on our enhanced grounding model, we can construct human-like traversers that interact in the digital world with or without specific purposes, similar to how humans navigate digital environments. This approach can further collect interaction data to improve grounding capabilities and even enhance model knowledge. We also leave these explorations for future work.

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

# NeurIPS Paper Checklist

1. **Claims**

   Question: Do the main claims made in the abstract and introduction accurately reflect the paper's contributions and scope?

   Answer: [Yes]

   Justification: The abstract and introduction accurately present the paper's contributions: (1) introducing OSWORLD-G, a comprehensive benchmark with 564 annotated samples across diverse GUI grounding tasks; (2) creating JEDI, a 4-million example dataset for GUI grounding; (3) training multi-scale models, demonstrating improved performance across grounding benchmarks and showing enhanced agentic capabilities on computer use tasks.

   Guidelines:

   - The answer NA means that the abstract and introduction do not include the claims made in the paper.
   - The abstract and/or introduction should clearly state the claims made, including the contributions made in the paper and important assumptions and limitations. A No or NA answer to this question will not be perceived well by the reviewers.
   - The claims made should match theoretical and experimental results, and reflect how much the results can be expected to generalize to other settings.
   - It is fine to include aspirational goals as motivation as long as it is clear that these goals are not attained by the paper.

2. **Limitations**

   Question: Does the paper discuss the limitations of the work performed by the authors?

   Answer: [Yes]

   Justification: In the section 7, we acknowledge the limitations of our work. Our dataset could be significantly expanded with industrial-scale resources to achieve superior GUI grounding results. Additionally, we propose a promising scaling approach by deploying our trained grounding model as an autonomous traverser in digital environments, which remains an avenue for future exploration.

   Guidelines:

   - The answer NA means that the paper has no limitation while the answer No means that the paper has limitations, but those are not discussed in the paper.
   - The authors are encouraged to create a separate "Limitations" section in their paper.
   - The paper should point out any strong assumptions and how robust the results are to violations of these assumptions (e.g., independence assumptions, noiseless settings, model well-specification, asymptotic approximations only holding locally). The authors should reflect on how these assumptions might be violated in practice and what the implications would be.
   - The authors should reflect on the scope of the claims made, e.g., if the approach was only tested on a few datasets or with a few runs. In general, empirical results often depend on implicit assumptions, which should be articulated.
   - The authors should reflect on the factors that influence the performance of the approach. For example, a facial recognition algorithm may perform poorly when image resolution is low or images are taken in low lighting. Or a speech-to-text system might not be used reliably to provide closed captions for online lectures because it fails to handle technical jargon.
   - The authors should discuss the computational efficiency of the proposed algorithms and how they scale with dataset size.
   - If applicable, the authors should discuss possible limitations of their approach to address problems of privacy and fairness.
   - While the authors might fear that complete honesty about limitations might be used by reviewers as grounds for rejection, a worse outcome might be that reviewers discover limitations that aren't acknowledged in the paper. The authors should use their best

judgment and recognize that individual actions in favor of transparency play an important role in developing norms that preserve the integrity of the community. Reviewers will be specifically instructed to not penalize honesty concerning limitations.

3. **Theory assumptions and proofs**

   Question: For each theoretical result, does the paper provide the full set of assumptions and a complete (and correct) proof?

   Answer: [NA]

   Justification: Our work primarily focuses on empirical contributions rather than theoretical results. We present a new benchmark OSWORLD-G, a dataset JEDI, and experimental results demonstrating performance improvements, but do not include formal theorems, lemmas, or mathematical proofs requiring validation.

   Guidelines:

   - The answer NA means that the paper does not include theoretical results.
   - All the theorems, formulas, and proofs in the paper should be numbered and cross-referenced.
   - All assumptions should be clearly stated or referenced in the statement of any theorems.
   - The proofs can either appear in the main paper or the supplemental material, but if they appear in the supplemental material, the authors are encouraged to provide a short proof sketch to provide intuition.
   - Inversely, any informal proof provided in the core of the paper should be complemented by formal proofs provided in appendix or supplemental material.
   - Theorems and Lemmas that the proof relies upon should be properly referenced.

4. **Experimental result reproducibility**

   Question: Does the paper fully disclose all the information needed to reproduce the main experimental results of the paper to the extent that it affects the main claims and/or conclusions of the paper (regardless of whether the code and data are provided or not)?

   Answer: [Yes]

   Justification: We have provided comprehensive information to ensure reproducibility of our experimental results. For our dataset JEDI, we detailed the data sources in Section 2.2 and thoroughly documented the data processing methodology and representative examples in Appendix A.3. Additionally, we have open-sourced our complete data construction pipeline at `https://osworld-grounding.github.io/`, enabling researchers to not only reproduce our dataset but also extend it further. To facilitate verification of our model performance claims, we have released our model checkpoints and evaluation scripts. This ensures that all benchmark results presented in our paper can be independently validated by the research community.

   Guidelines:

   - The answer NA means that the paper does not include experiments.
   - If the paper includes experiments, a No answer to this question will not be perceived well by the reviewers: Making the paper reproducible is important, regardless of whether the code and data are provided or not.
   - If the contribution is a dataset and/or model, the authors should describe the steps taken to make their results reproducible or verifiable.
   - Depending on the contribution, reproducibility can be accomplished in various ways. For example, if the contribution is a novel architecture, describing the architecture fully might suffice, or if the contribution is a specific model and empirical evaluation, it may be necessary to either make it possible for others to replicate the model with the same dataset, or provide access to the model. In general. releasing code and data is often one good way to accomplish this, but reproducibility can also be provided via detailed instructions for how to replicate the results, access to a hosted model (e.g., in the case of a large language model), releasing of a model checkpoint, or other means that are appropriate to the research performed.

- While NeurIPS does not require releasing code, the conference does require all submissions to provide some reasonable avenue for reproducibility, which may depend on the nature of the contribution. For example
  (a) If the contribution is primarily a new algorithm, the paper should make it clear how to reproduce that algorithm.
  (b) If the contribution is primarily a new model architecture, the paper should describe the architecture clearly and fully.
  (c) If the contribution is a new model (e.g., a large language model), then there should either be a way to access this model for reproducing the results or a way to reproduce the model (e.g., with an open-source dataset or instructions for how to construct the dataset).
  (d) We recognize that reproducibility may be tricky in some cases, in which case authors are welcome to describe the particular way they provide for reproducibility. In the case of closed-source models, it may be that access to the model is limited in some way (e.g., to registered users), but it should be possible for other researchers to have some path to reproducing or verifying the results.

5. **Open access to data and code**

   Question: Does the paper provide open access to the data and code, with sufficient instructions to faithfully reproduce the main experimental results, as described in supplemental material?

   Answer: [Yes]

   Justification: We have made our benchmark, dataset, and model publicly accessible via our project website at `https://osworld-grounding.github.io/`. The complete codebase for data processing has been open-sourced in our repository. In the appendix, we provide comprehensive instructions with detailed examples to facilitate reproduction of our experimental results. The open-sourced repository includes data preparation steps, and execution commands to ensure other researchers can validate and build upon our work.

   Guidelines:

   - The answer NA means that paper does not include experiments requiring code.
   - Please see the NeurIPS code and data submission guidelines (`https://nips.cc/public/guides/CodeSubmissionPolicy`) for more details.
   - While we encourage the release of code and data, we understand that this might not be possible, so "No" is an acceptable answer. Papers cannot be rejected simply for not including code, unless this is central to the contribution (e.g., for a new open-source benchmark).
   - The instructions should contain the exact command and environment needed to run to reproduce the results. See the NeurIPS code and data submission guidelines (`https://nips.cc/public/guides/CodeSubmissionPolicy`) for more details.
   - The authors should provide instructions on data access and preparation, including how to access the raw data, preprocessed data, intermediate data, and generated data, etc.
   - The authors should provide scripts to reproduce all experimental results for the new proposed method and baselines. If only a subset of experiments are reproducible, they should state which ones are omitted from the script and why.
   - At submission time, to preserve anonymity, the authors should release anonymized versions (if applicable).
   - Providing as much information as possible in supplemental material (appended to the paper) is recommended, but including URLs to data and code is permitted.

6. **Experimental setting/details**

   Question: Does the paper specify all the training and test details (e.g., data splits, hyperparameters, how they were chosen, type of optimizer, etc.) necessary to understand the results?

   Answer: [Yes]

   Justification: We have fully disclosed the training and test datasets used in our work in both the main text and appendix, provided citations for previous works, and released our

complete code. Users can fully understand and reproduce our experiments and results with this information.

Guidelines:

- The answer NA means that the paper does not include experiments.
- The experimental setting should be presented in the core of the paper to a level of detail that is necessary to appreciate the results and make sense of them.
- The full details can be provided either with the code, in appendix, or as supplemental material.

7. **Experiment statistical significance**

Question: Does the paper report error bars suitably and correctly defined or other appropriate information about the statistical significance of the experiments?

Answer: [Yes]

Justification: For relatively stable grounding evaluations such as ScreenSpot-v2, ScreenSpot-Pro, and OSWorld-G, we follow previous work and run experiments only once. For experiments with higher variability, we conduct four runs, assuming a normal distribution of results to estimate and report the mean and standard deviation.

Guidelines:

- The answer NA means that the paper does not include experiments.
- The authors should answer "Yes" if the results are accompanied by error bars, confidence intervals, or statistical significance tests, at least for the experiments that support the main claims of the paper.
- The factors of variability that the error bars are capturing should be clearly stated (for example, train/test split, initialization, random drawing of some parameter, or overall run with given experimental conditions).
- The method for calculating the error bars should be explained (closed form formula, call to a library function, bootstrap, etc.)
- The assumptions made should be given (e.g., Normally distributed errors).
- It should be clear whether the error bar is the standard deviation or the standard error of the mean.
- It is OK to report 1-sigma error bars, but one should state it. The authors should preferably report a 2-sigma error bar than state that they have a 96% CI, if the hypothesis of Normality of errors is not verified.
- For asymmetric distributions, the authors should be careful not to show in tables or figures symmetric error bars that would yield results that are out of range (e.g. negative error rates).
- If error bars are reported in tables or plots, The authors should explain in the text how they were calculated and reference the corresponding figures or tables in the text.

8. **Experiments compute resources**

Question: For each experiment, does the paper provide sufficient information on the computer resources (type of compute workers, memory, time of execution) needed to reproduce the experiments?

Answer: [Yes]

Justification: We have made the claim of in Section 3 about CPU, GPU, and memory.

Guidelines:

- The answer NA means that the paper does not include experiments.
- The paper should indicate the type of compute workers CPU or GPU, internal cluster, or cloud provider, including relevant memory and storage.
- The paper should provide the amount of compute required for each of the individual experimental runs as well as estimate the total compute.
- The paper should disclose whether the full research project required more compute than the experiments reported in the paper (e.g., preliminary or failed experiments that didn't make it into the paper).

9. **Code of ethics**

   Question: Does the research conducted in the paper conform, in every respect, with the NeurIPS Code of Ethics https://neurips.cc/public/EthicsGuidelines?

   Answer: [Yes]

   Justification: Our research fully complies with the NeurIPS Code of Ethics. We have carefully reviewed all ethical guidelines and ensured our work meets these standards.

   Guidelines:

   - The answer NA means that the authors have not reviewed the NeurIPS Code of Ethics.
   - If the authors answer No, they should explain the special circumstances that require a deviation from the Code of Ethics.
   - The authors should make sure to preserve anonymity (e.g., if there is a special consideration due to laws or regulations in their jurisdiction).

10. **Broader impacts**

    Question: Does the paper discuss both potential positive societal impacts and negative societal impacts of the work performed?

    Answer: [NA]

    Justification: Our work focuses on improving the grounding capabilities of visual language models through a new dataset, which is primarily a technical advancement in the field of computer use agent research. While enhanced grounding capabilities could eventually contribute to more accurate and reliable agent systems, our current research is still at a foundational stage and several steps removed from direct deployment in real-world applications. The models we develop are research prototypes with limited capabilities and scale compared to production systems. They are not designed for or capable of tasks that could lead to immediate societal concerns such as generating misleading content, surveillance, or automated decision-making that might affect vulnerable populations.

    Guidelines:

    - The answer NA means that there is no societal impact of the work performed.
    - If the authors answer NA or No, they should explain why their work has no societal impact or why the paper does not address societal impact.
    - Examples of negative societal impacts include potential malicious or unintended uses (e.g., disinformation, generating fake profiles, surveillance), fairness considerations (e.g., deployment of technologies that could make decisions that unfairly impact specific groups), privacy considerations, and security considerations.
    - The conference expects that many papers will be foundational research and not tied to particular applications, let alone deployments. However, if there is a direct path to any negative applications, the authors should point it out. For example, it is legitimate to point out that an improvement in the quality of generative models could be used to generate deepfakes for disinformation. On the other hand, it is not needed to point out that a generic algorithm for optimizing neural networks could enable people to train models that generate Deepfakes faster.
    - The authors should consider possible harms that could arise when the technology is being used as intended and functioning correctly, harms that could arise when the technology is being used as intended but gives incorrect results, and harms following from (intentional or unintentional) misuse of the technology.
    - If there are negative societal impacts, the authors could also discuss possible mitigation strategies (e.g., gated release of models, providing defenses in addition to attacks, mechanisms for monitoring misuse, mechanisms to monitor how a system learns from feedback over time, improving the efficiency and accessibility of ML).

11. **Safeguards**

    Question: Does the paper describe safeguards that have been put in place for responsible release of data or models that have a high risk for misuse (e.g., pretrained language models, image generators, or scraped datasets)?

    Answer: [NA]

Justification: Computer grounding task and data we discuss in this paper itself don't pose safety risks.

Guidelines:

- The answer NA means that the paper poses no such risks.
- Released models that have a high risk for misuse or dual-use should be released with necessary safeguards to allow for controlled use of the model, for example by requiring that users adhere to usage guidelines or restrictions to access the model or implementing safety filters.
- Datasets that have been scraped from the Internet could pose safety risks. The authors should describe how they avoided releasing unsafe images.
- We recognize that providing effective safeguards is challenging, and many papers do not require this, but we encourage authors to take this into account and make a best faith effort.

12. **Licenses for existing assets**

Question: Are the creators or original owners of assets (e.g., code, data, models), used in the paper, properly credited and are the license and terms of use explicitly mentioned and properly respected?

Answer: [Yes]

Justification: We have provided explanations for all sources used, including URLs, repository addresses, and material locations. For pre-existing datasets that we directly used, we have cited their corresponding work.

Guidelines:

- The answer NA means that the paper does not use existing assets.
- The authors should cite the original paper that produced the code package or dataset.
- The authors should state which version of the asset is used and, if possible, include a URL.
- The name of the license (e.g., CC-BY 4.0) should be included for each asset.
- For scraped data from a particular source (e.g., website), the copyright and terms of service of that source should be provided.
- If assets are released, the license, copyright information, and terms of use in the package should be provided. For popular datasets, `paperswithcode.com/datasets` has curated licenses for some datasets. Their licensing guide can help determine the license of a dataset.
- For existing datasets that are re-packaged, both the original license and the license of the derived asset (if it has changed) should be provided.
- If this information is not available online, the authors are encouraged to reach out to the asset's creators.

13. **New assets**

Question: Are new assets introduced in the paper well documented and is the documentation provided alongside the assets?

Answer: [Yes]

Justification: We provide documentation for all new assets introduced in our paper, including our benchmark, dataset, and model. All assets are publicly accessible through our project website at `https://osworld-grounding.github.io/`. The documentation thoroughly covers implementation details, usage instructions, technical specifications, and example applications to facilitate adoption by other researchers. Each asset is accompanied by documentation in its corresponding repository. We are committed to maintaining long-term support for these resources to benefit the broader research community.

Guidelines:

- The answer NA means that the paper does not release new assets.
- Researchers should communicate the details of the dataset/code/model as part of their submissions via structured templates. This includes details about training, license, limitations, etc.

- The paper should discuss whether and how consent was obtained from people whose asset is used.
- At submission time, remember to anonymize your assets (if applicable). You can either create an anonymized URL or include an anonymized zip file.

14. **Crowdsourcing and research with human subjects**

Question: For crowdsourcing experiments and research with human subjects, does the paper include the full text of instructions given to participants and screenshots, if applicable, as well as details about compensation (if any)?

Answer: [NA]

Justification: Our work does not involve any crowdsourcing or human subjects. Therefore, no participant instructions, screenshots of interfaces, or compensation details are applicable to our study.

Guidelines:

- The answer NA means that the paper does not involve crowdsourcing nor research with human subjects.
- Including this information in the supplemental material is fine, but if the main contribution of the paper involves human subjects, then as much detail as possible should be included in the main paper.
- According to the NeurIPS Code of Ethics, workers involved in data collection, curation, or other labor should be paid at least the minimum wage in the country of the data collector.

15. **Institutional review board (IRB) approvals or equivalent for research with human subjects**

Question: Does the paper describe potential risks incurred by study participants, whether such risks were disclosed to the subjects, and whether Institutional Review Board (IRB) approvals (or an equivalent approval/review based on the requirements of your country or institution) were obtained?

Answer: [NA]

Justification: This research does not involve any human subjects and crowdsourcing in any capacity. All data processing and experiments were conducted without human involvement beyond the research team. Hence, this term does not apply to this work.

Guidelines:

- The answer NA means that the paper does not involve crowdsourcing nor research with human subjects.
- Depending on the country in which research is conducted, IRB approval (or equivalent) may be required for any human subjects research. If you obtained IRB approval, you should clearly state this in the paper.
- We recognize that the procedures for this may vary significantly between institutions and locations, and we expect authors to adhere to the NeurIPS Code of Ethics and the guidelines for their institution.
- For initial submissions, do not include any information that would break anonymity (if applicable), such as the institution conducting the review.

16. **Declaration of LLM usage**

Question: Does the paper describe the usage of LLMs if it is an important, original, or non-standard component of the core methods in this research? Note that if the LLM is used only for writing, editing, or formatting purposes and does not impact the core methodology, scientific rigorousness, or originality of the research, declaration is not required.

Answer: [Yes]

Justification: We incorporated LLMs as an integral component of our data synthesis pipeline, as detailed in section 2.2.4. Specifically, we leveraged LLMs to generate annotations for our data points. For transparency and reproducibility, we have provided the complete implementation code and all prompts used in this process at `https://osworld-grounding.github.io/`.

Guidelines:

- The answer NA means that the core method development in this research does not involve LLMs as any important, original, or non-standard components.
- Please refer to our LLM policy (`https://neurips.cc/Conferences/2025/LLM`) for what should or should not be described.

# A  Appendix

## A.1  OSWORLD-G Statistics

### A.1.1  Data Types

We categorize the examples into five categories that requires different grounding capabilities. And the classification can be refer to their corresponding element types in the Table 7.

Table 7: Full table of distribution of examples in the OSWORLD-G benchmark categorized by GUI grounding capabilities and their corresponding interface element types.

| Capabilities | Element Types | # of Examples |
|---|---|---|
| Text Matching | Label | 268 |
| Element Recognition | Icon, Image, Button | 337 |
| Layout Understanding | Tab, Banner/Notification, Accordion/Collapsible Panel, Pagination Control, Toolbar, Menu Bar, Dropdown Menu, List, Grid, Tree View, Dialog/Modal, Panel/-Container, Sidebar, Drawer | 252 |
| Fine-grained Manipulation | Slider, Stepper, Divider, Toggle/Switch, Accordion/Collapsible Panel, Checkbox, Radio Button, Color Picker, Date Picker, Table, Text Field/Input Box, Search Bar, Text Filed, Input Box | 154 |
| Refusal | – | 54 |

### A.1.2  Comparison with Previous Work

We show the comparison between OSWORLD-G and previous work in Table 8.

Table 8: Comparison between OSWORLD-G and previous benchmarks.

| Benchmarks | Platforms | # of Examples | # of Annotated UI-Types | Instruction Annotation | | Fine-grained Actions | Refusal Cases |
|---|---|---|---|---|---|---|---|
| | | | | Visual | Functional | | |
| ScreenSpot-v2 | Mobile, Desktop, Web | 1272 | 2 (Icon, Text) | ✗ | ✓ | ✗ | ✗ |
| ScreenSpot Pro | Desktop | 1581 | 2 (Icon, Text) | ✗ | ✓ | ✗ | ✗ |
| OmniAct | Desktop, Web | 9802 | 3 (Icon, Text, Color) | ✗ | ✓ | ✗ | ✗ |
| **OSWORLD-G** | **Desktop** | **564** | **32** | ✓ | ✓ | ✓ | ✓ |

### A.1.3  Data Examples

We show examples of text matching type and element recognition type in Figure 2 (layout understanding), 3 (fine-grained manipulation), 8 (text matching, element recognition and refusal instruction).

**Layout Understanding**   Layout understanding tasks require models to comprehend the hierarchical structure of interface elements. In the example shown in Figure 2, closing the top notification bar requires recognizing that such bars typically appear at the top region of the editing area in Libreoffice Calc.

**Fine-grained Manipulation**   Fine-grained manipulation tasks demand high-precision actions within small or tightly packed screen regions. In the example in Figure 3, selecting the position between the word "person" and the number "1" requires the model to operate at a character-level granularity.

**Text Matching**   Text matching tasks involve grounding actions based on explicit textual cues in the instruction. As shown in Figure 8a, choosing "As Attachment" requires the model to locate and match this phrase within the screenshot.

**Element Recognition**   Element recognition tasks require identifying visual patterns such as icons or images. In the example in Figure 8b, clicking on the ellipse icon involves recognizing the ellipse shape visually within the interface.

**Refusal Instruction** Refusal instruction tasks assess whether the model can recognize when an action is infeasible. In the example in Figure 8c, the instruction refers to "Cindy Williams," who is not visible on the screen. Therefore, clicking on her email address is not possible, and the model is expected to refrain from taking action.

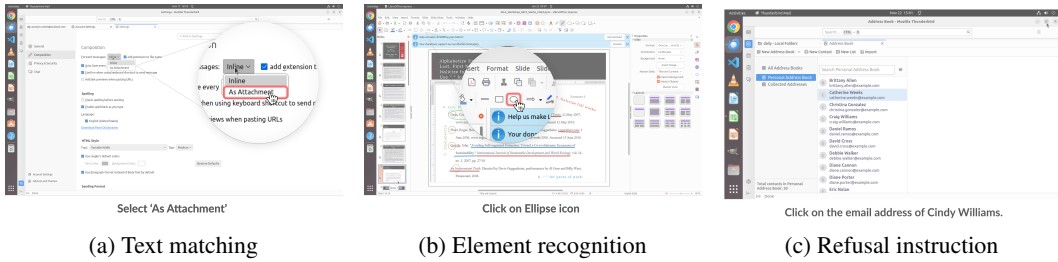

| (a) Text matching | (b) Element recognition | (c) Refusal instruction |

Figure 8: Examples in OSWORLD-G that require text matching and element recognition abilities.

### A.1.4 Annotation Details

The annotation process for OSWORLD-G comprised the following systematic steps:

1. **Failure Case Collection:** We systematically gathered grounding failure cases from state-of-the-art model trajectories, categorizing each failure according to its primary grounding capability requirements.

2. **Expert Annotation:** Annotators with extensive experience in various software applications performed initial precise annotations. Using the collected failure cases as guidance, they crafted descriptive low-level instructions that were designed to be unambiguous and map uniquely to specific screen actions.

3. **Bounding Box Annotation:** For each instruction-screenshot pair, corresponding bounding boxes were carefully annotated to indicate the precise regions of interest.

4. **Quality Verification:** We conducted multi-round verification procedures, leveraging predictions from strong models to resolve cases with inconsistent annotation results.

All examples in OSWORLD-G were annotated following this rigorous process to ensure high-quality and consistent annotations.

### A.2 JEDI Statistics

### A.2.1 Overview

Table 9 provides an overview of the statistics for JEDI. To enhance the quality of our dataset, we made several improvements upon AGUVIS, where we name it AGUVIS++. First, we manually filter out low-quality annotations and samples unrelated to computer use scenarios. We then augment the dataset by incorporating OS-Atlas data. For synthetic data sources such as SeeClick and OS-Atlas, which inherently contain rendering artifacts and alignment issues, we employ UI-TARS [4] model for quality control—comparing predicted outputs against ground truth values to ensure deviations remained within acceptable thresholds. In-house data is annotated by human workers. We ask them to use computers while recording timestamps of their actions and capturing screenshots from their screens as observations. These are later used with models like GPT to construct input instructions.

Table 9: Data statistics of our dataset. The '# Line' indicates the compression of multiple query-answer pairs to improve training efficiency.

| Data Source | # Image | # Line | # Turn | Sampling |
|---|---|---|---|---|
| **JEDI** | | | | |

---

[4]`https://huggingface.co/ByteDance-Seed/UI-TARS-72B-DPO`

| Data Source | # Image | # Line | # Turn | Sampling |
|---|---|---|---|---|
| Icon Captioning | 267,102 | 403,584 | 251,837 | All |
| Icon Grounding | 202,399 | 202,419 | 202,419 | All |
| Component Manipulation (Rule-based) | 29,303 | 40,653 | 40,653 | All |
| Component Manipulation (Generated) | 60,085 | 529,749 | 1,192,687 | All |
| Layout Captioning (App) | 5,117 | 17,721 | 366,774 | All |
| Layout Grounding (App) | 5,117 | 25,133 | 916,539 | All |
| Layout Captioning (OS) | 2,901 | 14,351 | 258,334 | All |
| Layout Grounding (OS) | 2,901 | 26,190 | 774,546 | All |
| | | | | |
| **JEDI Refusal** | | | | |
| Refusal Data (Various Sources) | 165,235 | 2,666,124 | 2,666,124 | Random:5% |
| | | | | |
| **AGUVIS++ [48]** | | | | |
| SeeClick [10] | 66,426 | 69,634 | 525,442 | All |
| WebUI [44] | 57,389 | 57,389 | 143,187 | All |
| GUIEnv [8] | 70,394 | 327,972 | 327,972 | All |
| GUIAct (web single) [8] | 17,545 | 17,572 | 17,572 | All |
| Widget Captioning [22] | 14,409 | 101,426 | 101,426 | All |
| RicoSCA [21] | 18,146 | 173,212 | 173,212 | All |
| UI RefExp [4] | 4,646 | 15,624 | 15,624 | All |
| RICO Icon [11] | 16,133 | 16,133 | 32,091 | All |
| OmniACT [18] | 6,720 | 6,720 | 6,720 | All |
| DocVQA Grounding [27, 48] | 9,756 | 34,060 | 34,060 | All |
| MM-Mind2Web [12] | 7,351 | 7,351 | 7,351 | All |
| GUIAct (web multi) [8] | 13,262 | 65,740 | 65,740 | All |
| AitZ [54] | 12,002 | 11,914 | 11,914 | All |
| AndroidControl [20] | 54,678 | 54,678 | 54,678 | All |
| Guide [37] | 12,422 | 12,422 | 12,422 | All |
| OS-Atlas [45] | 303,472 | 303,472 | 303,472 | All |
| | | | | |
| **In-house Data** | | | | |
| Additional In-house Annotated and Augmented Data | 1,392,009 | 1,392,016 | 1,486,289 | All |

### A.2.2 Icon Statistics

The Source Statistics of icon data in JEDI are detailed in Table 10.

Table 10: Data sources for icon collection in the JEDI dataset. Due to the diverse and scattered nature of these sources, they are presented here collectively rather than being broken down in the overall dataset overview.

| Data | Source | Link |
|------|--------|------|
| Ubuntu 2204 | Crawl | `https://github.com/ubuntu/yaru/tree/master/icons` |
| Snap Store | Crawl | `https://snapcraft.io/store` |
| Windows XP | Reverse engineering | - |
| Windows Vista | Reverse engineering | - |
| Windows 7 | Reverse engineering | - |
| Windows 10 | Reverse engineering | - |
|  | Crawl | `https://learn.microsoft.com/en-us/windows/apps/design/style/segoe-ui-symbol-font` |
| Windows 11 | Reverse engineering | - |
|  | Crawl | `https://github.com/microsoft/fluentui-system-icons/tree/main/assets` |
| Miscrosoft App Store | Crawl | `https://apps.microsoft.com/apps?hl=en-gb&gl=US` |
| macOS Ventura | Reverse engineering | - |
| macOS Sonoma | Reverse engineering | - |
| macOS Sequoia | Reverse engineering | - |
| macOS icon Collection | Crawl | `https://macosicons.com/` |
| Apple App store | Crawl | |
| iOS App store | Crawl | |
| Calculator | Crawl | `https://github.com/microsoft/calculator/tree/main/src/Calculator/Assets` |
| Audacity | Crawl | `https://github.com/audacity/audacity/tree/master/au3/libraries/lib-theme-resources` |
| Google | Crawl | `https://fonts.google.com/icons` |
| VSCode | Crawl | `https://github.com/microsoft/vscode-icons` |
| LibreOffice | Crawl | `https://github.com/LibreOffice/core/tree/master/icon-themes` |
| GitHub | Crawl | `https://github.com/primer/octicons/tree/main/icons` |
| GIMP | Crawl | `https://github.com/GNOME/gimp` |
| VLC | Crawl | `https://github.com/videolan/vlc` |
| PhotoShop | Reverse engineering | - |

### A.2.3 Component Statistics

The following Table 11 provides a detailed list of the component libraries we use, along with the contribution of each component to the JEDI dataset.

Table 11: Statistics of Material UI Components

| Component Type | Conversations | Images |
|---|---|---|
| material (Total) | 385,493 | 31,309 |
| no-ssr | 321 | 24 |
| box | 560 | 47 |
| textarea-autosize | 445 | 37 |
| click-away-listener | 764 | 45 |
| links | 886 | 35 |
| floating-action-button | 689 | 51 |
| bottom-navigation | 6,709 | 535 |
| popper | 3,258 | 169 |
| modal | 1,699 | 71 |
| speed-dial | 3,974 | 630 |
| accordion | 1,840 | 82 |
| rating | 9,409 | 869 |
| use-media-query | 7,285 | 113 |
| dividers | 2,318 | 83 |
| skeleton | 2,103 | 85 |
| alert | 7,290 | 1,378 |
| typography | 511 | 38 |
| button-group | 2,474 | 102 |
| radio-buttons | 3,020 | 115 |
| steppers | 5,252 | 869 |
| container | 625 | 37 |
| badges | 2,991 | 108 |
| cards | 3,881 | 160 |
| progress | 4,448 | 231 |
| icons | 4,663 | 173 |
| image-list | 2,389 | 96 |
| popover | 658 | 41 |
| toggle-button | 7,919 | 1,183 |
| checkboxes | 8,447 | 1,148 |
| buttons | 4,545 | 206 |
| selects | 5,122 | 194 |
| backdrop | 214 | 16 |
| menus | 15,498 | 1,839 |
| transitions | 1,794 | 92 |
| masonry | 7,932 | 106 |
| text-fields | 3,964 | 285 |
| portal | 134 | 26 |
| dialogs | 9,478 | 1,445 |
| breadcrumbs | 3,693 | 110 |
| switches | 7,050 | 1,050 |
| stack | 2,371 | 82 |
| paper | 5,993 | 97 |
| tooltips | 5,648 | 266 |
| timeline | 7,893 | 219 |
| chips | 13,440 | 1,951 |
| transfer-list | 2,100 | 295 |
| tabs | 52,425 | 2,917 |
| snackbars | 6,891 | 1,477 |
| app-bar | 17,474 | 2,096 |
| table | 11,536 | 839 |
| lists | 17,377 | 2,094 |
| drawers | 15,942 | 1,846 |
| grid-legacy | 5,979 | 149 |
| pagination | 12,497 | 197 |
| slider | 27,843 | 2,210 |

Table 11 – continued from previous page

| Component Type | Conversations | Images |
|---|---|---|
| autocomplete | 10,356 | 322 |
| avatars | 5,634 | 154 |
| grid | 7,842 | 174 |
| mantine (Total) | 27,814 | 762 |
| InputValidation | 577 | 14 |
| DndTable | 118 | 6 |
| ButtonProgress | 45 | 6 |
| ActionToggle | 852 | 17 |
| HeaderMenu | 62 | 3 |
| AutocompleteLoading | 17 | 3 |
| AuthenticationImage | 34 | 3 |
| NavbarMinimalColored | 97 | 4 |
| PasswordStrength | 98 | 4 |
| HeaderTabs | 75 | 3 |
| NavbarLinksGroup | 5,322 | 56 |
| ArticleCard | 185 | 7 |
| HeroBullets | 111 | 4 |
| InputWithButton | 73 | 4 |
| FeaturesGrid | 121 | 4 |
| CardsCarousel | 79 | 4 |
| UsersRolesTable | 94 | 4 |
| ContainedInputs | 89 | 6 |
| FeaturesImages | 140 | 5 |
| NavbarMinimal | 64 | 3 |
| HeroImageBackground | 102 | 7 |
| TableSelection | 245 | 6 |
| CardGradient | 718 | 15 |
| HeroContentLeft | 95 | 6 |
| ButtonCopy | 52 | 5 |
| FeaturesCards | 128 | 4 |
| TableReviews | 140 | 3 |
| UserCardImage | 202 | 7 |
| StatsGrid | 214 | 7 |
| NavbarSearch | 141 | 5 |
| ArticlesCardsGrid | 144 | 6 |
| ProgressCard | 60 | 4 |
| NotFoundImage | 22 | 3 |
| ProgressCardColored | 1,852 | 28 |
| UserInfoAction | 138 | 8 |
| ImageCheckboxes | 263 | 9 |
| StatsCard | 111 | 5 |
| ImageActionBanner | 60 | 4 |
| HeaderSearch | 81 | 4 |
| CustomSwitch | 32 | 3 |
| FaqSimple | 97 | 4 |
| HeaderSimple | 63 | 4 |
| ForgotPasswordInput | 37 | 4 |
| DndList | 167 | 6 |
| ArticleCardFooter | 118 | 4 |
| CarouselCard | 94 | 4 |
| CommentSimple | 107 | 5 |
| StatsGroup | 78 | 3 |
| StatsControls | 124 | 5 |
| DoubleHeader | 70 | 5 |
| TableOfContentsFloating | 74 | 4 |
| FaqWithImage | 71 | 4 |
| CardWithStats | 250 | 8 |
| EmailBanner | 146 | 6 |
| LeadGrid | 145 | 7 |
| Subgrid | 73 | 4 |
| SliderIcon | 132 | 3 |
| UserButton | 72 | 4 |

| Component Type | Conversations | Images |
|---|---|---|
| NavbarSegmented | 54 | 4 |
| NavbarSimple | 103 | 4 |
| NothingFoundBackground | 116 | 11 |
| FeaturesTitle | 181 | 5 |
| HeroImageRight | 132 | 5 |
| UsersStack | 240 | 4 |
| FooterLinks | 208 | 5 |
| NotFoundTitle | 19 | 3 |
| ContactUs | 398 | 12 |
| ButtonMenu | 261 | 17 |
| GradientSegmentedControl | 102 | 5 |
| ArticleCardVertical | 99 | 7 |
| NavbarSimpleColored | 99 | 4 |
| CurrencyInput | 43 | 5 |
| SliderLabel | 196 | 3 |
| ArticleCardImage | 48 | 4 |
| FeaturesAsymmetrical | 76 | 3 |
| FooterSocial | 157 | 8 |
| HeaderMegaMenu | 91 | 4 |
| StatsRingCard | 74 | 5 |
| TableSort | 108 | 4 |
| AuthenticationTitle | 77 | 3 |
| TableScrollArea | 92 | 3 |
| CommentHtml | 147 | 7 |
| AuthenticationForm | 195 | 15 |
| GetInTouch | 305 | 8 |
| HeroTitle | 57 | 3 |
| DropzoneButton | 24 | 4 |
| ServerOverload | 124 | 8 |
| SliderMarks | 32 | 4 |
| GetInTouchSimple | 109 | 4 |
| SliderWhite | 64 | 4 |
| StatsRing | 138 | 4 |
| StatsSegments | 149 | 5 |
| HeroText | 117 | 8 |
| FloatingLabelInput | 19 | 4 |
| CookiesBanner | 48 | 4 |
| TaskCard | 1,383 | 19 |
| ForgotPassword | 49 | 3 |
| InputTooltip | 40 | 4 |
| TableOfContents | 119 | 4 |
| CheckboxCard | 15 | 4 |
| ServerError | 35 | 5 |
| FaqWithBg | 74 | 4 |
| SplitButton | 76 | 3 |
| LanguagePicker | 100 | 5 |
| BadgeCard | 38 | 3 |
| SwitchesCard | 1,339 | 16 |
| FeaturesCard | 91 | 4 |
| ImageCard | 115 | 8 |
| DoubleNavbar | 110 | 4 |
| FaqWithHeader | 151 | 8 |
| UserMenu | 136 | 4 |
| UserInfoIcons | 103 | 4 |
| NavbarNested | 167 | 7 |
| SliderInput | 113 | 4 |
| StatsGridIcons | 186 | 5 |
| FooterSimple | 137 | 4 |
| UsersTable | 2,927 | 23 |
| SocialButtons | 325 | 10 |
| SliderHover | 84 | 4 |
| FooterCentered | 58 | 2 |

| Component Type | Conversations | Images |
|---|---|---|
| DndListHandle | 76 | 2 |
| ActionsGrid | 122 | 4 |
| GridAsymmetrical | 172 | 2 |
| ant-design (Total) | 473,723 | 16,837 |
| switch | 1,484 | 94 |
| watermark | 1,849 | 83 |
| skeleton | 1,913 | 99 |
| divider | 2,038 | 98 |
| tooltip | 3,525 | 194 |
| rate | 5,492 | 135 |
| auto-complete | 4,359 | 203 |
| tour | 2,174 | 114 |
| checkbox | 6,531 | 255 |
| splitter | 4,878 | 254 |
| time-picker | 6,642 | 276 |
| collapse | 5,191 | 225 |
| qr-code | 2,711 | 160 |
| menu | 6,182 | 215 |
| segmented | 6,088 | 238 |
| flex | 3,386 | 113 |
| notification | 6,090 | 208 |
| alert | 6,124 | 199 |
| list | 7,470 | 190 |
| button | 8,060 | 329 |
| timeline | 5,727 | 163 |
| carousel | 1,997 | 120 |
| modal | 9,508 | 379 |
| drawer | 7,818 | 275 |
| steps | 9,873 | 338 |
| affix | 1,105 | 66 |
| card | 7,331 | 345 |
| progress | 9,485 | 274 |
| mentions | 3,829 | 190 |
| typography | 3,405 | 203 |
| tree-select | 5,515 | 248 |
| descriptions | 6,732 | 236 |
| message | 3,626 | 156 |
| transfer | 7,308 | 193 |
| popover | 3,009 | 163 |
| empty | 1,105 | 89 |
| badge | 8,339 | 292 |
| radio | 6,888 | 239 |
| spin | 1,825 | 126 |
| float-button | 3,457 | 215 |
| image | 3,765 | 217 |
| cascader | 6,992 | 372 |
| popconfirm | 2,156 | 153 |
| calendar | 10,389 | 141 |
| form | 10,818 | 679 |
| config-provider | 3,167 | 143 |
| app | 663 | 35 |
| statistic | 2,345 | 96 |
| back-top | 454 | 29 |
| breadcrumb | 3,964 | 130 |
| input-number | 5,420 | 286 |
| space | 6,426 | 240 |
| avatar | 5,120 | 150 |
| icon | 2,898 | 135 |
| slider | 9,588 | 231 |
| tabs | 36,465 | 609 |
| upload | 3,270 | 373 |
| anchor | 4,524 | 165 |

| Component Type | Conversations | Images |
| --- | --- | --- |
| tag | 5,118 | 214 |
| tree | 16,354 | 433 |
| input | 7,012 | 478 |
| select | 15,055 | 550 |
| color-picker | 6,342 | 410 |
| pagination | 9,673 | 221 |
| layout | 6,604 | 188 |
| dropdown | 9,035 | 301 |
| grid | 13,957 | 241 |
| date-picker | 20,445 | 739 |
| table | 44,891 | 844 |
| result | 744 | 42 |
| chakra (Total) | 330,074 | 11,784 |
| mark | 288 | 24 |
| loader | 488 | 35 |
| bleed | 420 | 35 |
| aspect | 1,045 | 74 |
| center | 964 | 60 |
| skeleton | 1,097 | 80 |
| fieldset | 834 | 45 |
| locale | 424 | 26 |
| list | 1,758 | 64 |
| theme | 787 | 41 |
| separator | 1,751 | 83 |
| editable | 1,684 | 111 |
| code | 1,630 | 72 |
| float | 1,669 | 83 |
| visually | 1,725 | 50 |
| box | 2,337 | 118 |
| segmented | 2,405 | 110 |
| spinner | 1,921 | 110 |
| simple | 2,256 | 58 |
| link | 1,910 | 92 |
| for | 1,057 | 33 |
| hover | 1,231 | 63 |
| blockquote | 1,770 | 109 |
| flex | 1,653 | 74 |
| alert | 2,702 | 143 |
| accordion | 3,855 | 160 |
| steps | 3,763 | 168 |
| timeline | 2,784 | 85 |
| stat | 3,082 | 123 |
| switch | 3,285 | 173 |
| radiomark | 680 | 28 |
| text | 1,250 | 69 |
| highlight | 2,075 | 97 |
| drawer | 13,309 | 272 |
| menu | 5,395 | 193 |
| tooltip | 11,407 | 448 |
| toggle | 846 | 56 |
| collapsible | 325 | 24 |
| button | 5,358 | 252 |
| container | 672 | 27 |
| checkmark | 624 | 29 |
| badge | 2,590 | 91 |
| close | 446 | 34 |
| show | 636 | 47 |
| field | 1,850 | 124 |
| card | 3,155 | 108 |
| empty | 987 | 70 |
| textarea | 2,550 | 183 |
| action | 673 | 30 |

| Component Type | Conversations | Images |
|---|---|---|
| image | 1,173 | 83 |
| password | 2,104 | 68 |
| toaster | 9,295 | 231 |
| rating | 9,917 | 195 |
| pin | 2,384 | 147 |
| qr | 2,394 | 141 |
| status | 1,879 | 60 |
| group | 1,453 | 66 |
| popover | 5,154 | 200 |
| file | 1,644 | 168 |
| prose | 2,275 | 90 |
| tabs | 23,088 | 397 |
| native | 1,429 | 67 |
| em | 215 | 23 |
| kbd | 8,119 | 197 |
| portal | 218 | 26 |
| dialog | 5,663 | 220 |
| select | 6,850 | 232 |
| tag | 4,416 | 150 |
| clipboard | 1,444 | 93 |
| grid | 1,784 | 45 |
| table | 10,317 | 210 |
| heading | 1,719 | 86 |
| presence | 1,450 | 77 |
| stack | 2,259 | 86 |
| breadcrumb | 3,537 | 116 |
| radio | 13,935 | 496 |
| progress | 11,892 | 359 |
| format | 2,572 | 134 |
| pagination | 16,881 | 254 |
| icon | 3,051 | 139 |
| checkbox | 13,930 | 464 |
| input | 3,568 | 224 |
| avatar | 11,467 | 403 |
| wrap | 3,186 | 56 |
| number | 3,941 | 206 |
| slider | 14,841 | 302 |
| color | 10,339 | 560 |
| data | 888 | 29 |

### A.2.4 Layout Statistics

We collected layout data from two primary sources: the UI design community Figma and through rollouts across operating systems. Statistics can be found in the following Tables 12 13. The number of elements shown in the following table is not the exact number in the final dataset. These elements are filtered in later processing stage.

Table 12: Statistics of OS Layout Data

| Rollout Environments | Screenshots | Elements |
|---|---|---|
| OSWorld (Ubuntu) | 2000 | 183889 |
| WindowsAgentArena (Windows) | 903 | 74445 |
| **Total** | **2903** | **258334** |

Table 13: Statistics of Layout Data Collected from Figma Commnuity Design Templates

| Design Templates | Images | Elements |
|---|---|---|
| [Freebie]-Home-Rent-App-UI-Design-(Community) | 3 | 59 |

| Design Templates | Images | Elements |
|---|---|---|
| (Variants)-macOS-Big-Sur-UI-Kit-for-Figma-(Community) | 10 | 685 |
| 10-Real-Chat/Messaging-Pages—Facebook, Reddit, Snapchat-&-more-(Community) | 10 | 1269 |
| 10-Real-Dashboard-Pages—AirBnB, Basecamp, Github, &-more-(Community) | 10 | 2214 |
| 10-Real-Homepages—AirBnb, Github, and-more-(Community) | 10 | 2162 |
| 10-Real-Notification-Pages—AirBnB, Dropbox, Notion, &-more-(Community) | 10 | 1103 |
| 10-Real-Pricing-Pages—Basecamp, Dribble, &-more-(Community) | 10 | 3554 |
| 10-Real-Search-Results-Pages—Github, Loom, Notion-&-more-(Community) | 11 | 1714 |
| 10-Real-Sign-Up-Pages—Calendly, Dribbble, &-more-(Community) | 11 | 570 |
| 10-Real-User-Settings-Pages—Calendly, Github, Behance, &-more-(Community) | 13 | 1006 |
| 11-Real-Sign-In-Pages—AirBnB, Calendly, &-more-(Community) | 12 | 557 |
| 20-Modals, Popups, Alerts-(Community) | 13 | 135 |
| AWS-Admin-Redesign-by-FluentUI-(Community) | 21 | 1456 |
| AWS-Amplify-UI-Kit-(Community) | 28 | 713 |
| AWS-Platform-(Community) | 5 | 690 |
| Ai-Design-Templates-(Community) | 10 | 1071 |
| Airbnb—Home, Search, and-Listing-Pages-(Community) | 5 | 905 |
| Airbnb-UI-Kit-(Community) | 10 | 49 |
| Amazon-UI-Design-(Community) | 18 | 3522 |
| Android-UI-Kit-(Community) | 28 | 2364 |
| App-Clips-(Community) | 4 | 77 |
| App-Store-Template—See-how-your-App-looks-like-in-App-Store-(Community) | 3 | 756 |
| Apple Design Resources - macOS (Community) | 29 | 1164 |
| Apple-Mail-(Community) | 1 | 283 |
| Apple-Mail-Design-(Community)-(Community) | 3 | 72 |
| Apple-Maps-iOS-(Community) | 7 | 351 |
| Apple-Messages-Templates-(Community) | 8 | 567 |
| Apple-Pay-(Community) | 18 | 442 |
| Apple-TV+-UI-Kit-(Community) | 16 | 1290 |
| Apple-Website-UI-2023-(apple.com)-(Community) | 7 | 2618 |
| Apple-Widgets-UI-Kit-(Community) | 78 | 264 |
| Apple-and-Google-Play-store-UI-(Community) | 6 | 130 |
| Apple-iCloud-Login-(Community) | 2 | 2 |
| Apps-Paywalls-and-Subscription-Screens-(Community) | 5 | 48 |
| Assets-Kit-UI-Mobile, Tablet-&-Desktop-(Community) | 45 | 986 |
| Audiobooks-by-Booksbury-(Community) | 3 | 76 |
| Betting-Mobile-app-(Community) | 6 | 162 |
| Binance-Market-Trade-Dashboard-UI-Design-(Community) | 1 | 584 |
| Booking.com-Mobile-App-Redesign—UX/UI-Case-Study-(Community) | 4 | 85 |
| Budddy-Chatbot-Freebie-(Community) | 8 | 241 |
| CAPTCHA-UI-Kit-(Community) | 16 | 124 |
| CAR-RENTAL-WEBSITE-(RESOONSIVE-DESIGN)-(Community) | 3 | 60 |
| Calendar-Interactive-UI-Kit-(Community) | 6 | 578 |
| Call-Center-Desktop-App-(Community) | 4 | 80 |
| Car-Rent-Website-Design—Pickolab-Studio-(Community) | 10 | 1232 |
| Car-Rental-Mobile-App-(Community) | 3 | 83 |
| Casino-Web-Site-(Community) | 12 | 3831 |
| Chat-for-desktop/mobile-l-Free-to-use-(Community) | 3 | 169 |
| ChatGPT-UI-Kit, AI-Chat-(Community) | 2 | 182 |
| Cinema-4D-GUI-Redesign-(Community) | 2 | 250 |
| Clicon—eCommerce-Marketplace-Website-Figma-Template-(Community) | 48 | 7178 |
| Club-Website-Design-l-WEB-UI-(Community) | 4 | 66 |
| Code-block, Syntax-highlighting | 2 | 20 |
| Coding-Website—UI-Kit-(Community) | 11 | 416 |
| Coinbase-Clone—Website-Prices-Page-(Community) | 1 | 448 |
| Components-library—Light-&-Dark-mode-(Community) | 13 | 256 |
| Concept- -Mailbox-Design-(Community) | 1 | 78 |

| Design Templates | Images | Elements |
|---|---|---|
| Coursera-UI-KIT-(Community) | 0 | 0 |
| Crypto-App-Ui-Kit-(Community) | 61 | 2430 |
| Customer-onboarding-designs-&-components—by-Bento-(Community) | 10 | 1315 |
| Dark-UI-Elements, Dropdowns-&-Calendar-(Community) | 4 | 254 |
| Dashboard—Online-Learning-Profile-(Community) | 3 | 201 |
| Dashboard-UI-Kit—Dashboard, Free-Admin-Dashboard-(Community) | 6 | 2224 |
| Data-table-design-components.-Free-UI-Kit-(Community) | 13 | 31059 |
| Dating-Mobile-App-(Community) | 42 | 722 |
| Delivery-App-Ui-Kit-(Community) | 54 | 2393 |
| Desktop-Messaging-App-Concept-(Community) | 1 | 26 |
| Deupload—Decentralized-Cloud-Storage-Landing-Pages-(Community) | 45 | 2188 |
| Discord-(Community) | 2 | 116 |
| Discord-Redesign-(Community) | 16 | 3236 |
| Discord-UI-Mockup-(Community) | 11 | 672 |
| Disney+-App-Redesign-(Community) | 2 | 6 |
| DocketHub-(Community) | 10 | 1121 |
| Doordash-FREE-UI-Kit—By-Marvilo-(Community) | 5 | 569 |
| Dota-2-UI-Redesign-(Community) | 12 | 2287 |
| Duolingo-Pages-Collection-by-DesignDrops.io-(Community) | 13 | 714 |
| Duolingo-Workflows—Onboarding, Learning-a-language, Upgrading, &-Cancelling-(Community) | 145 | 4974 |
| E-Store—Mobile/web-(Community) | 15 | 1502 |
| E-Tutor—Learning-Management-System-(Community) | 69 | 8475 |
| E-commerce-UI—Figma-Ecommerce-UI-Kit-(Demo-Version)-(Community) | 160 | 6522 |
| E-commerce-Website-Template-(Freebie)-(Community) | 9 | 858 |
| Ebay-New-Design-Concept-(Community) | 1 | 34 |
| Ecommerce-Website-Design-(Community) | 1 | 71 |
| Element-UI-Kit-2.15.7-(Community) | 42 | 2078 |
| Elite—Food-Restaurant-&-Coffee-Free-Figma-Template-(Community) | 16 | 769 |
| Email-Message-Modal-(Community) | 2 | 165 |
| Embed-Media-Components-(Community) | 6 | 38 |
| Eonify—Mobile-App-Authentication-Page-(Community) | 7 | 91 |
| FREE-Gmail-Mockup-2024-template!-(Community) | 4 | 98 |
| FREEBIES-Landingpage-LaslesVPN-(Community) | 1 | 28 |
| Facebook-Page-Mockup-(2022)-(Community) | 1 | 53 |
| Facebook-ReDesign-2023-(Community) | 7 | 449 |
| Fantastical-Calendar-(Community) | 1 | 655 |
| FigmaSharp-Toolkit: macOS-Big-Sur-2.0.0-(Community) | 5 | 573 |
| Finance-Market-Trading-Terminal-(Community) | 14 | 766 |
| Fitness-App-UI-Kit-for-Gym-Workout-App-Fitness-Tracker-Mobile-App-Gym-Fitness-Mobile-App-UI-Kit-(Community) | 87 | 1723 |
| Fiverr–UI-Redesigned—Freelance-Marketplace-Website-Design-(Community) | 3 | 681 |
| Flight-Booking-App-UI-Kits-(Community) | 20 | 249 |
| Food-Catering-Service-App-With-Landing-Page—Figma-Freebies-|-Doradesign-(Community) | 13 | 696 |
| Food-Delivery-Website-&-App-Design-UI-Kit-(Community) | 17 | 37 |
| Food-delivery-app-Ui-kit-(Community) | 18 | 117 |
| FoodWagon-Food-Delivery-Landing-Template-by-ThemeWagon-(Community) | 1 | 304 |
| Forms–/–Desktop-&-Mobile-(Community) | 12 | 341 |
| Forum-Concept-for-Alem.school-(Community) | 6 | 403 |
| Free-Fitness-App-Ui-Kit-(Community) | 48 | 841 |
| Free-Instagram-UI-Mockups-2023-(Community) | 12 | 364 |
| Free-Modal-Upload-Files-Kit-for-Web-and-Mobile—Include-4-modes-(Community) | 27 | 1150 |
| Free-Trading-UI-Kit-(Community) | 43 | 725 |
| Free-YouTube-Shorts-Mockups-(Community) | 18 | 634 |
| Free-YouTube-Video-Player-Mockups-(Community) | 6 | 506 |
| Freebies—Apps-Tracking-Truck-Cargo-Courier-Delivery-(Community) | 2 | 152 |
| Freebies—Scooter-Renting-App-(Community) | 4 | 77 |

**Table 13 – countined from previous page**

| Design Templates | Images | Elements |
| --- | --- | --- |
| Full-Apple-Music-Classical-App-(Community) | 160 | 18706 |
| Full-E-Commerce-Website-UI-UX-Design-(Community) | 15 | 2318 |
| GitHub-UI-(Community) | 2 | 1101 |
| Github-UI—Free-UI-Kit-(Recreated)-(Community) | 18 | 6568 |
| Gmail-UI-Mobile-Design-Template-2024!-(Community) | 4 | 38 |
| Gmail-UI-Part-1: Inbox-(Community) | 4 | 1940 |
| Gmail-UI-Part-2: Reading-&-Composing-Emails-(Community) | 9 | 2505 |
| Google-Anlytics-Dashobard-(Community) | 1 | 28 |
| Google-Calendar—Web-version-revamp-(Community) | 4 | 392 |
| Google-Chrome-Browser-UI-Kit-2025-(Community) | 14 | 695 |
| Google-Chrome-UI-Kit-2022-(Community) | 1 | 39 |
| Google-Drive-Reverse-Engineer-(Community) | 14 | 1620 |
| Google-Gemini—Built-with-Material-3-Design-Kit-(Community) | 4 | 646 |
| Google-Maps—Bus-ticket-booking-(Community) | 19 | 34 |
| Google-Maps-Parking-Prototype-Testing-(Community) | 5 | 68 |
| Google-Meet-UI-(Community) | 1 | 2 |
| Google-Scholar-re-designed-(Community) | 3 | 33 |
| Google-Search-Result-Page-(SERP)-(Community) | 2 | 167 |
| Google-Sheet—Template-(Unofficial)-(Community) | 23 | 3152 |
| Google-Sign-in-GIS—Google-Identity-Services-(Community) | 9 | 136 |
| Google-Translate-Redesign-(Community) | 9 | 140 |
| Google-Weather-App-Redesign-(Community) | 3 | 47 |
| Google-search-(Community) | 8 | 313 |
| Health-Fitness-Workout-App-(FREEBIE—Prototype)-(Community) | 8 | 332 |
| HealthRise-Health-Tech-Dashboard-(Community) | 4 | 4513 |
| Hero-Giveaway—Redesigns-(Community) | 7 | 1391 |
| Hotel-booking-website-UI-(Community) | 1 | 5 |
| Hoteliq—Booking-Hotel-App-Design-(Community) | 3 | 222 |
| IKEA-/-eCommerce-Concept-Design-(Community) | 4 | 26 |
| IMDb-Redesign-(Community) | 15 | 2293 |
| InTouch—Messaging-App-UI-Kit-(Community) | 4 | 87 |
| Instagram-UI-Screens-(Community) | 36 | 441 |
| IntelliJ-Platform-UI-Kit-(Community) | 48 | 4598 |
| Invoice/Payment-Components—Dipa-Inhouse-(Community) | 17 | 828 |
| Job-Finder-App-UI-Kit-(Community) | 1 | 43 |
| Job-Finder-Ui-App-Kit-(Community) | 83 | 85 |
| Jobpilot—Job-Portal-Figma-UI-Template-(Community) | 3 | 913 |
| LOGIFY—WEB-LOGIN-UI-KIT-(Community) | 40 | 273 |
| Leetcode-Homepage-(Community) | 1 | 1 |
| Lenskart-Redesigned—HiFi-Wireframes-(Community) | 5 | 85 |
| LinkedIn-Business-Page-Mockup-(2024)-(Community) | 1 | 111 |
| LinkedIn-Redesign-UI-Kit-(Community) | 8 | 223 |
| Linkedin-Page-Mockup-(2022)-(Community) | 1 | 30 |
| Linkedin-UI-Screens-(Community) | 28 | 948 |
| Liquipedia-Web-Redesign-(Community) | 4 | 1606 |
| Live-Score-UI-KIT-(FREEBIES)-(Community) | 12 | 349 |
| Login-&-Register-Web-UI-Kit-(Freebie)-(Community) | 5 | 170 |
| Loom-UI—Free-UI-Kit-(Recreated)-(Community) | 28 | 5405 |
| MEDDICAL—Hospital-website-template-(Community) | 10 | 189 |
| MacOS-file-upload-&-download-(Community) | 3 | 363 |
| Map-Navigation-Mobile-App-UI-Kit-Template-(Community) | 2 | 15 |
| Market-Stock-Exchange-(Community) | 6 | 601 |
| Medical-Clinic-Booking-(Doctor-Appointment)-App-UI-Concept-(Community) | 3 | 20 |
| Mercedes-Benz-App-(Community) | 8 | 9 |
| Messager-Dashboard-design.-(Community) | 9 | 582 |
| Metroway—Train-Ticket-booking-website-(Community) | 5 | 301 |
| Microsoft-365-UI-Kit-(Community) | 358 | 95741 |
| Microsoft-Excel-+-Word-2024-(Community) | 4 | 1733 |
| Mobile-Chat-Figma-UI-Kits-|-BRIX-Templates-(Community) | 70 | 10979 |
| Mobile-eCommerce-Clothing-Store-App-Design-(Community) | 6 | 322 |

*Continued on next page*

| Design Templates | Images | Elements |
|---|---|---|
| Modern-Profile-UI-Kit—Freebies-UI-(Community) | 4 | 6 |
| Money-transfer-Ui-App-Kit-(Community) | 55 | 236 |
| Movie-App-Redesigned-HULU-(Community) | 23 | 400 |
| Movie-Ticket-Booking-Application—Coursera-UX-Specialization-(Community) | 22 | 40 |
| Movie-Ticket-Booking-Apps-(Community) | 5 | 49 |
| MyCourses.io—Course-Website-|-Course-Online-|-Course-details-|-Course-landing-page-|-Untitled-UI-(Community) | 50 | 3731 |
| Native-Web-Components—Browser-Default's-UI-Kit-(Community) | 12 | 214 |
| Navigation-App-Design-(Waze-App-Redesign)-(Community) | 2 | 17 |
| Neomorphism-music-player-for-desktop-(Community) | 6 | 19 |
| Netflix-Home-Page-desktop-&-TV-(Community) | 1 | 8 |
| Netflix-home-page—Mobile-&-TV/Desktop-(Community) | 2 | 24 |
| News-&-Blog-App-UI-Kit-By-Al-Ferdous-(Community) | 6 | 46 |
| News-Website-UI-and-Presentation-for-Opportunists-(Community) | 2 | 2 |
| Nike-UI—Free-UI-Kit-(Recreated)-(Community) | 18 | 2769 |
| Nowted-—A-Note-taking-App-(Community) | 6 | 362 |
| Officevibe-UI—Free-UI-Kit-(Recreated)-(Community) | 18 | 3577 |
| On-Demand-Medicine-Delivery-App-(My-Orders-Flow)-(Community) | 9 | 379 |
| Onboarding-Appointment-booking-(Community) | 1 | 183 |
| Onest—Classified-Ads-Listing-Figma-Template-(Commnity) | 44 | 6139 |
| PDF-Viewer-(Community) | 1 | 18 |
| Papery—News-Magazine-Mobile-App-(Community) | 21 | 1145 |
| Parking-App-Design-UI-|-Figma-(Community) | 30 | 270 |
| Patterns & Layouts UI Kit (Community) | 108 | 12275 |
| Payment-Page-(Desktop)-(Community) | 2 | 184 |
| Picto—Personal-Portfolio-Free-Template-(Community) | 1 | 202 |
| Pinterest-Redesign-(Community) | 5 | 363 |
| Pinterest-UI—Free-UI-Kit-(Recreated)-(Community) | 4 | 716 |
| Plant-App-Freebies-(Community) | 13 | 69 |
| Print-dialog–Firefox-macOS-(Community) | 2 | 72 |
| Quiz-Game-(Community) | 12 | 206 |
| QuizGrad-webapp-(Community) | 6 | 132 |
| Quora-Redesign-(Community) | 2 | 10 |
| REIS—Real-State-Listing-Figma-Template-(Community) | 3 | 572 |
| Real-Estate-App-UI-Kit-(Community) | 79 | 3255 |
| Recreating-Google-Drive-Using-Lexicon-(Community) | 2 | 1002 |
| Reddit-Design-System-(Community) | 16 | 762 |
| Reddit-Material-Design-Redesign-(Community) | 14 | 142 |
| Redesign—ChatGPT-(Community) | 1 | 53 |
| Registration-Form-for-a-Medical-Laboratory-|-Medical-Analyzes-(Community) | 5 | 502 |
| Restaurant-Booking-Uikit-(Community) | 20 | 122 |
| Roommates-Apartments-Booking-(Community) | 2 | 54 |
| Sass Plat form Layouts - Wireframe Kit (Community) | 11 | 1073 |
| Scheddo—Bookings-&-Reservations-UI/UX—Freebie-(Community) | 7 | 255 |
| Shell-Template—Windows-11-(Community) | 87 | 6881 |
| Shopcart—Online-Ecommerce-website-(Community) | 1 | 67 |
| Shopery—Organic-eCommerce-Shop-Website-Figma-Template-(Community) | 37 | 6916 |
| Simple-Chat-Widget-for-Desktop-(Community) | 4 | 80 |
| Siri-&-App-Shortcuts-(Community) | 57 | 5077 |
| Slack-Desktop-App-Clone-(Community) | 5 | 238 |
| Slack-UI—Desktop-(Community) | 1 | 118 |
| Snow-Dashboard-UI-Kit-(Community) | 6 | 2608 |
| Soccer-Score-App-(Community) | 5 | 328 |
| Social-Login-Auth-Modals-(Community) | 5 | 49 |
| Sportify—Sports-streaming-app-(Community) | 34 | 1832 |
| Spotify—Mobile-UI-Kit-(Community) | 22 | 468 |
| Spotify-Redesign-(Community) | 32 | 6253 |
| Spotify-UI—Free-UI-Kit-(Recreated)-(Community) | 10 | 1329 |

| Design Templates | Images | Elements |
|---|---|---|
| Spotify-UI-Design-(Search/Artist-Profile)-(Community) | 2 | 580 |
| Starbucks-Redesign-Mobil-App-(Community) | 8 | 330 |
| Steam-Redesign-(Community) | 36 | 6855 |
| Stock-Trading-App—UI-Concept-(Community) | 6 | 302 |
| Stripe-Apps-UI-toolkit-(Community) | 35 | 2625 |
| Stripe-Connect-Embedded-Components–UI-Toolkit-(Community) | 63 | 6002 |
| Subscription-Paywall-Modal-(Community) | 1 | 56 |
| Table-Booking-Restaurant-Application-(Web-+-Mobile-+-Admin-Panels)-(Community) | 99 | 2524 |
| Table-UI-3.0-l-Variants-Update-(Community) | 1 | 223 |
| Tap-to-Pay-on-iPhone-(Community) | 2 | 20 |
| Tasky—Task-and-Time-Management-Dashboard-(Community) | 1 | 5 |
| Taxi-Booking-App-(Community) | 8 | 46 |
| Technical-Support-Applications-Page-(Community) | 4 | 266 |
| Telegram-Design-System-(Community) | 46 | 8535 |
| Terminal-app-UI-(Community) | 6 | 26 |
| Tesla-Mobile-App-Redesign-(Community) | 3 | 185 |
| The-Unofficial-Spotify-Design-System-(Community) | 5 | 854 |
| Ticketing-App-Freebies-(Community) | 8 | 549 |
| TikTok-UI-Screens-(Community) | 14 | 100 |
| Tinder-Mobile-App-(Community) | 23 | 192 |
| TipKit-(Community) | 8 | 67 |
| To-do-list-dashboard-(Freebie)-(Community) | 2 | 86 |
| ToDoHQ–Activity-management-website-design-(Community) | 18 | 329 |
| Todoist-Free-UI-Kit—By-Marvilo-(Community) | 8 | 868 |
| Todoist-for-macOS-app-concept-(Community) | 3 | 1196 |
| Tour-Guide—travel-agency/travel-booking-website-(Community) | 4 | 155 |
| Travel-&-Hotel-Booking-Light-Mobile-App-(Community) | 4 | 171 |
| Trello-Concept-(Community) | 2 | 2 |
| Twitch-UI—Autolayout-Interface-(Community) | 1 | 182 |
| Twitch-UI—Free-UI-Kit-(Recreated)-(Community) | 3 | 1670 |
| Twitter-UI-Clone-Design-(Community) | 9 | 992 |
| Twitter-UI-Screens-(Community) | 22 | 315 |
| Twitter-desktop-pages-(feed, sigup, login, profile)-(Community) | 5 | 175 |
| UF-File-Manager-(Community) | 15 | 280 |
| UI-DESIGN-FOR-MOCK-INTERVIEW-PLATFORM-(Company-side)-(Community) | 50 | 528 |
| Uber-App-UI—Free-UI-Kit-(Recreated)-(Community) | 3 | 306 |
| Uber-Redesign-(Community) | 3 | 19 |
| Ubuntu-Shiro-(Community) | 10 | 606 |
| VPN-App—UI-Kit-(Community) | 9 | 200 |
| Video-Player-For-Web-&-Mobile-(Community) | 9 | 186 |
| Video-Streaming-Website—Responsive-web-app-prototype-(Community) | 3 | 50 |
| Visual-Studio-Code-Toolkit-(Community) | 45 | 8887 |
| Wallet-(Community) | 12 | 230 |
| WeChat-(Community) | 11 | 318 |
| WeUI-kit(Wechat)-(Community) | 33 | 391 |
| Web-Browser-Mockups-(Community) | 4 | 60 |
| Web-Dashboard-UI—Task-&-Project-Management-(Community) | 1 | 7 |
| Website-FAQ-Accordions-Figma-Template-l-BRIX-Templates-(Community) | 3 | 3 |
| Website-Wireframes-UI-Kit-l-BRIX-Templates-(Community) | 108 | 1800 |
| WhatsApp-Pay-&-Split-(Community) | 24 | 294 |
| Wikipedia-(Community) | 44 | 11773 |
| Windows-11-Chat-UI-Kit-(Community) | 15 | 857 |
| Windows-File-Explorer-—-Ego's-Take-(Community) | 18 | 3384 |
| Windows-Install-Redesigned-(Concept)-(Community) | 47 | 22947 |
| Windows-Outlook-Template-(Community) | 17 | 2269 |
| WordPress-Design-System-(Community) | 12 | 5289 |
| YouTube-Music-App-Redesign: Elevating-the-Music-Experience-(Community) | 3 | 50 |
| YouTube-Redesign-(Community) | 34 | 9176 |

| Design Templates | Images | Elements |
|---|---|---|
| YouTube-UI-Clone-Design-(Community) | 10 | 1801 |
| Zoom-Apps-UI-Overview-(Community) | 19 | 4507 |
| aeroSpeed-Bus-Booking-Application-UI-Kit-[User-+-Driver]-(Community) | 14 | 330 |
| chat-app-UI-kit-(Community) | 5 | 7 |
| eDex—Online-Course-E-Learning-Website-(Comunity) | 2 | 447 |
| iBank—Banking-&-E-Money-Management-App-|-FinPay-|-Digital-|-Finance-Mobile-Banking-App-Ui-Kit-(Community) | 89 | 2027 |
| iMessage-Apps-and-Stickers-(Community) | 22 | 1761 |
| iOS-17-Apple-music-Now-Playing-interface-(Community) | 4 | 4 |
| iOS-18-and-iPadOS-18-(Community) | 91 | 7171 |
| lark-(Community) | 28 | 45169 |
| macOS-Big-Sur-UI-Kit-(Community) | 20 | 3214 |
| macOS-Browser-UI-Kit-(Big-Sur-Update)-(Community) | 6 | 51 |
| telegram-app-(Community) | 15 | 297 |
| ui—Design-System-(Community) | 17 | 343 |
| **Total** | **5273** | **563721** |

## A.2.5 Cost Analysis

We utilized GPT-4o throughout our data generation pipeline, with all cost calculations based on GPT-4o's token pricing. The comprehensive cost breakdown for the three components of the JEDI dataset is detailed below:

- **Icon Data (0.4M samples):** We employed input prompts and images to generate visual and functionality descriptions for icons. Each sample incurred approximately $0.01 in processing costs, resulting in a total expenditure of ~$4,000.

- **Component Data (1M samples):** This category comprised two distinct subsets:
    - *Template-based fine-grained operations* (~40K samples): Generated using predefined template rules for slides and sheets data, incurring no additional costs.
    - *Code-rendered data* (~1M samples): Costs were distributed across component rendering, action generation, and filtering processes, averaging ~$0.025 per sample, totaling ~$25,000.

- **Layout Data (2.3M samples from 0.8M captions):** We leveraged GPT-4o to generate comprehensive screenshot captions. Each caption required processing of approximately 3 images (~2,100 tokens) plus prompts (~550 tokens), with an average output of ~250 tokens. This resulted in a cost of ~$0.0091 per caption, totaling ~$7,000 for the complete set of 0.8M captions.

The aggregate cost for utilizing GPT-4o across all data generation tasks amounted to approximately **$36,000**.

### A.3    JEDI Dataset Construction: A Detailed Pipeline for Component

#### A.3.1    Component Collection and Style Augmentation

We begin by collecting example components from four mainstream UI libraries hosted on GitHub: `Material UI`, `Ant Design`, `Mantine UI`, and `Chakra UI`. From each repository, we extract example code snippets(in `typescript`) that showcase usage of individual components.

To diversify these examples, we apply style augmentation using two LLMs: `GPT-4o` and `Claude-3.5-Sonnet`. For each original code snippet, we first ask the model to envision a unique UI usage scenario. Based on the original code and the imagined context, it then generates a stylistically augmented variant code.

This process is repeated multiple times per example, each time with a different context to promote diversity. Previously generated variants are included in the prompt to prevent redundancy across augmented examples.

#### A.3.2    Rendering and Interaction Preparation

Each augmented component is rendered on a React application. Components are wrapped in a container with a randomized position to mitigate positional overfitting. Using `Playwright`, we programmatically open and interact with the rendered pages.

We extract **screenshots** of the rendered component and **element tree information** (positioning, hierarchy, etc.) using `Playwright`'s `evaluate` method and custom JavaScript.

These outputs are used to generate component-grounded actions via two distinct pipelines.

**Pipeline 1: Component-level Action Generation**

**Step 1: Generate Action Intents**

We prompt GPT-4o with component name, component code and screenshot. GPT-4o returns a list of action intents, each representing a high-level user interaction. We use few-shot examples to guide this process.

**Step 2: Generate Action Details**

For each intent, we generate detailed interaction metadata using component name, component code, screenshot, element position tree and action intent.

Each action detail includes:

1. **Thought Process**: The thinking process of generating an action detail
2. **Action Space Type**:
   - `None`: No action space exists,
   - `Unique`: Only one possible action exists (e.g., clicking a button),
   - `Discrete`: Limited/unlimited set of distinct possible actions (e.g., selecting from a list of options),
   - `Continuous`: Infinite possible actions within a range (e.g., dragging a slider to any position)
3. **Action Description**: Describe what the action does, which serves as the instantiation/implementation of the action intent.
4. **Action Parameters**: List of all parameter names for the action function(in action code)
5. **Discrete Values**: List of all possible parameter values for discrete action spaces (if applicable)
6. **Continuous Intervals**: List of interval for all possible parameter values for continuous action spaces (if applicable)
7. **Action Code**: A function using `PyAutoGUI` to represent one action or a kind of actions

**Example**

```
{
    "thought_process": "The target element is a slider, which
        provides a continuous range of values from 0 to 100. The
        action involves setting a specific value within this range by
        determining the corresponding position on the slider bar and
        simulating a click at that position. The slider's endpoints
        are identified, and linear interpolation will be used to
        calculate the appropriate position based on the desired
        value.",
```

```
3      "action_space_type": "continuous",
4      "action_desc": "Set saturation to <saturation>%",
5      "action_params": [
6          "saturation"
7      ],
8      "action_discrete_values": null,
9      "action_continuous_interval": {
10         "saturation": [
11             [
12                 0.0,
13                 100.0
14             ]
15         ]
16     },
17     "action_code": "def action(saturation):\n    x_0, y_0 = 600.5,
          830  # Left endpoint\n    x_1, y_1 = 1064.5, 830  # Right
          endpoint\n    x = x_0 + (x_1 - x_0) * (saturation / 100)\n
          pyautogui.click(x, y_0)"
18 }
```

We then convert the action code (e.g., `def action(parameter):    ...`, often involving `pyautogui`) into one or more pieces of *grounding data*—such as `pyautogui.click(x, y)`—by sampling from the corresponding action space. If the action space is `None`, no sampling is needed. This conversion is guided using few-shot examples. An example of this process can be seen below.

### Example

**Instruction:** *Set saturation to <saturation>*

**Action code:**

```
def action(saturation):
    x_0, y_0 = 600.5, 830   # Left endpoint of the saturation slider
    x_1, y_1 = 1064.5, 830  # Right endpoint of the saturation slider
    x = x_0 + (x_1 - x_0) * (saturation / 100)
    pyautogui.click(x, y_0)
```

**Sampled grounding data:**

```
# Set saturation to 24%
pyautogui.click(711.86, 830)

# Set saturation to 60%
pyautogui.click(878.90, 830)
...
```

### Pipeline 2: Element-Level Action Generation

**Step 1: Element Extraction and Filtering**   We render each augmented component in a browser and traverse the DOM tree to collect element nodes. Two filtering rules are applied:

- **Duplicate boxes**: Only one node is retained if multiple share the same bounding box.
- **Abnormal sizes**: Nodes with very small or very large bounding boxes are discarded.

For each valid node, we collect position, text, visibility, interactivity, parent-child relationships, and metadata.

**Step 2: Multimodal Context Encoding**   To help GPT-4o understand each element, we provide element box, parent box, cropped screenshot(cropped screenshot with only the element region), context screenshot(cropped screenshot with element region and nearby surroundings, with the element highlighted in red bounding box) and full-page screenshot(full screenshot with the element highlighted in red bounding box) as input. And the model outputs include visual description(a detailed account of the element's appearance), position textual information(spatial relationship relative to the viewport and its parent), element functionality, UI type (e.g., button, slider) and possible actions at element center.

To ensure quality, we also include visibility check and atomicity check, to check whether this element is a single visible UI unit.

**Step 3: Action Detail Generation** For each possible action, we prompt GPT-4o with the action and relevant element information—including visual description, position, text content, functionality, and UI type. The model is asked to generate detailed action information, including the thought process, action description, action parameters, and action code. This is similar to the action detail in Pipeline 1, but limited to the **unique action space**.

**Step 4: Continuous Action Detection** To identify elements like sliders that support continuous interactions, the model determines whether the element has a continuous action space and generates the corresponding thought process, action description, action parameters, value range(action_continuous_interval), and action code. This step parallels the action detail in Pipeline 1, but focuses solely on the **continuous action space**.

**Step 5: Grounding Actions** We convert each action code into one or more grounding samples, similar to that in Pipeline 1.

### A.3.3 Comparison of Pipelines

- **Pipeline 1** is simpler. However, it may suffer from inaccurate bounding box targeting, limited action diversity and action vagueness.
- **Pipeline 2** generates data with better localization and diversity. **In practice, most of our dataset is generated using Pipeline 2.**

### A.3.4 Post-Processing and Filtering

To ensure data quality, we apply multiple filtering stages.

**1. Visual Filter (via `GPT-4o`)**

Given:

- Cropped screenshot
- Marked screenshot (click position highlighted with a green dot and circle)
- Full screenshot (element highlighted)

`GPT-4o` filters out data that:

1. Shows visible errors (e.g., "Compiled with problems" or red overlays)
2. Targets an incorrect GUI element
3. Has incorrect click localization (e.g., not centered on button/text)

**2. Instruction Filter (LLM-Based)**

Using `GPT-4o-mini`, we filter out ambiguous or low-quality instructions from Pipeline 1:

1. Unclear or vague semantics
2. Multiple interactive targets
3. References to non-visual identifiers like "index 1"
4. Multi-step or compound interactions

**3. Instruction Filter (Rule-Based)**

We filter instructions with high error likelihood based on pattern rules:

1. Contains explicit coordinates (e.g., `(x, y)`): Instructions referencing raw screen coordinates are filtered out, as such positional references are not meaningful in a vision-only context.
2. Mentions structural terms such as `child`, `parent`, `path`, or `container`: These terms imply hierarchical relationships derived from accessibility trees, which are not observable in visual input.
3. Mentions a `card` component without spatial qualifiers such as `in`, `within`, or `at`: Such instructions typically refer to an entire composite element (e.g., a card) rather than a specific atomic component within it, resulting in ambiguous interaction targets.
4. Includes directional terms in combination with `screen`: Phrases like "top-left of the screen" are frequently found to be incorrect or misaligned with actual component layouts, likely due to LLM misinterpretation.

5. **Refers to highlights or visual annotations** (e.g., red `dot`, `circle`, `highlight`): These often result from the model misidentifying annotation markers (used to denote interaction points) as intrinsic parts of the interface.

6. Mentions textual UI elements (e.g., `text`, `label`, `heading`) in combination with interaction verbs (e.g., `read`, `hover`, `click`, `interact`): If the associated bounding box is visually simple—based on low color variance and edge density—it often indicates that the relevant text is located on the periphery of the box, while its center is visually empty, leading to inaccurate click localization.

7. Refers to sliders without specifying interaction values: Instructions such as "interact with the slider" without numerical targets are prone to ambiguity and do not provide sufficient grounding for generating actionable behavior.

### A.3.5 Real-world augmentation pipeline

Office software, including document editors, presentation tools, and spreadsheets, is integral to daily work for many. Automating workflows in these applications can significantly boost productivity. However, a gap exists between synthetic use cases and real-world scenarios, as synthetic datasets often lack sufficient office software-related cases. To bridge this, we propose a targeted approach to designing and generating relevant data.

Our methodology centers on creating two pools: **a resource pool** and **an action pool**. The resource pool includes a diverse set of office files, such as Excel spreadsheets, Word documents, and PowerPoint slides, sourced from the web, including online tutorials. The action pool enumerates common tasks performed in these applications, such as scrolling through a document, clicking specific cells in a spreadsheet, or auto-filling data in Excel. For each action, we manually analyze the associated structural components and develop code to extract relevant coordinate arrangements. For example, consider the action of "scrolling a document" in Microsoft Word Online. The associated component is the scrollbar. We analyze the webpage structure to identify features that precisely locate the scrollbar, then use code to extract its coordinates, synthesizing a data instance. Similarly, in Excel, for the action "click the center of cell B3," we leverage the accessibility tree and HTML DOM structure to extract cell positions, generating precise instructions like "click the center of cell B3," "auto-fill from the bottom-right corner of cell A1," or "select column D." These rule-based extraction methods ensure accurate component-level interactions across productivity applications.

Additional actions and their components, including spreadsheet-specific tasks, are detailed in Table 14.

Table 14: Actions and Associated Components in Office Software

| Office Software | Action Type | Associated Component |
|---|---|---|
| Doc | Scroll | Scrollbar |
| Doc | Select | NormalTextRun |
| Doc | Click | NormalTextRun |
| Slide | Drag | Text Box |
| Slide | Click | Text Box |
| Slide | Click | Slide Thumbnail |
| Sheet | Click | Cell |
| Sheet | Click | Edge |
| Sheet | Click | Cell Corner |
| Sheet | Click | Column Header |
| Sheet | Click | Row Header |

## A.4 Additional Data Examples

### A.4.1 Icon data

**Icon description data**

In the icon description data, each data item includes an image of a single icon as input and produces a description that includes both its visual appearance and functionality as output.

An example in JSON format is shown below. The corresponding image (Figure 9a) features an icon labeled krl_access, which likely represents access to a commuter rail or train service—such as the KRL (Kereta Rel Listrik) system.

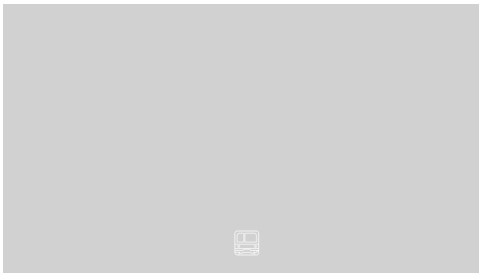
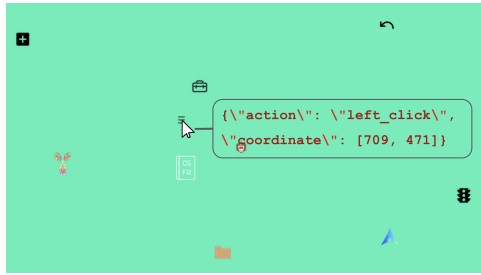

(a) Example of icon description data  (b) Example of icon grounding data

Figure 9: Examples of icon data

```
1  {
2      "conversations": [
3          {
4              "from": "system",
5              "value": {
6                  "text": "You are a GUI assistant that helps users
                        understand and interact with interface elements.
                        \nWhen describing elements, focus on their visual
                        appearance and functionality in clear, natural
                        language.\nFor each element, describe the visual
                        appearance and functionality:\n1. What it looks
                        like - including shape, color, symbols, and any
                        text\n2. What it does when used - its purpose and
                        effect\n3. When users might want to use it -
                        common use cases and scenarios"
7              }
8          },
9          {
10             "from": "user",
11             "value": [
12                 {
13                     "image":
                            "images_pure_color_background/categorized_icons/Arcticons-Team/Arc
14                     "width": 1920,
15                     "height": 1080,
16                     "resized_height": 1092,
17                     "resized_width": 1932,
18                     "seq_len": 2693
19                 },
20                 {
21                     "text": "Please describe what this interface
                            element looks like and what it does."
22                 }
23             ]
24         },
25         {
26             "from": "assistant",
27             "value": [
28                 {
29                     "text": "Visual Appearance:\nThe icon is a
                            simple, white outline of a train or subway
                            car on a black background. It features a
                            rectangular shape with rounded corners, two
                            windows at the top, and a front bumper at the
                            bottom. The design is minimalistic and uses
                            clean lines to represent the front view of a
                            train.\n\nFunctionality: The icon labeled
                            'krl_access' likely represents access to a
                            commuter rail or train service, such as a KRL
```

```
                                 (Kereta Rel Listrik) system. It might be used
                                 in applications or websites to indicate train
                                 schedules, ticket purchasing, or station
                                 information. Clicking this icon could lead to
                                 more details about train services or access
                                 to a train-related feature."
30                        }
31                   ]
32             }
33        ],
34        "image_id": "d4e5596b-c166-4345-a7d0-947014505f0a"
35 }
```

### Icon grounding data

In the icon grounding data, each data item includes an image containing multiple icons and a description of the target icon as input, and produces a grounding action (e.g., a click) in the Qwen2.5-VL tool-use format as output.

An example in JSON format is provided below. The image (Figure 9b) contains several icons, and the description corresponds to one commonly used icon which represents "density" or "line spacing" in user interfaces.

```
1 {
2      "conversations": [
3           {
4                "from": "system",
5                "value": {
6                     "text": "You are a helpful assistant.\n\n#
                          Tools\n\nYou may call one or more functions to
                          assist with the user query.\n\nYou are provided
                          with function signatures within <tools></tools>
                          XML tags:\n<tools>\n{\"type\": \"function\",
                          \"function\": {\"name\": \"computer_use\",
                          \"description\": \"Use a mouse and keyboard to
                          interact with a computer, and take
                          screenshots.\\n* This is an interface to a
                          desktop GUI. You do not have access to a terminal
                          or applications menu. You must click on desktop
                          icons to start applications.\\n* Some
                          applications may take time to start or process
                          actions, so you may need to wait and take
                          successive screenshots to see the results of your
                          actions. E.g. if you click on Firefox and a
                          window doesn't open, try wait and taking another
                          screenshot.\\n* The screen's resolution is 1092
                          x1932.\\n* Whenever you intend to move the cursor
                          to click on an element like an icon, you should
                          consult a screenshot to determine the coordinates
                          of the element before moving the cursor.\\n* If
                          you tried clicking on a program or link but it
                          failed to load, even after waiting, try adjusting
                          your cursor position so that the tip of the
                          cursor visually falls on the element that you
                          want to click.\\n* ..."
7                }
8           },
9           {
10               "from": "user",
11               "value": [
12                    {
13                         "image": "images_grounded/categorized_icons-
14                         /tabler/tabler-icons/baseline-density-medium.png",
15                         "width": 1920,
16                         "height": 1080,
17                         "resized_height": 1092,
18                         "resized_width": 1932,
```

```
19                      "seq_len": 2693
20                  },
21                  {
22                      "text": "Click on the icon with functionality:
                           This icon is typically used to represent a
                           'density' or 'line spacing' setting in user
                           interfaces. It suggests adjusting the spacing
                           between lines or items to a medium level.
                           Clicking this icon might change the display
                           settings to a medium density, balancing
                           between compact and spacious layouts."
23                  }
24              ]
25          },
26          {
27              "from": "assistant",
28              "value": [
29                  {
30                      "text": "<tool_call>\n{\"name\":
                           \"computer_use\", \"arguments\": {\"action\":
                           \"left_click\", \"coordinate\": [709, 471
                           ]}}\n</tool_call>"
31                  }
32              ]
33          }
34      ],
35      "image_id": "cb1cb4ba-61ab-4caf-8c8d-a57ba3a6c310"
36 }
```

### A.4.2 Component data

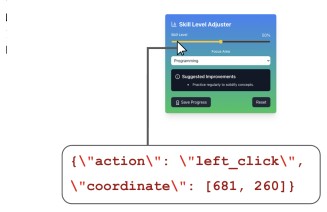

(a) Example of rendered component grounding data

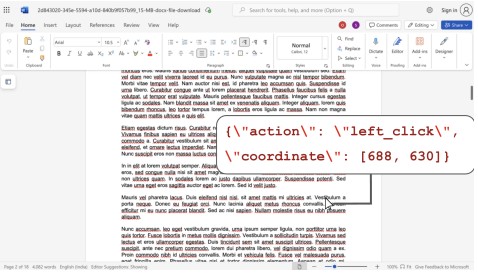

(b) Example of doc grounding data

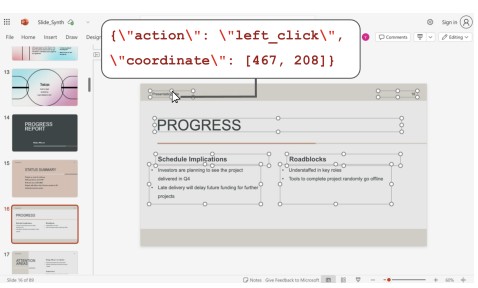

(c) Example of slide grounding data

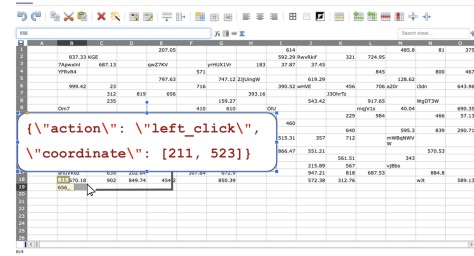

(d) Example of sheet grounding data

Figure 10: Examples of component data

**Component data from Code-and-rendering pipeline**

In the component data from the Code-and-Rendering pipeline, each data item includes an image containing a rendered UI component and a corresponding user instruction as input, and produces a grounding action (e.g., a click) in the Qwen2.5-VL tool-use format as output.

An example in JSON format is shown below. The associated image (Figure 10a) displays a slider component, and the instruction reads "...Read the text label displaying 'UI/UX Design Basics' located in the lower third of the catalog interface, which is part of a book listing..."

```
 1  {
 2      "conversations": [
 3          {
 4              "from": "system",
 5              "value": {
 6                  "text": "You are a helpful assistant.\n\n#
                        Tools\n\nYou may call one or more functions to
                        assist with the user query.\n\nYou are provided
                        with function signatures within <tools></tools>
                        XML tags:\n<tools>\n{\"type\": \"function\",
                        \"function\": {\"name\": \"computer_use\",
                        \"description\": \"Use a mouse and keyboard to
                        interact with a computer, and take
                        screenshots.\\n* This is an interface to a
                        desktop GUI. You do not have access to a terminal
                        or applications menu. You must click on desktop
                        icons to start applications.\\n* Some
                        applications may take time to start or process
                        actions, so you may need to wait and take
                        successive screenshots to see the results of your
                        actions. E.g. if you click on Firefox and a
                        window doesn't open, try wait and taking another
                        screenshot.\\n* The screen's resolution is 1092
                        x1932.\\n* Whenever you intend to move the cursor
                        to click on an element like an icon, you should
                        consult a screenshot to determine the coordinates
                        of the element before moving the cursor.\\n* If
                        you tried clicking on a program or link but it
                        failed to load, even after waiting, try adjusting
                        your cursor position so that the tip of the
                        cursor visually falls on the element that you
                        want to click..."
 7              }
 8          },
 9          {
10              "from": "user",
11              "value": [
12                  {
13                      "image": "data/chakra/slider/other_screenshot/-
14                      original/SkillLevelAdjuster_1742005837.475969.png",
15                      "width": 1920,
16                      "height": 1080,
17                      "resized_height": 1092,
18                      "resized_width": 1932,
19                      "seq_len": 2693
20                  },
21                  {
22                      "text": "Please generate the next move according
                            to the UI screenshot and
                            instruction.\n\nInstruction: Set the skill
                            level to 4% on the slider."
23                  }
24              ]
25          },
26          {
27              "from": "assistant",
28              "value": [
```

```
29                        {
30                            "text": "<tool_call>\n{\"name\":
                                  \"computer_use\", \"arguments\": {\"action\":
                                  \"left_click\", \"coordinate\": [681, 260
                                  ]}}\n</tool_call>"
31                        }
32                    ]
33                }
34        ],
35        "image_id": "568b8930-ec6a-4574-9b75-a18ed2c87cc0"
36 }
```

### Component data for real-world augmentation

In the component data for real-world augmentation, each data item includes an image containing a real-world screenshots from an existing website or application and a corresponding user instruction as input, and produces a grounding action (e.g., a click) in the Qwen2.5-VL tool-use format as output.

We provide one example for each of the three data sources: *doc*, *slide*, and *sheet*. The user instructions for these examples are as follows:

- **Doc:** *Given the following text: ". Vestibulum a", find this text in the document and click the space between the consecutive characters "t" and "i".*
- **Slide:** *Please generate the next move according to the UI screenshot and instruction. Instruction: Select the handle located at the top of the text box that contains the text "Presentation title."*
- **Sheet:** *Navigate to the top-left corner of cell C19.*

Corresponding examples in JSON format are shown below. The associated UI screenshots are provided in Figures 10b, 10c, and 10d, respectively.

Example of doc grounding data:

```
1  {
2      "conversations": [
3          {
4              "from": "system",
5              "value": {
6                  "text": "You are a helpful assistant.\n\n#
                          Tools\n\nYou may call one or more functions to
                          assist with the user query.\n\nYou are provided
                          with function signatures within <tools></tools>
                          XML tags:\n<tools>\n{\"type\": \"function\",
                          \"function\": {\"name\": \"computer_use\",
                          \"description\": \"Use a mouse and keyboard to
                          interact with a computer, and take
                          screenshots.\\n* This is an interface to a
                          desktop GUI. You do not have access to a terminal
                          or applications menu. You must click on desktop
                          icons to start applications.\\n* Some
                          applications may take time to start or process
                          actions, so you may need to wait and take
                          successive screenshots to see the results of your
                          actions. E.g. if you click on Firefox and a
                          window doesn't open, try wait and taking another
                          screenshot.\\n* The screen's resolution is 728
                          x1288.\\n* Whenever you intend to move the cursor
                          to click on an element like an icon, you should
                          consult a screenshot to determine the coordinates
                          of the element before moving the cursor.\\n* If
                          you tried clicking on a program or link but it
                          failed to load, even after waiting, try adjusting
                          your cursor position so that the tip of the
                          cursor visually falls on the element that you
                          want to click..."
7              }
```

```
 8              },
 9              {
10                  "from": "user",
11                  "value": [
12                      {
13                          "image":
                              "AmHHgw-Nep9dv1S3X9n5gaoKsDxY_1280x720_SPACE_563_5.png",
14                          "width": 1280,
15                          "height": 720,
16                          "resized_height": 728,
17                          "resized_width": 1288,
18                          "seq_len": 1198
19                      },
20                      {
21                          "text": "Given the following text:\n\".
                              Vestibulum a \"\n, find the text in the
                              document and click the space between the
                              continuous character \"t\" and \"i\" in the
                              text."
22                      }
23                  ]
24              },
25              {
26                  "from": "assistant",
27                  "value": [
28                      {
29                          "text": "<tool_call>\n{\"name\":
                              \"computer_use\", \"arguments\": {\"action\":
                              \"left_click\", \"coordinate\": [688, 630
                              ]}}\n</tool_call>"
30                      }
31                  ]
32              }
33          ],
34          "image_id": "3d35b0b2-d541-45ea-be23-b668263b5b69"
35 }
```

Example of slide grounding data:

```
 1 {
 2      "conversations": [
 3          {
 4              "from": "system",
 5              "value": {
 6                  "text": "You are a helpful assistant.\n\n#
                      Tools\n\nYou may call one or more functions to
                      assist with the user query.\n\nYou are provided
                      with function signatures within <tools></tools>
                      XML tags:\n<tools>\n{\"type\": \"function\",
                      \"function\": {\"name\": \"computer_use\",
                      \"description\": \"Use a mouse and keyboard to
                      interact with a computer, and take
                      screenshots.\\n* This is an interface to a
                      desktop GUI. You do not have access to a terminal
                      or applications menu. You must click on desktop
                      icons to start applications.\\n* Some
                      applications may take time to start or process
                      actions, so you may need to wait and take
                      successive screenshots to see the results of your
                      actions. E.g. if you click on Firefox and a
                      window doesn't open, try wait and taking another
                      screenshot.\\n* The screen's resolution is 728
                      x1288.\\n* Whenever you intend to move the cursor
                      to click on an element like an icon, you should
                      consult a screenshot to determine the coordinates
```

```
                        of the element before moving the cursor.\\n* If
                        you tried clicking on a program or link but it
                        failed to load, even after waiting, try adjusting
                        your cursor position so that the tip of the
                        cursor visually falls on the element that you
                        want to click..."
 7                  }
 8              },
 9              {
10                  "from": "user",
11                  "value": [
12                      {
13                          "image": "slides_1280*720/slide_15/original.png",
14                          "width": 1280,
15                          "height": 720,
16                          "resized_height": 728,
17                          "resized_width": 1288,
18                          "seq_len": 1198
19                      },
20                      {
21                          "text": "Please generate the next move according
                              to the UI screenshot and
                              instruction.\n\nInstruction: Select the
                              handle located at the top of the text box
                              that contains the text \"Presentation
                              title.\""
22                      }
23                  ]
24              },
25              {
26                  "from": "assistant",
27                  "value": [
28                      {
29                          "text": "<tool_call>\n{\"name\":
                              \"computer_use\", \"arguments\": {\"action\":
                              \"left_click\", \"coordinate\": [467, 208
                              ]}}\n</tool_call>"
30                      }
31                  ]
32              }
33      ],
34      "image_id": "3f2ebbae-dee3-4fea-bbc0-ab93136bedab"
35 }
```

Example of sheet grounding data:

```
 1 {
 2      "conversations": [
 3          {
 4              "from": "system",
 5              "value": {
 6                  "text": "You are a helpful assistant.\n\n#
                      Tools\n\nYou may call one or more functions to
                      assist with the user query.\n\nYou are provided
                      with function signatures within <tools></tools>
                      XML tags:\n<tools>\n{\"type\": \"function\",
                      \"function\": {\"name\": \"computer_use\",
                      \"description\": \"Use a mouse and keyboard to
                      interact with a computer, and take
                      screenshots.\\n* This is an interface to a
                      desktop GUI. You do not have access to a terminal
                      or applications menu. You must click on desktop
                      icons to start applications.\\n* Some
                      applications may take time to start or process
                      actions, so you may need to wait and take
```

```
                            successive screenshots to see the results of your
                            actions. E.g. if you click on Firefox and a
                            window doesn't open, try wait and taking another
                            screenshot.\\n* The screen's resolution is 728
                            x1288.\\n* Whenever you intend to move the cursor
                            to click on an element like an icon, you should
                            consult a screenshot to determine the coordinates
                            of the element before moving the cursor.\\n* If
                            you tried clicking on a program or link but it
                            failed to load, even after waiting, try adjusting
                            your cursor position so that the tip of the
                            cursor visually falls on the element that you
                            want to click..."
 7              }
 8          },
 9          {
10              "from": "user",
11              "value": [
12                  {
13                      "image":
                            "cell_left_top_corner/images/1280x720_cell_C19.png",
14                      "width": 1280,
15                      "height": 720,
16                      "resized_height": 728,
17                      "resized_width": 1288,
18                      "seq_len": 1198
19                  },
20                  {
21                      "text": "Navigate to top left corner of C19"
22                  }
23              ]
24          },
25          {
26              "from": "assistant",
27              "value": [
28                  {
29                      "text": "<tool_call>\n{\"name\":
                            \"computer_use\", \"arguments\": {\"action\":
                            \"mouse_move\", \"coordinate\": [211, 523
                            ]}}\n</tool_call>"
30                  }
31              ]
32          }
33      ],
34      "image_id": "15987cec-74d4-4624-b539-43a114caac8f"
35 }
```

### A.4.3 Layout data

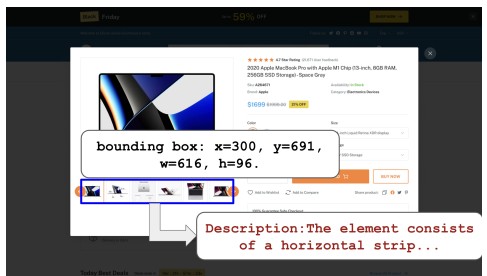

(a) Example of icon description data          (b) Example of icon grounding data

Figure 11: Examples of component grounding data

**Layout description data**

In the layout description data, each item includes a screenshot and the bounding box coordinates of a target element as input, and generates a description covering the element's visual composition, spatial context, user interaction, and type.

An example in JSON format is provided below. The associated image (Figure 11a) shows a product interface from a shopping website, where the target element is an image carousel featuring thumbnail views of a laptop.

```
1  {
2      "conversations": [
3          {
4              "from": "system",
5              "value": {
6                  "text": "Working as a UI navigation guide, you will
                        analyze a screenshot and bounding box
                        coordinates. Your objective is to provide a
                        detailed, user-friendly description of the UI
                        element within the specified region, helping
                        users identify it without relying on coordinates.
                        Address these key points:\n\n1. Visual
                        Composition: Detail the element's visual aspects,
                        including its geometry, color scheme, and
                        styling\n2. Spatial Context: Describe the
                        element's position in relation to surrounding
                        interface components\n3. User Interaction:
                        Explain the element's functionality and how users
                        should interact with it\n4. Element Type: Specify
                        the category of UI component (such as progress
                        bar, tooltip)\n\nMaintain precision and clarity
                        in your descriptions, avoiding vague references.
                        Emphasize distinctive characteristics and
                        contextual relationships that make the element
                        easily identifiable."
7              }
8          },
9          {
10             "from": "user",
11             "value": [
12                 {
13                     "image": "ce0ff960..."(Truncated due to layout
                        constraints),
14                     "width": 1920,
15                     "height": 1080,
16                     "resized_height": 1092,
17                     "resized_width": 1932,
18                     "seq_len": 2693
19                 },
20                 {
21                     "text": "bounding box: x=300, y=691, w=616, h=96.
                        Generate pls."
22                 }
23             ]
24         },
25         {
26             "from": "assistant",
27             "value": [
28                 {
29                     "text": "## Visual Composition: The element
                        consists of a horizontal strip of thumbnail
                        images, each depicting different views of a
                        laptop. The thumbnails are bordered by a thin
                        line and are evenly spaced. On either side of
                        the strip, there are orange circular buttons
                        with white arrows, indicating navigation
                        options to scroll through more images. The
```

```
                                   thumbnails include images of the laptop from
                                   various angles and a box content view.\n\n##
                                   Spatial Context: The element is located
                                   centrally below the main product image within
                                   a pop-up product detail view. It is
                                   positioned above the product description and
                                   purchase options, acting as a secondary image
                                   gallery for the product.\n\n## User
                                   Interaction: The primary function of this
                                   element is to allow users to preview
                                   different images of the product. Users can
                                   click on the thumbnails to view a larger
                                   version of the image in the main display area
                                   above. The navigation arrows enable scrolling
                                   through additional images.\n\n## Element
                                   Type: Image carousel with navigation buttons."
30                         }
31                     ]
32                 }
33        ],
34        "image_id": "e54778c0-4d45-4997-a382-33244ced2513"
35 }
```

## Layout grounding data

In the layout grounding data, each data item includes a screenshot and a corresponding user description of an element as input, and produces a bounding box of corresponding element as output.

An example in JSON format is shown below. The associated image (Figure 11b) displays the layout of adding a Microsoft account, and the description refers to the text input box that allows users to input their phone number, email, or Skype ID.

```
1  {
2      "conversations": [
3          {
4              "from": "system",
5              "value": {
6                  "text": "Operating as a GUI element localization
                           agent, your primary function is to process
                           screenshots and element descriptions to identify
                           the exact position of interface components. You
                           work with coordinates (x, y, width, height)."
7              }
8          },
9          {
10             "from": "user",
11             "value": [
12                 {
13                     "image": "06c6786b..."(Truncated due to layout
                             constraints),
14                     "width": 1280,
15                     "height": 720,
16                     "resized_height": 728,
17                     "resized_width": 1288,
18                     "seq_len": 1198
19                 },
20                 {
21                     "text": "The Text input box's intended
                             function:\nThe primary function of this
                             element is to allow users to input their
                             phone number, email, or Skype ID as part of
                             the Microsoft account sign-in process. Users
                             are expected to click or tap on the box to
                             activate it and then type their credentials."
22                 }
```

```
23          ]
24      },
25      {
26          "from": "assistant",
27          "value": [
28              {
29                  "text": "492 344 296 32"
30              }
31          ]
32      }
33    ],
34    "image_id": "db53ba9b -7416 -4a01 -bb19 - e2a66d3a666d"
35 }
```

For demonstration purposes, we only show one round of dialogue in each "conversations" entry. In practice, to speed up training and save storage space, multiple rounds of dialogue related to a single screenshot are consolidated into a single JSON object's "conversations" list.

## A.5 Additional Case Studies

In this section, we explain more logic behind the models' actions and analyze the factors that contribute to their improvements or failures.

### A.5.1 Cases of improvement

**Case 12a** Leveraging training on icon, component, and layout data, JEDI places greater emphasis on keywords like `button` and `on the left`. In contrast, Qwen concentrates more on literal text matching, resulting in clicking on `Sheet1`.

**Case 12b** JEDI effectively identifies the correct icon with the specified function from numerous elements on the screen, showcasing its deep understanding of common icon functionalities. In contrast, traditional models often struggle to learn the association between icons and their functions when trained with coarse-grained data.

**Case 12c** To execute this example correctly, models must thoroughly understand both the specific component (what constitutes a `horizontal scroll bar`) and the overall layout (where the scroll bar is located). The Qwen model, however, interacted with an unrelated element.

**Case 12d** We found that the base model, which has not been trained on components and layouts, may not accurately manage subpages such as pop-ups and message bars. In contrast, JEDI successfully identifies clickable text links.

**Case 12e** This task involves having the model click on a specific mathematical symbol. Although the Qwen model demonstrates strong mathematical skills, these abilities do not improve its GUI grounding capability without fine-tuning on decomposed GUI data.

**Case 12f** The GUI for this task includes a variety of elements and complex functions. However, JEDI successfully identified the area relevant to mode switching through precise text matching.

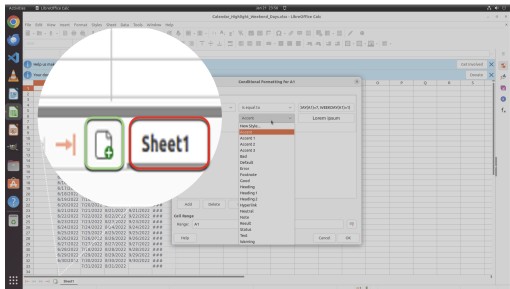

(a) Instruction: Add a sheet by clicking the button on the left of "Sheet1".

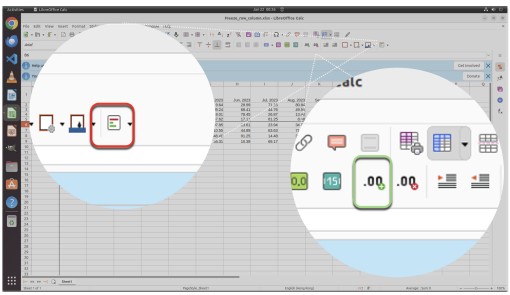

(b) Instruction: Add Decimal Place for the current cell.

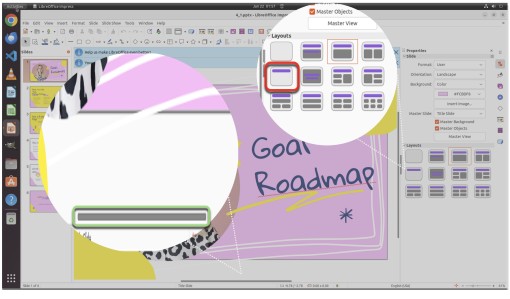

(c) Instruction: Drag the horizontal scroll bar to center the image in the viewing area.

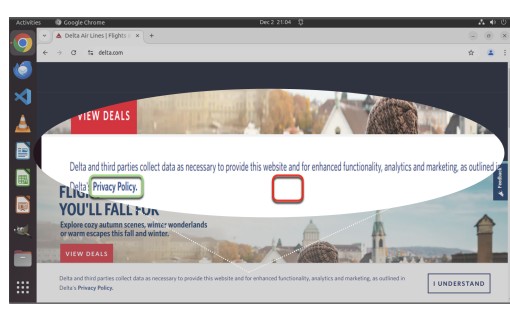

(d) Instruction: Check the privacy policy of delta.com.

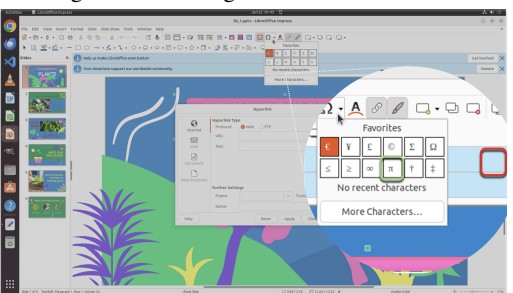

(e) Instruction: Click on the character of PI.

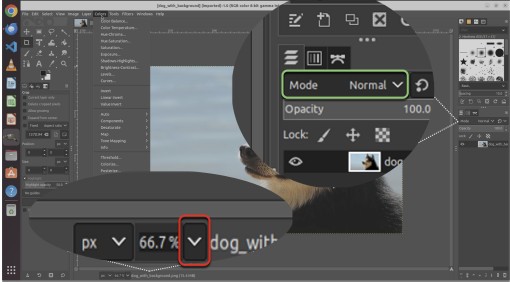

(f) Instruction: Change the mode of this image.

Figure 12: Additional cases demonstrate JEDI's improvement compared to Qwen2.5-VL-7B-Instruct. The green square represents the click position of JEDI, while the red square indicates the click position of Qwen.

## A.5.2 Cases of failure

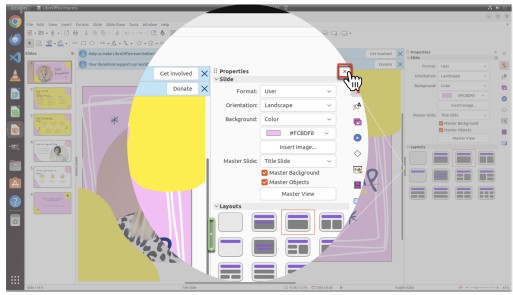

(a) Instruction: Collapse the Properties panel by clicking on the right arrow.

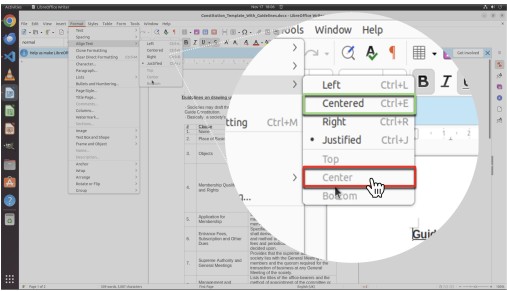

(b) Instruction: Align the text to the center.

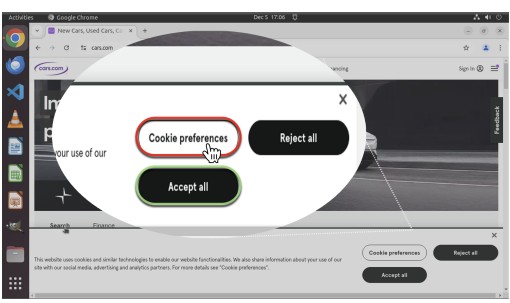

(c) Instruction: Accept the cookie preferences.

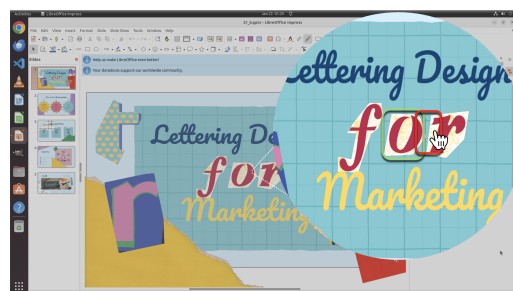

(d) Instruction: Click on the letter "o" of the word "for" in the slide.

Figure 13: Four OSWORLD-G failure cases for JEDI.

JEDI still faces challenges in certain situations. We present a selection of representative examples, with solutions to these challenges reserved for future work.

**Case 13a**    The task required the model to click the right-pointing arrow to close the right panel. Instead, JEDI clicked the `x` button. Although this action was functionally correct, it did not strictly follow the requirement to `click the right arrow`. Therefore, we classify this as a failure case.

**Case 13b**    The interface for this task shows the word `center` twice. One is displayed in white, indicating it is active, while the other is in gray, indicating it is inactive. JEDI misunderstood the color difference and mistakenly clicked the lower, inactive `center` option.

**Case 13c**    In this task, JEDI was instructed to accept the cookie options. However, it mistakenly focused on the `cookie preferences` option. We believe the wording in the instruction misled the model.

**Case 13d**    This task required clicking on a letter in an artistic font, highlighting the model's current limitations in handling grounding tasks involving artistic or stylized designs. This shortcoming may arise from the lack of art and design data in the current training corpus.

## A.6    Agentic Benchmark Results of JEDI

Table 15: Detailed performance of JEDI on OSWorld with four runs for each configuration

| Model | Configuration | Overall | OS | Calc | Impress | Writer | VLC | TB | Chrome | VSC | GIMP | Workflow | OS | Office | Daily | Pro | Workflow |
|---|---|---|---|---|---|---|---|---|---|---|---|---|---|---|---|---|---|
| | **15 Steps** | | | | | | | | | | | | | | | | |
| | Run 1 | 21.95 | 39.13 | 6.38 | 8.57 | 26.08 | 29.41 | 20.00 | 26.09 | 56.52 | 50.00 | 11.83 | 39.13 | 11.13 | 25.64 | 53.06 | 11.83 |
| | Run 2 | 22.76 | 39.13 | 6.38 | 12.83 | 17.38 | 29.41 | 33.33 | 28.06 | 60.87 | 53.85 | 9.68 | 39.13 | 11.13 | 29.37 | 57.14 | 9.68 |
| | Run 3 | 22.37 | 43.48 | 6.38 | 12.77 | 17.38 | 17.65 | 33.33 | 25.94 | 56.52 | 53.85 | 11.39 | 43.48 | 11.11 | 25.55 | 55.10 | 11.39 |
| | Run 4 | 22.36 | 39.13 | 8.51 | 10.70 | 34.77 | 32.72 | 26.67 | 21.54 | 60.87 | 50.00 | 8.60 | 39.13 | 14.55 | 24.96 | 55.10 | 8.60 |
| | *Pass@4* | 32.50 | 60.87 | 12.77 | 14.89 | 43.48 | 47.06 | 33.33 | 34.78 | 78.26 | 73.08 | 15.05 | 60.87 | 19.66 | 37.18 | 75.51 | 15.05 |
| | *Avg* | 22.36 | - | - | - | - | - | - | - | - | - | - | - | - | - | - | - |
| | **50 Steps** | | | | | | | | | | | | | | | | |
| JEDI-3B | Run 1 | 23.83 | 47.83 | 6.38 | 10.70 | 26.08 | 29.41 | 33.33 | 32.61 | 52.17 | 53.85 | 10.48 | 47.83 | 11.99 | 32.05 | 53.06 | 10.48 |
| | Run 2 | 24.73 | 43.48 | 8.70 | 12.77 | 39.12 | 23.53 | 40.00 | 30.43 | 47.83 | 64.00 | 9.78 | 43.48 | 16.54 | 30.77 | 56.25 | 9.78 |
| | Run 3 | 23.61 | 45.45 | 6.38 | 10.64 | 43.47 | 29.41 | 33.33 | 26.09 | 47.83 | 53.85 | 10.75 | 45.45 | 15.38 | 28.21 | 51.02 | 10.75 |
| | Run 4 | 22.36 | 39.13 | 8.51 | 12.77 | 21.73 | 23.53 | 33.33 | 23.91 | 56.52 | 53.85 | 9.68 | 39.13 | 13.46 | 25.64 | 55.10 | 9.68 |
| | *Pass@4* | 33.33 | 52.17 | 10.64 | 12.77 | 56.52 | 47.06 | 46.67 | 36.96 | 78.26 | 76.92 | 15.05 | 52.17 | 20.51 | 41.03 | 77.55 | 15.05 |
| | *Avg* | 23.63 | - | - | - | - | - | - | - | - | - | - | - | - | - | - | - |
| | **100 Steps** | | | | | | | | | | | | | | | | |
| | Run 1 | 24.43 | 38.10 | 8.51 | 13.11 | 26.08 | 23.53 | 46.67 | 32.40 | 43.48 | 68.00 | 9.68 | 38.10 | 13.82 | 33.21 | 56.25 | 9.68 |
| | Run 2 | 25.19 | 40.91 | 8.51 | 14.95 | 39.12 | 29.41 | 40.00 | 34.58 | 39.13 | 56.00 | 12.62 | 40.91 | 17.12 | 34.49 | 47.92 | 12.62 |
| | Run 3 | 23.66 | 45.45 | 10.64 | 17.47 | 30.42 | 23.53 | 20.00 | 25.88 | 56.52 | 53.85 | 9.78 | 45.45 | 17.27 | 24.24 | 55.10 | 9.78 |
| | Run 4 | 22.74 | 39.13 | 6.38 | 13.11 | 21.73 | 31.32 | 20.00 | 32.40 | 56.52 | 46.15 | 11.55 | 39.13 | 12.09 | 29.78 | 51.02 | 11.55 |
| | *Pass@4* | 34.44 | 56.52 | 14.89 | 21.28 | 43.48 | 41.18 | 46.67 | 39.13 | 69.57 | 73.08 | 18.28 | 56.52 | 23.08 | 41.03 | 71.43 | 18.28 |
| | *Avg* | 24.00 | - | - | - | - | - | - | - | - | - | - | - | - | - | - | - |
| | **15 Steps** | | | | | | | | | | | | | | | | |
| | Run 1 | 22.20 | 43.48 | 8.51 | 10.70 | 30.42 | 35.29 | 26.67 | 23.71 | 56.52 | 56.00 | 5.38 | 43.48 | 13.70 | 24.00 | 56.25 | 5.38 |
| | Run 2 | 23.04 | 43.48 | 10.64 | 10.70 | 30.42 | 17.65 | 33.33 | 28.05 | 65.22 | 50.00 | 7.53 | 43.48 | 14.55 | 23.68 | 57.14 | 7.53 |
| | Run 3 | 22.42 | 34.78 | 10.64 | 10.70 | 21.73 | 29.41 | 33.33 | 25.88 | 65.22 | 42.31 | 10.75 | 34.78 | 12.84 | 26.20 | 53.06 | 10.75 |
| | Run 4 | 23.31 | 34.78 | 17.02 | 8.51 | 43.47 | 23.53 | 20.00 | 28.05 | 60.87 | 50.00 | 7.61 | 34.78 | 18.80 | 19.74 | 55.10 | 7.61 |
| | *Pass@4* | 31.86 | 52.17 | 21.28 | 10.70 | 43.48 | 41.18 | 40.00 | 39.13 | 82.61 | 65.38 | 11.83 | 52.17 | 21.37 | 31.58 | 73.47 | 11.83 |
| | *Avg* | 22.74 | - | - | - | - | - | - | - | - | - | - | - | - | - | - | - |
| | **50 Steps** | | | | | | | | | | | | | | | | |
| JEDI-7B | Run 1 | 26.06 | 30.43 | 17.02 | 8.59 | 30.42 | 40.40 | 46.67 | 34.58 | 60.87 | 61.54 | 8.60 | 30.43 | 16.27 | 38.17 | 61.22 | 8.60 |
| | Run 2 | 26.27 | 47.83 | 19.15 | 10.71 | 43.47 | 38.96 | 40.00 | 25.88 | 56.52 | 46.15 | 10.75 | 47.83 | 20.54 | 31.45 | 51.02 | 10.75 |
| | Run 3 | 23.87 | 39.13 | 14.89 | 14.95 | 43.47 | 17.65 | 33.33 | 32.40 | 56.52 | 46.15 | 5.38 | 39.13 | 20.54 | 29.36 | 51.02 | 5.38 |
| | Run 4 | 23.87 | 34.78 | 12.77 | 10.70 | 34.77 | 23.53 | 33.33 | 25.88 | 69.57 | 53.85 | 8.60 | 34.78 | 16.26 | 26.8 | 61.22 | 8.60 |
| | *Pass@4* | 35.56 | 52.17 | 25.53 | 14.89 | 56.52 | 47.06 | 53.33 | 39.13 | 86.96 | 65.38 | 13.98 | 52.17 | 27.35 | 43.59 | 75.51 | 13.98 |
| | *Avg* | 25.02 | - | - | - | - | - | - | - | - | - | - | - | - | - | - | - |
| | **100 Steps** | | | | | | | | | | | | | | | | |
| | Run 1 | 25.94 | 39.13 | 14.89 | 16.30 | 34.77 | 29.41 | 26.67 | 32.40 | 60.87 | 46.15 | 12.90 | 39.13 | 19.39 | 30.65 | 53.06 | 12.90 |
| | Run 2 | 29.40 | 52.17 | 12.77 | 14.95 | 43.47 | 29.41 | 46.67 | 36.75 | 73.91 | 57.69 | 11.83 | 52.17 | 19.68 | 37.06 | 65.31 | 11.83 |
| | Run 3 | 25.64 | 43.48 | 6.38 | 14.95 | 36.35 | 20.72 | 53.33 | 28.05 | 73.91 | 46.15 | 11.68 | 43.48 | 15.54 | 31.32 | 59.18 | 11.68 |
| | Run 4 | 26.86 | 34.78 | 10.64 | 10.71 | 39.12 | 29.41 | 46.67 | 32.40 | 78.26 | 53.85 | 11.56 | 34.78 | 16.27 | 34.49 | 65.31 | 11.56 |
| | *Pass@4* | 38.89 | 65.22 | 21.28 | 21.28 | 60.87 | 41.18 | 53.33 | 45.65 | 95.65 | 65.38 | 17.2 | 65.22 | 29.06 | 46.15 | 79.59 | 17.2 |
| | *Avg* | 27.04 | - | - | - | - | - | - | - | - | - | - | - | - | - | - | - |

Table 16: Detailed performance of JEDI on WindowsAgentArena with four runs for each configuration

| Model | Configuration | Overall | Chrome | File Explorer | Notepad | Edge | OS Settings | VLC | VS Code | Calculator | Libre Calc | Libre Writer | Paint |
|---|---|---|---|---|---|---|---|---|---|---|---|---|---|
| | **15 Steps** | | | | | | | | | | | | |
| | Run 1 | 28.86 | 0.00 | 47.37 | 50.00 | 30.77 | 60.00 | 38.10 | 45.83 | 0.00 | 4.17 | 26.30 | 33.33 |
| | Run 2 | 28.72 | 5.88 | 36.84 | 50.00 | 23.08 | 80.00 | 52.75 | 45.83 | 0.00 | 8.33 | 10.53 | 33.33 |
| | Run 3 | 29.92 | 5.88 | 47.37 | 50.00 | 30.77 | 40.00 | 42.32 | 50.00 | 0.00 | 4.17 | 26.30 | 33.33 |
| | Run 4 | 28.72 | 5.88 | 42.11 | 50.00 | 38.46 | 60.00 | 43.23 | 37.50 | 0.00 | 8.33 | 21.04 | 33.33 |
| | *Pass@4* | 41.33 | 5.88 | 57.89 | 50.00 | 38.46 | 80.00 | 57.14 | 70.83 | 0.00 | 8.33 | 42.11 | 33.33 |
| | *Avg* | 29.06 | - | - | - | - | - | - | - | - | - | - | - |
| | **50 Steps** | | | | | | | | | | | | |
| JEDI-3B | Run 1 | 32.05 | 5.88 | 47.37 | 50.00 | 30.77 | 60.00 | 43.23 | 45.83 | 0.00 | 8.33 | 31.57 | 66.67 |
| | Run 2 | 32.48 | 5.88 | 44.44 | 50.00 | 38.46 | 40.00 | 47.99 | 52.17 | 0.00 | 4.17 | 36.83 | 33.33 |
| | Run 3 | 32.05 | 0.00 | 57.89 | 0.00 | 23.08 | 60.00 | 52.75 | 50.00 | 0.00 | 8.33 | 26.30 | 33.33 |
| | Run 4 | 28.72 | 5.88 | 42.11 | 50.00 | 38.46 | 60.00 | 42.23 | 37.50 | 0.00 | 8.33 | 21.04 | 33.33 |
| | *Pass@4* | 44.00 | 5.88 | 63.16 | 50.00 | 53.85 | 60.00 | 52.38 | 75.00 | 0.00 | 8.33 | 47.37 | 66.67 |
| | *Avg* | 31.33 | - | - | - | - | - | - | - | - | - | - | - |
| | **100 Steps** | | | | | | | | | | | | |
| | Run 1 | 34.57 | 5.88 | 52.63 | 50.00 | 38.46 | 80.00 | 46.91 | 45.83 | 0.00 | 12.50 | 31.57 | 33.33 |
| | Run 2 | 30.72 | 5.88 | 57.89 | 50.00 | 30.77 | 80.00 | 33.70 | 33.33 | 0.00 | 8.33 | 36.83 | 33.33 |
| | Run 3 | 33.23 | 5.88 | 63.16 | 50.00 | 7.69 | 60.00 | 42.15 | 58.33 | 0.00 | 8.33 | 31.58 | 33.33 |
| | Run 4 | 33.61 | 6.25 | 47.37 | 50.00 | 30.77 | 80.00 | 43.23 | 58.33 | 0.00 | 8.33 | 21.04 | 66.67 |
| | *Pass@4* | 46.67 | 11.76 | 63.16 | 50.00 | 53.85 | 80.00 | 57.14 | 70.83 | 0.00 | 12.50 | 52.63 | 66.67 |
| | *Avg* | 33.03 | - | - | - | - | - | - | - | - | - | - | - |
| | **15 Steps** | | | | | | | | | | | | |
| | Run 1 | 30.00 | 5.88 | 31.58 | 50.00 | 23.08 | 40.00 | 52.38 | 41.67 | 0.00 | 8.33 | 36.83 | 66.67 |
| | Run 2 | 29.38 | 0.00 | 31.58 | 50.00 | 23.08 | 60.00 | 43.23 | 50.00 | 0.00 | 8.33 | 31.57 | 66.67 |
| | Run 3 | 31.90 | 0.00 | 42.11 | 50.00 | 38.46 | 60.00 | 42.15 | 50.00 | 0.00 | 4.17 | 36.83 | 66.67 |
| | Run 4 | 29.38 | 0.00 | 42.11 | 50.00 | 30.77 | 60.00 | 42.23 | 41.67 | 0.00 | 8.33 | 26.30 | 66.67 |
| | *Pass@4* | 42.67 | 5.88 | 52.63 | 50.00 | 46.15 | 60.00 | 57.14 | 70.83 | 0.00 | 8.33 | 52.63 | 66.67 |
| | *Avg* | 30.17 | - | - | - | - | - | - | - | - | - | - | - |
| | **50 Steps** | | | | | | | | | | | | |
| JEDI-7B | Run 1 | 32.57 | 0.00 | 52.63 | 50.00 | 30.77 | 80.00 | 46.91 | 50.00 | 0.00 | 4.17 | 26.30 | 66.67 |
| | Run 2 | 32.57 | 11.76 | 47.37 | 50.00 | 46.15 | 60.00 | 51.67 | 41.67 | 0.00 | 4.17 | 26.30 | 33.33 |
| | Run 3 | 34.05 | 0.00 | 47.37 | 50.00 | 46.15 | 80.00 | 43.23 | 50.00 | 33.33 | 4.17 | 31.57 | 66.67 |
| | Run 4 | 32.00 | 0.00 | 42.11 | 50.00 | 46.15 | 60.00 | 52.38 | 45.83 | 0.00 | 8.33 | 26.30 | 33.33 |
| | *Pass@4* | 46.00 | 11.76 | 52.63 | 50.00 | 61.54 | 80.00 | 61.90 | 70.83 | 33.33 | 8.33 | 47.37 | 66.67 |
| | *Avg* | 32.80 | - | - | - | - | - | - | - | - | - | - | - |
| | **100 Steps** | | | | | | | | | | | | |
| | Run 1 | 33.90 | 0.00 | 52.63 | 50.00 | 30.77 | 80.00 | 46.91 | 54.17 | 0.00 | 8.33 | 31.57 | 33.33 |
| | Run 2 | 34.67 | 5.88 | 47.37 | 50.00 | 38.46 | 60.00 | 52.38 | 45.83 | 0.00 | 8.33 | 36.83 | 66.67 |
| | Run 3 | 33.46 | 0.00 | 47.37 | 50.00 | 38.46 | 80.00 | 43.76 | 45.83 | 0.00 | 8.33 | 42.09 | 33.33 |
| | Run 4 | 32.67 | 5.88 | 52.63 | 50.00 | 38.46 | 40.00 | 42.86 | 45.83 | 33.33 | 8.33 | 31.57 | 33.33 |
| | *Pass@4* | 47.33 | 5.88 | 63.16 | 50.00 | 53.85 | 80.00 | 61.90 | 75.00 | 33.33 | 8.33 | 52.63 | 66.67 |
| | *Avg* | 33.68 | - | - | - | - | - | - | - | - | - | - | - |

