# OpenReview forum: "Scaling Computer-Use Grounding via User Interface Decomposition and Synthesis"
_NeurIPS.cc/2025/Datasets_and_Benchmarks_Track — NeurIPS 2025 Datasets and Benchmarks Track spotlight_

### Official Review · Reviewer_3TdT · 2025-06-02

**Rating:** 5
**Confidence:** 5

**Summary:**

This paper introduces a benchmark **OSWORLD-G** comprising 564 finely annotated samples across diverse task types including text matching, element recognition, layout understanding, and precise manipulation. Additionally, this paper released **JEDI**, the largest UI grounding dataset up to now. With the help of the dataset, they trained multi-scale models that outperform the current approach on  Screenspot pro, Screenspot v2, and OSWORLD-G. Detailed ablation and case studies have identified key factors that contribute to the improvement of the current and proposed benchmark performance.

**Additional Feedback:**

Please refer to the points **1~3** in the section **Limitations Weakness** for questions and suggestions.
The paper's contributions and evaluation results are generally positive and promising from my perspective. I encourage the authors to further improve the work by addressing the concerns outlined above.

**Dataset Code Accessibility:**

Yes

**Dataset Code Comments:**

The authors have released the test benchmark, collected data **JEDI** and fine-tuned models corresponding to the paper.
Additionally, the evaluation scripts for the proposed benchmark and existing benchmarks are available and reproducible.
The evaluation results are consistent with the statistics provided in the paper.

**Ethical Considerations:**

No, there are no or only very minor ethics concerns

**Final Justification:**

The author has addressed my concerns about ambiguity evaluation and labeling.

**Limitations Weaknesses:**

1. **Refusal task**: Although some instructions can be treated as infeasible actions through visual observations, they can often be executed via keyboard shortcuts (e.g., in Windows), as mentioned in **Line 140**.  Would labeling these cases as infeasible potentially hinder the accurate evaluation of the model’s ability to handle such scenarios? In other words, would all these tasks be treated equivalently as refusal tasks?

2. **Drag actions**: Drag actions are generally more challenging to ground due to the difficulty of determining precise locations and quantifying movement (e.g. Resizing or repositioning the target image in PowerPoint.) Could the authors provide concrete examples of how the **JEDI** dataset optimizes the data annotations through fine-grained refinement?

3. **Failure case instructions**: In **A.5.x** the authors showcase grounding failure cases, some of these failures could be attributed to the ambiguity of the instructions themselves. To address this issue, can the authors provide multiple, diverse instruction variants for each benchmark example?  Additionally, providing a success rate @ n metric might better reflect the model’s robustness to instruction variability and reduce the effect of misleading phrasing.

**Strengths Contributions:**

*  Developed a comprehensive benchmark comprising 564 examples that cover the diverse task types that previous work has overlooked.
*  Synthesize a grounding dataset that contains 4 million examples.
*  This work highlights the model’s capability to reject unreasonable instructions, a critical yet often neglected aspect in current benchmark settings.

---

> ### Author Rebuttal · Authors · 2025-07-31
>
> Thank you for your appreciation and detailed evaluation of our work. During the recent period of rapid growth in GUI agents, particularly computer-use agents, which have pushed ScreenSpot[1] performance to nearly 100%, the community has begun to view GUI grounding as a nearly solved problem. However, we aim to draw attention to the continued importance of GUI grounding in computer-use agents and highlight the challenges that remain unaddressed. We are particularly pleased that you recognize the value of our dataset and benchmark, especially your acknowledgment of OSWorld-G as a comprehensive benchmark that addresses diverse task types overlooked by previous work.
>
> We have carefully addressed each of your concerns and are providing detailed responses in the following reply:
>
> ---
> > ### **W1: Concerns about keyboard shortcuts and equivalent treatment of refusal cases**
>
> A: Thank you for this thoughtful question refusal task design.
>
> Our refusal data is systematically constructed based on our grounding datasets through careful mismatching. For each subset in our grounding data, we mismatch images with instructions from different samples within the same subset. Since interfaces within each subset belong to similar types, this approach generates realistic refusal cases that are more likely to cause model errors, compared to cross-subset mismatching which often produces obviously infeasible tasks.
>
> We conducted sampling validation for each subset using an infeasibility threshold of over 95%. From this process, we selected 10 subsets whose infeasibility rates significantly exceeded this threshold. These chosen subsets cover most categories of our grounding data, thereby ensuring comprehensive coverage of refusal scenarios. This systematic approach guarantees the correctness and quality of our refusal data.
>
> Our paper primarily focuses on revealing previously overlooked aspects of grounding such as fine-grained grounding and layout understanding, exploring how to automatically scale synthesized data with minimal human intervention to improve GUI grounding capabilities. Our goal is to enhance model accuracy in generating (x,y) coordinates to assist strong planning models like GPT-4o or serve as auxiliary training data for grounding capability improvement.
>
> From our perspective, keyboard prediction capability represents a form of knowledge that falls outside the grounding scope. In previous planner-grounder frameworks (UGround[2], OS-Atlas[3], Aguvis[4]), such capabilities are typically handled by the planner, while the grounder focuses on replacing inaccurate coordinates from the planner. Therefore, keyboard shortcuts somewhat exceed the grounding domain and our paper's scope.
>
> ---
>
> > ### **W2: Drag actions**
>
> A: Thank you for your attention to drag operation modeling! We completely agree that drag operations are among the more challenging desktop interactions to model precisely, as they involve both starting position determination and continuous trajectory/displacement quantification.
>
> We designed a specialized data generation system for simulating and precisely controlling drag operations in typical scenarios like PowerPoint. Here's our implementation approach:
>
> ```
> dx = round(random.randint(-100, 100) / screensize[0], 4)
> x_trend = "right" if dx > 0 else "left"
> dy = round(random.randint(-100, 100) / screensize[1], 4)
> y_trend = "downward" if dy > 0 else "upward"
> # corner_x, corner_y are fetched using powerpoint tool
> ...
> instruction = f"Resize the bounding box which {feature}, by dragging the handle on the {corner_name} side of the bounding box {y_trend} by {abs(dy)} and to the {x_trend} by {abs(dx)}."
> ...
> action = f"pyautogui.moveTo({round(corner_x, 4)}, {round(corner_y, 4)})\npyautogui.drag({round(dx, 4)}, {round(dy, 4)}, duration=0.5)"
> ```
>
> Through this approach, we can synthesize instruction-action pairs with explicit movement direction and quantified drag distances:
>
> **Image:** [PowerPoint interface screenshot]
> **Instruction:** "Resize the text box labeled 'Agenda' by dragging the handle located at the bottom-right corner downwards by 65 and to the left by 9."
> **Action:** `pyautogui.moveTo(1010, 360)\npyautogui.drag(-9, 65, duration=0.5)`
>
> Based on our technical solution, we can model drag operations and other complex fine-grained operations across various scenarios. While we currently cover multiple common drag scenarios (element scaling, position adjustment, etc.), time and resource constraints prevent us from covering all complex cases. We believe this direction has significant expansion potential, particularly for large-scale deployment and automated annotation frameworks in industrial systems.
>
> ---
>
> > ### **W3:  Instruction ambiguity and robustness evaluation**
>
> A: Thank you for this valuable suggestion regarding instruction robustness evaluation.
>
> We want to clarify that every evaluation sample in the Jedi dataset has undergone multiple rounds of human verification and calibration to minimize ambiguity. The four examples shown in Appendix A.5.2 all contain clearly directed instructions:
>
> - **Figure (a):** Uses definitive descriptions like "click the rightward arrow" to eliminate ambiguity from multiple ways to "close the properties panel"
> - **Figure (b):** Clicking "Center" in the green box represents the fastest, most direct way to complete the instruction on the current page
> - **Figures (c) and (d):** Both contain unambiguous instructions that human users would execute without error
>
> We have human-verified every evaluation sample to ensure descriptive instructions can directly and exclusively map to unique actions on screen.
>
> Your suggestion for multiple instruction variants and success rate@n metrics is excellent. However, grounding tasks are highly sensitive to descriptive text in instructions (as shown in Figure 5, where instruction rewording dramatically affects evaluation scores). Direct LLM-based instruction rewriting has poor controllability, while human rewriting would incur substantial additional costs, which falls outside our exploration scope.
>
> Given that grounding tasks require precise correspondence between language descriptions and visual elements, we prioritize ensuring each instruction has a clear, unambiguous mapping to screen actions rather than testing robustness across instruction variations. This approach ensures reliable evaluation while maintaining the precision necessary for effective grounding capability assessment.
>
> ---
>
> We sincerely appreciate your detailed feedback. We hope the above response can address all your concerns. If you have any questions, we are pleased to provide further clarification!
>
> ---
> ### Reference
>
> [1] Cheng, Kanzhi, et al. "Seeclick: Harnessing gui grounding for advanced visual gui agents." arXiv preprint arXiv:2401.10935 (2024).
>
> [2] Gou, Boyu, et al. ‘Navigating the Digital World as Humans Do: Universal Visual Grounding for GUI Agents’. The Thirteenth International Conference on Learning Representations, 2025.
>
> [3] Wu, Zhiyong, et al. "Os-atlas: A foundation action model for generalist gui agents." arXiv preprint arXiv:2410.23218 (2024).
>
> [4] Xu, Yiheng, et al. "Aguvis: Unified pure vision agents for autonomous gui interaction." arXiv preprint arXiv:2412.04454 (2024).

---

> > ### Comment · Reviewer_3TdT · 2025-08-05
> >
> > I thank the authors for providing the detailed explanation on *refusual tasks and drag action perfection*. Although there seems to be no revolution in the training paradigm of grounding, I think this paper scales up the current data utilities and makes contributions to the GUI automation domain. I would maintain positive with top confidence.

---

> > > ### Author Response · Authors · 2025-08-05
> > >
> > > Thank you, Reviewer 3TdT, for your thoughtful appreciation and positive evaluation of our work. Your encouraging comments mean a great deal to us. We hope that our work will serve as a valuable foundation for future research in GUI grounding. And we will try to explore other possibilities and paradigms in our future research. Once again, we sincerely appreciate the time and effort you have invested in reviewing our submission.

---

### Official Review · Reviewer_iKhP · 2025-06-15

**Rating:** 5
**Confidence:** 4

**Summary:**

This paper investigates GUI grounding problem in computer-use agents, which are currently mostly powered by vision-language models. It introduces a benchmark (OSWorld-G) and a training dataset (JEDI), and the authors fine-tune a Qwen model on the proposed dataset to demonstrate improvements. The paper also includes several experiments to support their findings.

**Additional Feedback:**

I understand the challenges involved in constructing datasets and benchmarks; however, I would strongly recommend that the authors revise the text—at least in the main body of the paper. Clarity can be further improved, and minor formatting and grammatical issues should be addressed. Additionally, all table captions should be self-explanatory. For example, when presenting a table with numerical results, it should be clear which values the reader should focus on and what the purpose of any color coding is, etc.

I also noticed that the “Broader Impacts” section in the checklist appears to be empty. The development of any form of computer agents can have societal impacts, either directly or indirectly, and this deserves careful consideration. Would the dataset and model presented in this paper contribute to scaling automated bots—for example, by helping them interact with computer interfaces or even solve CAPTCHAs to bypass basic restrictions?

**Dataset Code Accessibility:**

Yes

**Dataset Code Comments:**

The Hugging Face dataset contains the data presented in the paper, and the GitHub repository includes the code for evaluation.
The paper comes with an extensive appendix that describes many aspects of the dataset sources and the methods used to collect or generate them.

I noticed that the code for fine-tuning the Qwen model is not included in the GitHub repository. While fine-tuning well-known models has become relatively straightforward, releasing such scripts would further support the community and encourage work in this area.

**Ethical Considerations:**

No, there are no or only very minor ethics concerns

**Final Justification:**

In this work, authors introduced OSWorld-G, a benchmark for GUI grounding, and Jedi, the largest-scale GUI grounding dataset with 4 million synthesized examples. The consensus from all reviewers is that this paper is a high-impact contribution to the field of computer-use agents and vision-language grounding. I foresee many people using both the benchmark and dataset introduced by this work. I will maintain my original positive score.

**Limitations Weaknesses:**

Looking at the benchmark results for the _Refusal_ section, I suspect that the subset intended to improve this capability in the model may not be effective. The Jedi models with 3B and 7B parameters achieve the same accuracy (~7\%), which raises an important question: is the method for constructing the refusal subset actually effective? If the methodology is sound but the model still fails to learn the desired behavior, then what is the missing piece? The authors have acknowledged this limitation in the paper, but I did not see any deeper analysis or investigation into the issue. Was any analysis conducted to verify whether the generated refusal data is accurate and of high quality?


The authors present _JEDI_ as a major contribution (which it is), demonstrate clear but _limited_ improvements, yet don't seriously grapple with why M examples still leave substantial performance gaps. They largely frame remaining challenges as ``_more data needed_'' rather than exploring whether their decomposition approach captures the right abstractions for computer use.

The paper would be much stronger with a substantive discussion about:

- Why large-scale grounding data alone appears insufficient
- Whether their decomposition strategy misses critical aspects of GUI interaction
- How their findings relate to broader questions about data vs. reasoning in AI systems

I encourage authors to add a limitations section.

**Strengths Contributions:**

- The paper addresses one of the most important challenges in agentic workflow research: GUI grounding. A major bottleneck in developing agents that operate directly in the pixel space is their limited ability to accurately locate specific UI elements or understand the combination of visual and textual cues. This paper focuses on this critical issue, making a valuable contribution to the current literature on agent-based systems.

- The paper presents a training dataset called JEDI, which features a very comprehensive task decomposition and contains a total of 4 million examples. I like the authors’ approach to data collection pipleine—a combination of web crawling, synthetic data generation, and reverse engineering across various types of tasks.

- Authors fine-tuned a Qwen model on the JEDI dataset and demonstrated clear performance improvements across multiple benchmarks (ScreenSpot-v2, ScreenSpot-Pro, and their own OSWorld-G). These improvements also translate directly into enhanced agentic capabilities, with success rates on OSWorld increasing from 5% to 27%.

---

> ### Author Rebuttal · Authors · 2025-07-31
>
> Thank you for your insightful review and encouraging feedback on our work! Indeed, the GUI grounding is challenging and critical for computer-use agents. And we hope our work could draw the communities’ attention on this matter. We shown the current issues of grounding model in our benchmark and with the improved grounding capabilities, foundation models could have critical performance gain in computer-use tasks. And we are appreciate that your constructive suggestions for the manuscript and social impact.
>
> We also noticed you have some constructive questions about our work, and we're happy to elaborate further below!
>
> ---
>
> > ### **W1: Refusal subset effectiveness**
>
> A: Thank you for this insightful analysis. We first want to clarify our refusal data construction methodology and quality validation process.
>
> Our refusal data is constructed based on our grounding datasets through systematic mismatching. For each subset in our grounding data, we mismatch images with instructions from different samples. Since interfaces within each subset belong to similar types, this mismatching generates realistic and challenging refusal cases that are more likely to cause model errors, compared to cross-subset mismatching which often generates obviously infeasible tasks. We conducted sampling validation for each subset and set a threshold requiring >95% infeasibility rate. We selected 10 subsets with infeasibility rates significantly exceeding the threshold, covering most categories of our grounding data.
>
> After training with Jedi refusal data, models did show measurable improvements on OSWorld-G's refusal domain. However, refusal modeling remains a significant challenge in current GUI grounding research, as evidenced by similar findings in recent work[1] where GUI defective data achieved extremely low recall rates.
>
> The limited improvement stems from vision-language models' pretraining paradigm. GUI grounding fundamentally requires associating semantic information with visual elements ("describing what's in the image"). However, large-scale pretraining data lacks negative examples of mismatched semantic-visual information. Supervised fine-tuning data alone cannot reverse this ingrained paradigm. Additionally, the inability to output refusal instructions is influenced by VLM hallucination phenomena[2,3].
>
> We acknowledge that better refusal modeling methods may exist, such as Gemini 2.5's coordinate list output scheme (naturally supporting 0 or multiple coordinates) or RefDrone's explicit target count estimation module[4].
>
> ---
>
> > ### **W2: Millions of examples still leave substantial performance gaps, and discussion on data vs. reasoning**
>
> A: This is an excellent question that touches on fundamental AI scaling principles. We provide both empirical evidence and theoretical perspective.
>
> In Section 4.2 "Performance as Data Scaling," we demonstrate that performance continues to improve with data scaling. Through iterative development, we consistently observed version-over-version improvements when generating more templates and synthesizing additional data, suggesting substantial room for further scaling.
>
> We believe scaling is the correct path for solving these problems, rather than relying on hand-crafted priors. Our decomposition strategy aims to achieve advanced-level grounding capabilities at minimal human annotation cost, comparable to large-scale manual annotation but more efficient.
>
> In Section 4.1 "Effectiveness of Knowledge," we refined task instructions (e.g., "Open the filter function for search settings" → "Click the button that includes an icon of a funnel on the right of the 'search settings' bar") and demonstrated that refined instructions yield further improvements. This validates that additional knowledge and reasoning are indeed helpful in grounding tasks.
>
> While knowledge and reasoning boundaries in grounding are difficult to define, we acknowledge that reasoning capabilities beyond pure data scaling may be necessary. However, extensive reasoning analysis falls outside our paper's scope, which focuses on establishing strong grounding foundations through comprehensive data synthesis.
>
> We have a Limitations section (Section 7) in our current manuscript, which we will enhance in the camera-ready version to address these broader questions about data scaling limits and reasoning requirements.
>
> ---
>
> > ### **W3: Writing quality improvements and "broader impact" discussions**
>
> A: Thank you for these important suggestions for improving our manuscript.
>
> We acknowledge the formatting and grammatical issues you identified. Our color coding highlights key metrics (e.g., overall accuracy) for quick comparison across benchmarks. We will enhance table captions with clearer explanations of which values readers should focus on and the purpose of color coding. All self-explanatory requirements for tables will be addressed in the camera-ready version.
>
> We recognize the important ethical considerations you raise. Computer use research inevitably challenges existing internet defense systems while enhancing model capabilities. We acknowledge that our dataset and models could potentially contribute to automated bot scaling, including interactions with computer interfaces and security measures like CAPTCHAs.
>
> We call for synchronous security research to address these challenges and will significantly expand our 'Broader Impact' section in the camera-ready version to thoroughly discuss these implications, including specific considerations about CAPTCHA circumvention and responsible deployment practices.
>
> We appreciate your suggestion about fine-tuning scripts. While fine-tuning has become more straightforward, we will include our fine-tuning scripts in the repository to better support the community and encourage further research in this area.
>
> ---
>
> Again, we appreciate your kindly suggestions for improving our works! And we hope our responses could address your concerns. If you have any questions, we are happy to discuss it further!
>
> ---
> ### Reference
>
> [1] Zhao, Kangjia, et al. "Gui testing arena: A unified benchmark for advancing autonomous gui testing agent." arXiv preprint arXiv:2412.18426 (2024).
>
> [2] Liu, Hanchao, et al. "A survey on hallucination in large vision-language models." arXiv preprint arXiv:2402.00253 (2024).
>
> [3] Lovenia, Holy, et al. "Negative object presence evaluation (nope) to measure object hallucination in vision-language models." arXiv preprint arXiv:2310.05338 (2023).
>
> [4] Sun, Zhichao, et al. "RefDrone: A Challenging Benchmark for Referring Expression Comprehension in Drone Scenes." arXiv preprint arXiv:2502.00392 (2025).

---

> > ### Comment · Reviewer_iKhP · 2025-08-03
> >
> > I thank the authors for their response and explanations. I will maintain my positive score.
> >
> > > We believe scaling is the correct path for solving these problems.
> >
> > Regarding scaling, I might be among the few who think scaling is not the way to solve these issues. I know that you’ve observed improvements version by version, but that might be because your model has already seen everything in the test set in one form or another. The efficiency of that scaling, for me, is questionable.
> >
> > This is mostly a personal take and not a criticism of this work. I do believe we (as a community) are still missing a piece here; either in model architecture, or something more fundamental, like how we train our models.

---

> > > ### Author Response · Authors · 2025-08-04
> > >
> > > Thank you, Reviewer iKhP, for positive evaluation of our work and sharing your thoughtful perspective on scaling. We agree that the research community would benefit from exploring other possibilities/paradigms. And we will try to explore one of these in our future research. Again, we sincerely appreciate the time and effort you invested in evaluating our work and engaging in the discussion.

---

### Official Review · Reviewer_wLbK · 2025-06-25

**Rating:** 5
**Confidence:** 4

**Summary:**

Graphical user interface (GUI) grounding, which maps specific instructions to detailed positions of on-screen elements, is a critical technique for computer-use agents. However, current benchmarks either oversimplify grounding tasks as short referring expressions or rely on manually annotated data. To address the research gap, this work develops the OSWorld-G, comprising 564 finely annotated samples and synthesize the largest-scale open grounding dataset, which contains 4 million examples through multi-perspective decoupling of tasks. The multi-scale models, Jedi, demonstrate their effectiveness by outperforming existing approaches on ScreenSpot-v2, ScreenSpot-Pro, and OSWorld-G. Furthermore, it demonstrates that improved grounding with Jedi directly enhances agentic capabilities of models on complex computer tasks, improving from 5% to 27% on OSWorld.

**Dataset Code Accessibility:**

Yes

**Ethical Considerations:**

No, there are no or only very minor ethics concerns

**Final Justification:**

Thank you for the authors’ detailed responses. I maintain my positive score.

**Limitations Weaknesses:**

1. The detailed data annotation instructions for OSWorld-G should be included in the Appendix.
2. The annotaion cost of GPT should be reported since the Jedi training data requires GPT to synthesize over 4 million examples, with multiple rounds of interactions (in a React way) with GPT.

**Strengths Contributions:**

1. The presented benchmark, OSWorld-G, and Jedi data - the largest-scale open grounding dataset containing 4 million examples are both valuable to the advancing research in GUI grounding as well as the improvements of computer-use agents.
2. The authors demonstrate detailed explanations of the annotation and data synthesis process. The automatic data collection process for Jedi data is promising and may inspire future data construction work. Besides, the introduction of refusal data type may enhance agents' ability to reject impossible instructions, enabling more robust interactions.
3. Comprehensive experiments validate the performance of Jedi across multiple benchmarks, including ScreenSpot-v2, ScreenSpot-Pro, and OSWorld-G, which highlights the quality of Jedi data. It is promising to see that Jedi achieves the state-of-the-art performance on Qwen2.5-VL models, surpassing baselines such as Operator (unpublished data and model) and UI-TARS (unpublished data).
4. The manuscript is well-written, with clear explanations and illustrative demonstrations.

---

> ### Author Rebuttal · Authors · 2025-07-31
>
> Thank you for your comprehensive review and recognition of our work. We hope our work will provide valuable insights to the computer-use agent community and benefit future research.
>
> We recently conducted experiments integrating o3 as the planner within our framework. The results demonstrate that Jedi-7B with o3 achieves state-of-the-art performance on OSWorld, reaching a 51% success rate. In comparison, o3 operating as a standalone end-to-end agent achieved only 23% under identical evaluation settings. This significant performance gap further underscores the critical importance of grounding capabilities in computer-use tasks.
>
> We noticed that you have some constructive questions about our work, and we are pleased to provide the following clarifications:
>
> ---
>
> > ### **W1: Missing data annotation details for OSWorld-G in the Appendix.**
>
> A: Thank you for this valuable suggestion. We appreciate the importance of providing comprehensive annotation details for reproducibility and transparency.
>
> We have outlined the annotation process in the main text (Section 2.1.1), where we describe how we collected failing grounding cases from trajectories of state-of-the-art models and frameworks across different capability areas (icon knowledge, layout understanding, fine-grained operations, and model hallucinations).
>
> The annotation process involved several key steps:
>
> - We systematically gathered failing grounding cases from state-of-the-art model trajectories, categorizing failures by their primary grounding capability requirements.
> - Expert annotators familiar with various software applications performed initial precise annotations. Guided by these failure cases, our annotators created descriptive low-level instructions. All instructions were designed to be unambiguous and map to unique screen actions.
> - Corresponding bounding boxes were annotated for each instruction-screenshot pair.
> - Multi-round verification was conducted using strong model predictions for cases with inconsistent results.
>
> Each sample required approximately **0.5 person-hours** on average.
>
> Since PDF modifications are not permitted during rebuttal, we will include comprehensive data annotation instructions with additional implementation details, annotation guidelines, and quality control procedures in the appendix of the camera-ready version.
>
> ---
>
> > ### **W2: Missing LLM annotation cost analysis for dataset examples**
>
> A: Thank you for raising this important cost transparency question. We would like to clarify our data synthesis methodology and provide detailed cost breakdowns.
>
> We did not use ReAct-style interactions for data synthesis. GPT annotation costs were primarily incurred for component code generation, action generation, data filtering, and caption generation, with costs varying by data type. We utilized GPT-4o in our data generation pipeline, with cost calculations based on GPT-4o's token pricing. The breakdown of costs for the three parts of the Jedi dataset is as follows:
>
> - **Icon data** (0.4M samples): We used input prompts and images for icons' visual and functionality descriptions, with approximately \$0.01 cost per sample, totaling \~4,000 USD.
>
> - **Component data** (1M samples): Template-based fine-grained operations (\~40K samples) were generated using template rules for slides and sheets data with no costs, while code-rendered data (\~1M samples) had costs distributed across component rendering, action generation, and filtering at \~\$0.025 per sample, totaling \~25,000 USD.
>
> - **Layout data** (2.3M samples generated from 0.8M captions): We used GPT to generate screenshot captions. Each sample involved \~3 images (\~2,100 tokens) plus prompts (\~550 tokens) with \~250 tokens output on average, costing \~\$0.0091 per caption for a total of \~7,000 USD for 0.8M captions.
>
> The total annotation cost of using GPT-4o was approximately 36,000 USD. And we have open-sourced all these data and scripts for data synthesis to facilitate future works of computer-use agents. We will include a cost analysis section for data synthesis in the appendix of the camera-ready version.
>
> ---
>
> Again, thanks for your kindly suggestion! And if you have any further questions, we are glad to have discussions!

---

> > ### Comment · Reviewer_wLbK · 2025-08-04
> >
> > Thank you for the authors’ detailed responses. I maintain my positive score.

---

> > > ### Author Response · Authors · 2025-08-04
> > >
> > > We sincerely appreciate Reviewer wLbK's positive evaluation of our work. We are also grateful for the constructive suggestions provided to help improve our manuscript. Thank you again for your time and effort in reviewing our work and engaging in this discussion.

---

### Official Review · Reviewer_Fu2U · 2025-07-02

**Rating:** 5
**Confidence:** 4

**Summary:**

Graphical user interface (GUI) grounding plays a critical role to map natural language instruction to specific actions on GUI function, enabling better computer usage. Existing benchmarks usually fail to capture the complexity of real-world interactions for software commonsense and other skills. To this end, in this paper, authors propose OSWorld-G, a comprehensive benchmark comprising 564 high-quality samples across diverse task types for evaluation. Moreover, this paper also collect a large-scale computer use dataset, termed JEDI, over 4 million examples. By training on JEDI dataset, the improved model can significantly outperform existing approaches on ScreenSpot-v2, ScreenSpot-Pro and the proposed OSWorld-G. Experimental results also demonstrate the model trained on JEDI can enhance agentic capability of foundation model in computer use tasks.

**Dataset Code Accessibility:**

Yes

**Ethical Considerations:**

No, there are no or only very minor ethics concerns

**Limitations Weaknesses:**

1. Have authors analyzed the affect of foundation model based on proposed JEDI dataset. From Table 2, it seems that Qwen2.5-VL-7B can achieve 88.8, and on Mobile task, Qwen2.5-VL-7B is comparable to JEDI-7B. Do authors explore other different backbone networks?
2. Can author report the details or agreement about human annotations for building OSWorld-G benchmark?

**Strengths Contributions:**

1. A benchmark, termed OSWorld-G, is proposed to measure the capability of foundation model in GUI grounding tasks, including text matching, element recognition, and so on.
2. A large-scale GUI grounding dataset, termed JEDI, with 4 million examples is released. Besides, two foundation model with different scales (3B and 7B) based on JEDI are released.
3. Experimental results on multiple benchmarks, including ScreenSpot-v2, ScreenSpot-Pro and OSWorld-G demonstrate the capability of foundation model based on the proposed datasets.

---

> ### Author Rebuttal · Authors · 2025-07-31
>
> Thank you for your detailed review and valuable assessment of our work! We believe our OSWorld-G benchmark and Jedi dataset will draw the GUI agents community's attention to the importance of grounding capabilities. In recent experiments, we integrated o3 as the planner within our framework alongside Jedi. Jedi-7B with o3 achieved state-of-the-art performance on OSWorld with a 51% success rate, while o3 alone achieved only 23% under the same evaluation conditions. These results demonstrate that combining a powerful foundation model with a specialized grounding model represents a promising approach for computer-use agents. We have open-sourced our dataset, model, and relevant scripts to support future research.
>
> Besides, we have carefully considered your concerns and are addressing the specific question below:
>
> ---
>
> > ### **W1: Limited exploration of different foundation model backbones**
>
> A: Thank you for this excellent question regarding backbone model exploration. Due to time constraints, we conducted experiments primarily on **Qwen2-VL** (which has different ViT structures and training corpora compared to the base models in this paper).
>
> The experimental results demonstrate the broad effectiveness of the JeDi dataset across different architectures, and the following results are using **Qwen2-VL-7B** as base model:
>
> - **ScreenSpot-v2[1]**: Baseline 49.53% → After training on Jedi dataset 91.62% (+42.09%)
> - **OSWorld-G**: Baseline 14.89% → After training on Jedi dataset 43.97% (+29.08%)
>
> These substantial improvements align with our expectations for different foundation models' performance capabilities on the Jedi dataset and demonstrate the broad applicability of our dataset across various model architectures.
>
> Regarding the mobile domain performance, we would like to clarify that our paper focuses specifically on computer use scenarios. While we did not deliberately filter mobile-related data during collection, the lack of significant improvement on mobile tasks aligns with our expectations and research scope, as our primary contribution targets desktop/computer GUI grounding tasks.
>
> ---
>
> > ### **W2: Annotation details for OSWorld-G benchmark**
>
> A: Thank you for this important question about annotation quality control. The annotation process followed several steps:
>
> -  First, experienced annotators familiar with various software applications performed initial precise annotations. They created descriptive low-level instructions based on failed cases from current state-of-the-art model rollouts.
>     - Each instruction targeted the primary grounding capabilities needed for the specific failure case. All instructions were designed to be unambiguous and map to unique screen actions.
> - Corresponding bounding boxes were annotated for each instruction-screenshot pair.
> - Finally, we conducted multi-round verification using strong model predictions, with careful review of cases where annotations disagreed with model predictions.
>
> Each sample required approximately **0.5 person-hours** on average.
>
> We used single-pass expert annotation followed by model-assisted verification rather than traditional multi-annotator approaches.
> Due to our single-pass expert annotation approach with model-assisted verification rather than traditional multi-annotator agreement calculation, we cannot provide standard inter-annotator agreement metrics. However, our quality control process through model-assisted verification ensures high annotation quality.
>
> We acknowledge that detailed annotation procedures should be better clarified in manuscript. Since PDF modifications are not permitted during rebuttal, we will include comprehensive annotation details in both the main text and appendix of the camera-ready version.
>
> ---
>
> Thanks for your suggestions, and we hope this covers your main concerns! If anything remains unclear, we're happy to provide further clarification!
>
> ---
>
> ### Reference
>
> [1] Wu, Zhiyong, et al. "Os-atlas: A foundation action model for generalist gui agents." arXiv preprint arXiv:2410.23218 (2024).

---

### Official Review · Reviewer_c6gR · 2025-07-06

**Rating:** 5
**Confidence:** 3

**Summary:**

This paper proposes the OSWorld-G benchmark to address the limitations of existing evaluation protocols for GUI grounding tasks. To support this benchmark, the authors construct the JeDi dataset with 4 million samples using a multi-source synthesis approach. The benchmark and dataset together significantly enhance the performance of VLLMs on fine-grained operations and layout understanding tasks, and improve the efficiency of agents in executing complex computer-based tasks.

**Dataset Code Accessibility:**

Yes

**Dataset Code Comments:**

The authors provide a project overview and demonstration link, along with a downloadable dataset, inference code, and detailed reproduction instructions in the associated GitHub repository.

**Ethical Considerations:**

No, there are no or only very minor ethics concerns

**Final Justification:**

This paper proposes a novel dataset for GUI grounding tasks, covering multiple aspects of the problem and contributing positively to advancements in the field. After reviewing all the comments and considering the authors’ responses, I believe this paper should be accepted.

**Limitations Weaknesses:**

1. Many models in OSWorld-G exhibit extremely low accuracy on Refusal task (Table 4), despite the training set containing 2.6 million rejection samples, which may lead to optimization difficulties during training.
2. The success rate in OSWorld increases from 5% to 27% with the use of the GPT-4o planner (Table 5), raising the question of how to disentangle the performance gains attributed to the planner from those contributed by the grounding module.

**Strengths Contributions:**

1. OSWorld-G innovatively introduces evaluation protocols for Layout Understanding, Fine-grained Manipulation, and Infeasible tasks, filling a critical gap in GUI grounding research.
2. JeDi provides the largest GUI grounding dataset to date, where the integration of multi-source heterogeneous data facilitates improved generalization of models on complex tasks.

---

> ### Author Rebuttal · Authors · 2025-07-31
>
> Thank you for your thorough review and positive assessment of our work! We are delighted that you recognize the value of our work, including the OSWorld-G benchmark and Jedi dataset, for the GUI grounding and agents research community.
>
> During the recent period of rapid growth in GUI agents, particularly computer-use agents, which have pushed ScreenSpot[1] performance to nearly 100%, the community has begun to view GUI grounding as a nearly solved problem. However, we aim to draw attention to the continued importance of GUI grounding in computer-use agents and highlight the challenges that remain unaddressed. We believe that sharing our findings in this work, including our data scaling pipeline for grounding tasks, will benefit future research in this field.
>
> We have carefully considered your concerns and are addressing the specific question below:
>
> ---
>
> > ### **W1: Poor refusal task performance despite abundant rejection training data**
>
> A: Thank you for this insightful observation. We would like to clarify several important points regarding the refusal task performance:
>
> First, we note that after training with the refusal data in the Jedi dataset, models did show improvements in the refusal domain on OSWorld-G compared with the base model Qwen2.5VL-7B, though the absolute performance remains challenging.
>
> Second, refusal modeling in GUI grounding remains a significant challenge in current research. Recent work GUI Testing Arena[2] constructed similar GUI defective data and found extremely low recall rates for defective data (as shown in their Table 2), which aligns with our findings.
>
> The limited improvement in refusal performance compared to other domains can be attributed to the inherent paradigm established during vision-language model pretraining. GUI grounding is fundamentally a visual task where current VLM training approaches focus on associating semantic information with visual elements - essentially "describing what's in the image." However, large-scale pretraining data lacks negative examples of mismatched semantic and visual information. The limited supervised fine-tuning data is insufficient to reverse this ingrained paradigm.
>
> Additionally, the inability to output refusal instructions is partly influenced by the hallucination phenomenon in vision-language models[3, 4]. Therefore, while massive supervised fine-tuning data is one approach to improve model refusal capabilities, other methods depend on paradigm changes in pretraining data and hallucination mitigation techniques[5].
>
> We note that better modeling approaches for refusal may exist. For example, Gemini 2.5 adopts a coordinate list output scheme for object grounding tasks, allowing models to return zero or multiple coordinates for a single instruction, naturally supporting refusal modeling. RefDrone[6] introduces an explicit target count estimation module that predicts the number of targets before detection, mechanistically improving model adaptation to "no target" scenarios.
>
> In summary, our paper identifies the important problem that current models cannot accurately determine whether certain instructions are executable in screenshots, and conducts valuable exploratory research. While we find this problem has inherent complexity and challenges, this provides direction for future in-depth research and optimization.
>
> ---
>
> > ### **W2: Attribution ambiguity between planner and grounding module contributions**
>
> A: Thank you for pointing this out. It's important to understand that end-to-end approaches require handling many tasks beyond grounding, including enhancing GUI-related knowledge, improving instruction following capabilities, and training agentic ability. These aspects are beyond our work's scope.
>
> Therefore, we adopted the planner-grounder architecture used in previous works (UGround[7], OS-Atlas[8], Aguvis[9]). When we replaced the grounder from Aguvis-7B to Jedi-7B while maintaining GPT-4o as the planner, we achieved a significant improvement from 14.79% to 22.7%. Meanwhile, when GPT-4o handles both planning and grounding end-to-end, it only achieves 5% on OSWorld. The ablation study demonstrates the performance gains from improved grounding capabilities under the same planner.
>
> | Agents | | # of Steps | OS SR |
> |--------|----------|------------|-------|
> | Planner | Grounder |  |  |
> | GPT-4o | - | 15 | 5.0 |
> | GPT-4o | Aguvis-7B | 15 | 14.79 |
> | GPT-4o | Jedi-7B | 15 | 22.7 |
> | o3 | - | 15 | 9.1 |
> | o3 | Jedi-7B | 15 | 42.4 |
> | o3 | Jedi-7B | 50 | 50.6 |
>
> Additionally, we provide results using o3 as the planner with Jedi as the grounder, where we achieve state-of-the-art performance improvements from the o3 baseline. The significant improvements further demonstrate the critical role that grounding plays in computer-use agents.
>
> ---
>
> We hope these responses could address your key concerns and look forward to any additional discussion you may find helpful!
>
> ---
>
> ### References:
>
> [1] Cheng, Kanzhi, et al. "Seeclick: Harnessing gui grounding for advanced visual gui agents." arXiv preprint arXiv:2401.10935 (2024).
>
> [2] Zhao, Kangjia, et al. "Gui testing arena: A unified benchmark for advancing autonomous gui testing agent." arXiv preprint arXiv:2412.18426 (2024).
>
> [3] Liu, Hanchao, et al. "A survey on hallucination in large vision-language models." arXiv preprint arXiv:2402.00253 (2024).
>
> [4] Lovenia, Holy, et al. "Negative object presence evaluation (nope) to measure object hallucination in vision-language models." arXiv preprint arXiv:2310.05338 (2023).
>
> [5] Nguyen, Cong-Duy, et al. "CutPaste&Find: Efficient Multimodal Hallucination Detector with Visual-aid Knowledge Base." arXiv preprint arXiv:2502.12591 (2025).
>
> [6] Sun, Zhichao, et al. "RefDrone: A Challenging Benchmark for Referring Expression Comprehension in Drone Scenes." arXiv preprint arXiv:2502.00392 (2025).
>
> [7] Gou, Boyu, et al. ‘Navigating the Digital World as Humans Do: Universal Visual Grounding for GUI Agents’. The Thirteenth International Conference on Learning Representations, 2025.
>
> [8] Wu, Zhiyong, et al. "Os-atlas: A foundation action model for generalist gui agents." arXiv preprint arXiv:2410.23218 (2024).
>
> [9] Xu, Yiheng, et al. "Aguvis: Unified pure vision agents for autonomous gui interaction." arXiv preprint arXiv:2412.04454 (2024).

---

> > ### Comment · Reviewer_c6gR · 2025-08-03
> >
> > Thank you for the authors’ detailed and patient responses. Considering that this is a solid piece of work and the other reviewers have provided positive feedback, I will raise my score.

---

> > > ### Author Response · Authors · 2025-08-03
> > >
> > > We greatly appreciate Reviewer c6gR's positive feedback and consideration of an increased score. Your encouragement and constructive engagement throughout this process have been invaluable in helping us improve our work. Thank you for your thoughtful comments.

---

### Comment · Area_Chair_7yqh · 2025-08-04
**Author-Reviewer Discussions**

Dear Reviewers,

Thank you for your time and valuable feedback on this submission. The authors have submitted their responses to your comments and suggestions.

- If your concerns have been sufficiently addressed in the authors' response, we kindly ask you to update your rating accordingly.

- If you require further clarification or have additional questions for the authors, please submit them as soon as possible **before Aug 6**, which will allow the authors adequate time to respond.

Best regards,

AC

---

### Decision · Program_Chairs · 2025-09-18

**Decision:**

Accept (spotlight)

**Comment:**

This paper addresses limitations in existing benchmarks for GUI grounding—the critical task of mapping natural language instructions to specific on-screen elements for computer-use agents. To solve this, the authors introduce two main contributions: 1) **The OSWorld-G Benchmark**: A comprehensive, high-quality evaluation set comprising 564 finely annotated, diverse samples that capture the complexity of real-world computer interactions. 2) **The JeDi Dataset**: A large-scale, synthetically generated training dataset of over 4 million examples, created through a multi-source, multi-perspective approach to overcome the need for manual annotation. The results demonstrate that models trained on the JeDi dataset achieve state-of-the-art performance on established benchmarks (ScreenSpot-v2, ScreenSpot-Pro) and the new OSWorld-G benchmark.

**Importance**: Despite the recent success of GUI agents—which have nearly saturated benchmarks like ScreenSpot—GUI grounding is far from solved. Current approaches fail in complex, real-world scenarios. This work confronts this gap directly, demonstrating that progress hinges on moving beyond simplistic benchmarks. It provides a new dataset and scaling pipeline to address the unresolved challenges of fine-grained visual grounding, enabling the next generation of effective computer-use agents.

Following a successful rebuttal that addressed all concerns, all five reviewers unanimously recommend clear acceptance. This submission is well above the bar for the NeurIPS DB track and should be accepted.